# MAD-Logic: Multi-Agent Debate Enhances Symbolic Translation and Reasoning

**Haocheng Yang**[1,2,3]    **Fengxiang Cheng**[4]    **Tianjun Yao**[5]    **Mengyue Yang**[6]    **Jiajun Chai**[3]
**Xiaohan Wang**[3]    **Guojun Yin**[3]    **Wei Lin**[3]    **Soummya Kar**[7]    **Fenrong Liu**[8,*]
**Haoxuan Li**[1,9,10,*]    **Yisen Wang**[1,*]

[1]Peking University   [2]NUS   [3]Meituan   [4]UvA   [5]MBZUAI   [6]University of Bristol
[7]CMU   [8]Tsinghua University   [9]University of Oxford   [10]Institute for Decentralized AI
`haocheng_yang@u.nus.edu, fenrong@tsinghua.edu.cn`
`hxli@stu.pku.edu.cn, yisen.wang@pku.edu.cn`

## Abstract

Large language models (LLMs) struggle with complex logical reasoning. Previous methods can be briefly summarized into two pipelines: (1) translating natural language (NL) to symbolic language (SL) then reasoning via external solvers, and (2) adopting LLMs to reason directly in NL based on prompting or fine-tuning. However, we point out that on the one hand, the translation relying on a specific SL often fails to capture different important features of raw NL, leading to information loss or translation errors. On the other hand, both two pipelines have unignorable limitations. For example, the former (SL-based) methods are highly sensitive to imperfect translation, and the latter (NL-based) methods are prone to hallucinations. Motivated by this, we are the first to propose a multi-agent debate framework to leverage the strengths of different SLs and reasoning methods, achieving better performance in both translation and reasoning stages. Specifically, in the translation stage, multiple agents translate the NL into different SL and refine translations through debate. In the reasoning stage, multiple agents based on SL (obtained by the corresponding solver) and NL debate multiple rounds, with the final answer determined by majority vote. In addition, to address the inefficiency of multi-agent debates, we introduce an adaptive sparse communication strategy that prunes unnecessary interactions based on agent confidence and information gains. Extensive experiments on three datasets show that our method enhances logical QA performance while reducing computational cost. Our code is at `https://github.com/yhc-666/MAD-Logic`.

## 1 Introduction

Large language models (LLMs) have demonstrated exceptional capabilities across a wide range of tasks. However, they still face significant challenges when performing complex logical reasoning, limiting their applicability in real-world scenarios (Cheng et al., 2025). There are two categories of existing methods for logical question-answering (QA). One type of methods translate natural language (NL) problems into symbolic language (SL), such as logic programming (LP), first-order logic (FOL), or Boolean satisfiability (SAT) format, and then perform reasoning via a external logical solver using these symbolic representations, i.e., reasoning in the translated SL (Ye et al., 2023; Olausson et al., 2023). An alternate type of methods use prompting or fine-tuning strategies (Yao et al., 2023; Besta et al., 2024; Zhang et al., 2024) to enable LLMs to answer logical questions in NL directly, i.e., reasoning in NL. More details about related work is provided in Appendix A.

Previous methods generally include two stages: the symbolic translation stage and the reasoning stage. In the translation stage, existing works usually translate a problem in NL into a single, predefined SL (LP, FOL or SAT, etc.). However, each SL varies in its expressivity, and only relying on a specific SL often fails to capture different important features of raw NL, leading to information loss or translation errors (Pan et al., 2023; Ryu et al., 2025). In the reasoning stage, prior works perform reasoning either via SL solver or via LLMs, which leads a trade-off. For SL solvers, though enabling

---

*Fenrong Liu, Haoxuan Li, and Yisen Wang are the corresponding authors.

rigorous reasoning, they may fail to return a valid output when translations are imperfect (Feng et al., 2024; Callewaert et al., 2025; Liu et al., 2025a), i.e., strong reasoning, weak robustness. For direct reasoning via LLM, it can tolerate inaccurate translation, but is prone to hallucinations or logical inconsistencies in LLM itself (Liu et al., 2023a; Xu et al., 2024; 2025a), i.e., strong robustness, weak reasoning. Consequently, existing methods in single-agent struggle to simultaneously achieve strong logical reasoning and robustness to translation errors.

To address this issue, we are the first to propose an extended multi-agent framework to leverage the strengths of different symbolic languages and reasoning methods, achieving better performance in both translation and reasoning stages. Specifically, in the translation stage, we employ multiple agents, with each agent responsible for translating NL to a specific SL, and then correct the translation errors through mutual refinement, ultimately enhancing the accuracy of the translation. In the reasoning stage, we assign multiple agents perform reasoning process by using SL via solvers and NL via LLMs, and prompt them to debate multiple rounds, where they can benefit each other reasoning to achieve optimal reasoning performance.

Moreover, deploying a multi-agent debate framework suffers computational overhead and token consumption (Du et al., 2023), particularly when debates involve repetitive exchanges or redundant information sharing. To address this inefficiency, we propose an adaptive sparse communication strategy that prunes unnecessary communication by assessing the agent's confidence and information gains, allowing each agent to selectively retain only the most valuable outputs from others.

The main contribution of this paper can be summarized as follows:

- We analyze the complementarity between SL and NL reasoning paradigms, as well as the complementarity within various SL systems and NL reasoning approaches.
- We are the first to propose a multi-agent approach with an adaptive sparse communication mechanism for logical reasoning, which not only enables the absorption of advantages from multiple reasoning methods through debate but also optimizes computational efficiency and cost.
- Extensive experiments on three datasets demonstrate our method can improve the performance of logical QA while reducing the computational cost.

## 2 LOGICAL QUESTION ANSWERING PROBLEM SETUP

The task of logical question answering requires determining if a conclusion can be validly inferred from a provided set of facts and rules. For this type of problem, the model's objective is to classify the statement as true, false, or unknown. This challenge is illustrated by the example below, taken from the ProofWriter dataset (Tafjord et al., 2021):

---

**Premises:**
The bear chases the squirrel. The bear is not cold. The bear visits the cat. The bear visits the lion. The cat needs the squirrel. The lion needs the cat. The squirrel needs the lion. If something visits the lion then it visits the squirrel. If something chases the cat then the cat visits the lion.
**Rules:**
- If something visits the squirrel and it needs the lion then the lion does not chase the bear.
- If something is round and it visits the lion then the lion is not cold.
- If something visits the squirrel then it chases the cat.
- If the cat does not chase the bear then the cat visits the bear.
- If something visits the squirrel then it is not nice.
- If the bear is big then the bear visits the squirrel.
**Question:** Based on the above information, is the following statement true, false, or unknown? The squirrel does not need the lion.
**Options:** A) True    B) False    C) Unknown
**Answer:** B

---

Even with recent advancements, LLMs continue to face significant difficulties with logical reasoning, evidenced by their limited performance. For instance, prior work achieved only approximately 80% accuracy on ProofWriter (Xu et al., 2025a).

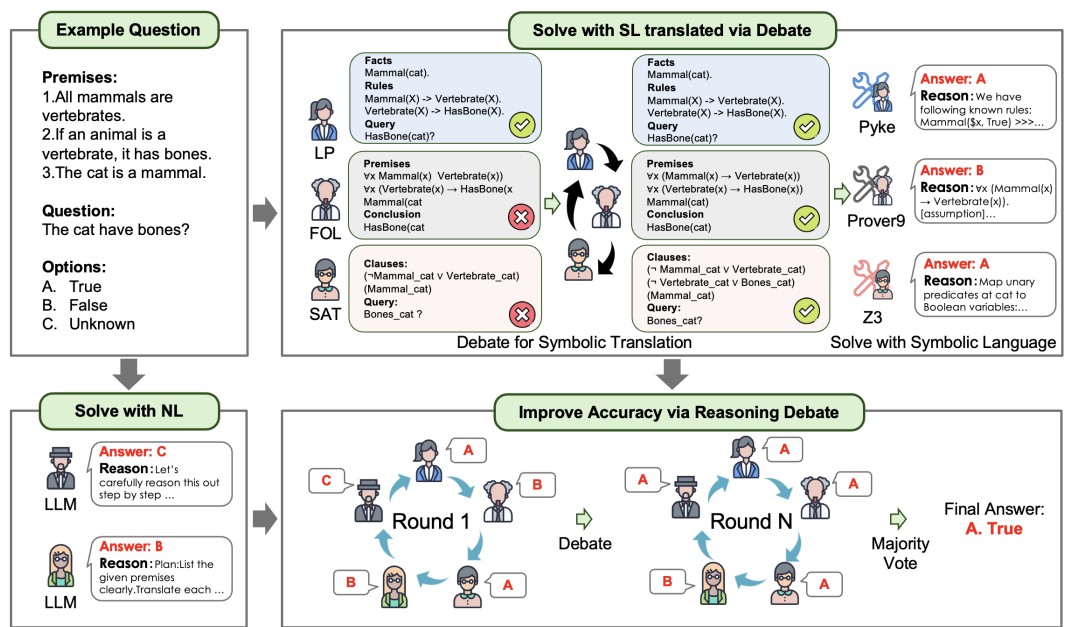

Figure 1: Overview of our sparse multi-agent debate framework for logical reasoning.

## 3 PROPOSED METHOD

### 3.1 OVERVIEW

To address the limitations of existing single-agent logical reasoning methods based on SL or NL, we propose a multi-agent debate framework. Specifically, as shown in Figure 1, we first translate NL logical questions into multiple SL, such as logic programming (LP), first-order logic (FOL), and Boolean satisfiability (SAT). Agents debate to refine their translations, ensuring translation accuracy for subsequent SL-based solving with solvers such as Pyke, Prover9, and Z3. Meanwhile, we adopt LLMs to directly solve the NL logical question based on the Chain-of-though and Plan-and-Solve techniques. Finally, agents based on results from SL and NL perform debates in multiple rounds to absorb the strengths of various methods, and a majority vote among agents is used to determine the final answer. Additionally, a sparse communication mechanism is proposed to optimize the efficiency and cost of multi-agent interactions.

### 3.2 DEBATE FOR SYMBOLIC TRANSLATION OF LOGICAL QA

To perform logical reasoning in a structured and unambiguous format, we begin by converting the raw natural language question into a formal symbolic expression. As illustrated in the top of Figure 1, this process first translates a logical reasoning question into three distinct symbolic languages (LP, FOL, and SAT) in parallel, then leverages a multi-agent debate to refine the final translations to improve the translation accuracy. In the following, we briefly introduce LP, FOL and SAT with their mutually different advantages and shortcomings, which motivates us to use them simultaneously.

**Logic Programming (LP).** Logic programming is tailored for rule-based deduction, providing a systematic framework for forward or backward inference chains. For example, a rule could be represented as $\mathrm{has\_parent}(x,y) \land \mathrm{has\_parent}(y,z) \rightarrow \mathrm{has\_grandparent}(x,z)$. While LP excels in its *brief and efficient deduction*, its *expressiveness is constrained to rule-based problems*.

**First-Order Logic (FOL).** First-Order Logic provides a highly expressive framework of representing complex relations and universal quantifiers. A typical expression might be $\forall x \forall y (\mathrm{Loves}(x,y) \rightarrow \neg \mathrm{Hates}(x,y))$. FOL's power lies in its ability to model *intricate logical structures*, but *limited to the computational complexity for large-scale problems*.

**Boolean Satisfiability (SAT).** SAT formalizes a problem as a set of Boolean variables and constraints, solvable by highly optimized solvers. An example is $A = Write(Cat), B = Write(Deer), C = Black(Cat), (A \lor B) \land (\neg A \lor C)$. This approach is *extremely efficient for*

*constraint-based problems*, though its limited expressiveness makes it *unsuitable for complex, non-Boolean logical relationships*.

### 3.3 DEBATE FOR REASONING IN SYMBOLIC LANGUAGE AND NATURAL LANGUAGE

**Reasoning via Corresponding Logical Solvers.** Given the translated symbolic languages such as LP, FOL, or SAT, solver-based reasoning methods use external logical solvers to perform logical reasoning. Despite the strong symbolic reasoning capabilities of these solvers, their effectiveness is highly sensitive to the accuracy of translation from natural to symbolic language, as even minor errors can distort solver outputs (Li et al., 2024a; Liu et al., 2025b), and information loss during translation often prevents execution, rendering the problem unsolvable (Feng et al., 2024).

---

LP example Example: LP Reasoning Extracted from Pyke Solver We have following known rules from the context:
  rule1: Sees($x, cat, True)  Green($x, False) » Sees($x, cow, True)
  rule2: Kind(rabbit, True)  Sees(rabbit, squirrel, True) » Needs(squirrel, rabbit, True)
  ... ...
Now begin reasoning to obtain all implied facts:
  Use rule1: Sees($x, cat, True)  Green($x, False) » Sees($x, cow, True)
  Use rule2: Kind(rabbit, True)  Sees(rabbit, squirrel, True) » Needs(squirrel, rabbit, True)
  ... ...
All newly implied Facts: Cold('cat', True), Cold('cow', True), Eats('squirrel', 'cow', True), Rough('cat', True), Round('cat', False), Round('cow', False), Round('squirrel', False), Sees('cat', 'rabbit', True), Sees('cow', 'rabbit', True), Sees('squirrel', 'rabbit', True)

---

**Reasoning Pipelines in Natural Language.** Prompt-based reasoning methods guide LLMs to explicitly construct logical chains during question answering, thereby producing step-by-step natural language reasoning (Wei et al., 2022; Yao et al., 2023; Zhang et al., 2023; 2024). By reasoning directly in natural language, these methods avoid rigid failures caused by symbolic translation errors, thus exhibiting high robustness. However, their reasoning ability is limited by the intrinsic capacity of LLMs, making them prone to errors on complex tasks, while repeated multi-step calls to the model incur substantial computational costs (Yang et al., 2023).

**Multi-agents' Debate to Improve the Accuracy of Reasoning.** Solver-based methods, which exhibit strong reasoning ability but low robustness, and prompt-based methods, which are highly robust but weaker in reasoning, are inherently complementary. This motivates our proposal of a multi-agent debate strategy for mutual benefit between these two paradigms, ultimately enhancing reasoning accuracy. Specifically, our approach begins by generating a set of initial natural language reasoning narratives. For the solver-based method, its symbolic reasoning process, encompassing the rules, steps, and conclusions, is visualized as a comprehensive natural language description, exemplified by a Logic Programming (LP) reasoning text from a Pyke solver. Concurrently, the prompt-based method directly outputs a narrative documenting its thought process. Subsequently, the process enters an iterative refinement loop driven by LLM. In each round, the LLM is prompted to rewrite each reasoning narrative, using all other narratives as the provided context to inform and guide its revision. This procedure is repeated for $N$ rounds (a predefined hyper-parameter), to facilitate deep interaction and mutual calibration. The final answer is then determined by a majority vote on the conclusions from all refined narratives.

### 3.4 IMPROVING EFFICIENCY VIA SPARSE DEBATE FRAMEWORK

To reduce the computational cost, we further introduce a sparse communication strategy, in which communication between agents is dynamically pruned based on a preference score. This metric assesses the potential benefit of an interaction between two LLMs in each turn by jointly considering the relative confidence of the agents and the information gains from the opponents.

#### 3.4.1 MULTI-TURN DYNAMIC INTERACTION PREFERENCE BETWEEN LLMS

We establish a sparse communication topology to improve the efficiency in multi-turn interactions by a dynamic pruning mechanism, which allows source agent $i$ to communicate its output to the receiving agent $j$ at round $d$. Specifically, we propose a preference score quantifying the potential

utility of the information in the communication, which is defined as:

$$\text{Pre}_{i\to j}^d = \frac{C_i^d}{C_j^d} + \lambda(1 - \cos(A_j^d, A_i^d||A_j^d)).$$

This score comprises two key components. The first is $C_i^d/C_j^d$, representing the ratio of confidence scores between the source agent $i$ and the receiving agent $j$ at round $d$. The second is $1 - \cos(A_j^d, A_i^d)$, measuring the difference of two outputs, regarded as information gain. The confidence score is generated by each LLM agent in the same response turn as its predicted answer. Concretely, every agent is prompted to output: (i) its predicted label, (ii) the reasoning trace, and (iii) a scalar confidence value in [0,1] during communication.

To guarantee efficiency, we propose a dynamic strategy to determine which agent should be communicated with. Specifically, in round $d$, we use this average preference score $\overline{\text{Pre}_{i\to j}^{d-1}}$ as the adaptive threshold, we define a binary communication gate $O_{i\to j}^d$. Communication from $i$ to $j$ is permitted only if the current preference score is greater than or equal to the historical average, indicating that the current interaction is at least as beneficial as the average past interaction between this pair. The indicator of whether agent $i$ benefits agent $j$ at round $d$ is formally defined as:

$$O_{i\to j}^d = \begin{cases} 1, & \text{Pre}_{i\to j}^d \ge \alpha \cdot \overline{\text{Pre}_{i\to j}^{d-1}} \\ 0, & \text{Pre}_{i\to j}^d < \alpha \cdot \overline{\text{Pre}_{i\to j}^{d-1}} \end{cases}.$$

### 3.4.2 Selective Memory Updating via Sparse Communication

The sparse communication mechanism directly informs how each agent updates its internal state or memory across debate rounds. Each agent maintains a personalized memory that aggregates valuable insights from others. At the beginning of the first round ($d = 1$), all agents start with an empty memory $M_s^1 \leftarrow \varnothing$ and communication is fully connected ($O_{i\to j}^d = 1$ for all pairs). From the second round, the sparse communication gate $O_{i\to j}^d$ is activated. At the end of each round $d$, every agent $s$ updates its memory for the next round $M_s^{d+1}$, by selectively incorporating the outputs $A_i^d$ from only those agents $i$ for which the communication channel was open (i.e., $O_{i\to j}^d = 1$). After the memory updated, agent $s$ generates its output for the next round $A_i^{d+1}$, by querying the symbolic question and $i$'s newly updated, personalized memory. After $D$ rounds of debate, the final outputs from all agents $A_1^{D+1}, \dots, A_n^{D+1}$, are aggregated via a majority vote to determine the final answer. The complete sparse communication Algorithm 1 is detailed in Appendix N.

## 4 Theoretical Analysis

We formulate Logical QA task as a multiclass classification problem. Denote the input space as $\mathcal{X}$ and output space $\mathcal{Y} = \{c_1, c_2, ..., c_k\}$ ($k \ge 2$), where $y \in \mathcal{Y}$ denotes the ground-truth label. We have a collection of $m$ agents $\mathcal{H} = \{h_1, h_2, ..., h_m\}$. For any agent $h_i$, we assume it is better than random guess, i.e., the overall accuracy $p = \mathbb{P}(h_i(x) = y) > 1/k$. For simplicity, assume uniform error answer distribution (relaxable to non-uniform with minor adjustments). For any two distinct agents $h_i, h_j$ ($i \ne j$), we define the **average pairwise class-wise correlation** $\rho \in [0, 1)$:

$$\rho = \frac{1}{\binom{k}{2}} \sum_{1 \le a < b \le k} \rho_{ab},$$

where $\rho_{ab} = \text{Cov}(Z_{i,ab}, Z_{j,ab})/\sqrt{\text{Var}(Z_{i,ab})\text{Var}(Z_{j,ab})}$, and $Z_{i,ab} = \mathbb{I}(h_i(x) = a) - \mathbb{I}(h_i(x) = b)$. The correlation $\rho$ captures how often two learners agree on answer pairs. The majority vote yields an ensemble learner $H(x) = \arg\max_{c\in\mathcal{Y}} \sum_{i=1}^m \mathbb{I}(h_i(x) = c)$, and we take random selection in case of a tie. We can show the accuracy lower bound as follows (see Appendix E for proofs).

**Theorem 1** (Accuracy Lower Bound for Majority Vote Ensemble). *Under the above setting, let $\delta = p - \frac{1-p}{k-1}$ and note that $\delta > 0$. For any incorrect class $c \ne y$, define $T_i = \mathbb{I}(h_i(x) = y) - \mathbb{I}(h_i(x) = c)$ and assume $Var(T_i) = \sigma^2$ and $Cov(T_i, T_j) = \rho\sigma^2$ for $i \ne j$, where $\rho$ is the average pairwise class-wise correlation defined above. Then the accuracy of the majority vote ensemble satisfies:*

$$\mathbb{P}(H(x) = y) \ge 1 - (k-1) \cdot \frac{\sigma^2[1 + (m-1)\rho]}{m\delta^2}.$$

*In particular, we have*

Table 1: Performance comparison across three **synthetic** benchmarks with *Temperature* set as 0.

| Methods | GPT-4 | | | Claude 3.7 Sonnet | | | DeepSeek-V3 | | |
|---|---|---|---|---|---|---|---|---|---|
| | ProntoQA | ProofWriter | LogiDeduct | ProntoQA | ProofWriter | LogiDeduct | ProntoQA | ProofWriter | LogiDeduct |
| Direct | 75.40% | 53.50% | 59.00% | 77.00% | 70.50% | 76.67% | 79.20% | 68.33% | 85.33% |
| 1-shot COT | 81.20% | 67.17% | 69.67% | 87.20% | 81.50% | 82.33% | 85.00% | 71.83% | 83.00% |
| LINC | 90.40% | 80.67% | 82.33% | 91.20% | 83.83% | 87.67% | 91.00% | 84.33% | 84.00% |
| LogicLM | 93.40% | 79.17% | 87.00% | 91.80% | 76.17% | 94.00% | 83.20% | 80.50% | 93.33% |
| Aristotle | 95.80% | 87.00% | 65.67% | 98.20% | 89.67% | 75.33% | 94.40% | 85.17% | 72.33% |
| SymbCOT | 96.00% | 82.33% | 86.33% | 97.20% | 92.33% | 94.00% | 98.00% | 85.83% | 94.00% |
| CR | 93.20% | 71.67% | 80.33% | 96.80% | 82.83% | 86.67% | 95.40% | 80.33% | 83.67% |
| DetermLR | 97.80% | 77.33% | 85.00% | 98.00% | 84.33% | 88.33% | 96.80% | 82.17% | 88.33% |
| SparseMAD | 99.50% | 89.50% | 88.67% | 99.80% | 92.83% | 99.83% | 98.00% | 92.50% | 95.33% |
| CortexDebate | 99.60% | 90.83% | 92.33% | 99.80% | 96.17% | 99.67% | 99.80% | 93.00% | 99.67% |
| Ours (w/o sparse) | 99.40% | 90.17% | 94.00% | **100.00%** | **97.00%** | 99.67% | 99.80% | 92.83% | **100.00%** |
| Ours (w/ sparse) | **100.00%** | **92.00%** | **94.33%** | **100.00%** | 96.83% | **100.00%** | **100.00%** | **93.33%** | **100.00%** |

Table 2: Performance comparison across three **real-world** benchmarks with *Temperature* set as 0.

| Methods | GPT-4 | | | DeepSeek-V3 | | |
|---|---|---|---|---|---|---|
| | AR-LSAT | FOLIO | Chinese LogiQA-V2 | AR-LSAT | FOLIO | Chinese LogiQA-V2 |
| Direct Answer | 32.90% | 65.20% | 62.27% | 36.80% | 66.18% | 74.33% |
| CoT | 35.06% | 70.59% | 65.22% | 45.45% | 76.96% | 77.97% |
| LogicLM | 40.86% | 76.96% | 25.99% | 43.72% | 78.92% | 28.63% |
| SymbCoT | 42.86% | 80.39% | 70.57% | 47.02% | 81.37% | 81.98% |
| CortexDebate | 51.08% | 84.80% | 74.13% | 74.03% | 88.73% | 83.04% |
| Ours (w/o sparse) | 50.42% | 84.31% | 74.01% | 73.62% | 89.22% | 85.68% |
| **Ours (w/ sparse)** | **53.25%** | **86.27%** | **74.76%** | **75.76%** | **90.67%** | **86.93%** |

Table 3: Performance comparison on **smaller models** using Qwen2.5-7B-Instruct.

| Method | ProofWriter | ProntoQA | LogiDeduct | AR-LSAT | FOLIO | Chinese LogiQA-V2 |
|---|---|---|---|---|---|---|
| Direct Answer | 40.17% | 55.20% | 42.67% | 24.24% | 32.35% | 62.96% |
| CoT | 40.50% | 75.60% | 40.33% | 29.87% | 53.24% | 59.32% |
| LogicLM | 61.63% | 73.80% | 63.00% | 14.29% | 59.61% | 25.33% |
| SymbCoT | 70.17% | 80.60% | 60.00% | 32.03% | 57.39% | 65.44% |
| CortexDebate (w/o NL–SL) | 47.50% | 78.20% | 39.67% | 19.91% | 53.92% | 62.96% |
| CortexDebate (w/ NL–SL) | 75.83% | 84.00% | 67.00% | 35.93% | 63.73% | 66.92% |
| Ours (w/o sparse) | 74.67% | 85.20% | 68.33% | 35.06% | 62.24% | 67.98% |
| **Ours (w/ sparse)** | **76.50%** | **86.40%** | **67.67%** | **37.20%** | **65.68%** | **68.11%** |

- *If $\rho = 0$, then $\lim_{m \to \infty} \mathbb{P}(H(x) = y) = 1$;*
- *If $\rho > 0$, then as $m \to \infty$, the accuracy lower bound converges to $1 - (k-1)\frac{\rho\sigma^2}{\delta^2}$;*
- *For any $\epsilon > 0$, if $\rho < \frac{\delta^2}{(k-1)\sigma^2}$, then there exists $m_0$ such that for all $m > m_0$, $\mathbb{P}(H(x) = y) > 1 - \epsilon$.*

This illustrates (i) if errors are independent $\rho = 0$, the lower bound goes to 1 as $m \to \infty$, and (ii) if errors are positively but moderately correlated $\rho > 0$, the bound converges to $1 - (k-1)\rho\sigma^2/\delta^2$ as $m \to \infty$, demonstrating that the majority vote remains well-behaved unless agents are highly correlated, supporting that the heterogeneous SL/NL agents have sufficiently low error correlation for the majority vote to remain well-behaved and avoid the failure mode of spurious agreement.

## 5 EXPERIMENTS

### 5.1 EXPERIMENTAL SETUP

**Datasets.** We evaluate our method on three **synthetic** benchmarks (**ProntoQA** (Saparov & He, 2022), **ProofWriter** (Tafjord et al., 2021) and **LogicalDeduction** (Srivastava et al., 2023)) and on three **real-world** benchmarks (**AR-LSAT** (Zhong et al., 2021), **FOLIO** (Han et al., 2022), and **Chinese LogiQA-V2** (Liu et al., 2023b)). More details are presented in Appendix B.

**Baselines.** We compare against nine representative methods that span different approaches, including solver-based, prompt-based and multi-agent methods. To isolate the effect of debate topology,

Table 4: Performance comparison with error bars using **3 prompt paraphrases** to incorporate randomness, **\*** indicates statistical significance using pairwise t-test ($p < 0.05$).

| Model | Method | ProofWriter | ProntoQA | LogicalDeduction | AR-LSAT | FOLIO | Chinese LogiQA-V2 |
|---|---|---|---|---|---|---|---|
| GPT-4 | w/o sparse | $89.78 \pm 0.35$ | $99.20 \pm 0.20$ | $93.78 \pm 0.19$ | $50.29 \pm 0.12$ | $84.47 \pm 0.28$ | $74.11 \pm 0.19$ |
| | CortexDebate | $90.78 \pm 0.09$ | $99.67 \pm 0.31$ | $92.44 \pm 0.51$ | $51.42 \pm 0.98$ | $85.13 \pm 0.57$ | $73.95 \pm 0.22$ |
| | w/ sparse | $\mathbf{91.83 \pm 0.17}$* | $\mathbf{99.87 \pm 0.12}$ | $\mathbf{94.61 \pm 0.26}$* | $\mathbf{53.17 \pm 0.14}$* | $\mathbf{86.60 \pm 0.43}$* | $\mathbf{74.66 \pm 0.14}$* |
| DeepSeek-V3 | w/o sparse | $92.61 \pm 0.38$ | $99.80 \pm 0.20$ | $99.11 \pm 0.51$ | $73.31 \pm 0.27$ | $89.22 \pm 0.49$ | $85.75 \pm 0.08$ |
| | CortexDebate | $92.83 \pm 0.17$ | $99.80 \pm 0.20$ | $99.67 \pm 0.33$ | $73.88 \pm 0.25$ | $89.18 \pm 0.44$ | $83.04 \pm 0.27$ |
| | w/ sparse | $\mathbf{93.50 \pm 0.14}$* | $\mathbf{99.93 \pm 0.12}$ | $\mathbf{99.89 \pm 0.19}$ | $\mathbf{75.76 \pm 0.44}$* | $\mathbf{90.85 \pm 0.28}$* | $\mathbf{86.55 \pm 0.48}$* |

Table 5: Impact of different debate components on performance

| Method | GPT-4 | | | Claude 3.7 Sonnet | | | DeepSeek-V3 | | |
|---|---|---|---|---|---|---|---|---|---|
| | ProntoQA | ProofWriter | LogiDeduct | ProntoQA | ProofWriter | LogiDeduct | ProntoQA | ProofWriter | LogiDeduct |
| w/o MA Trans. | 99.40% | 89.17% | 90.00% | 100.00% | 96.00% | 97.33% | 99.60% | 92.67% | 97.33% |
| w/o MA Rea. via SL | 95.60% | 79.33% | 84.67% | 98.00% | 83.33% | 91.00% | 96.00% | 86.17% | 93.00% |
| w/o MA Rea. via NL | 99.20% | 90.67% | 94.00% | **100.00%** | 96.67% | **100.00%** | 99.20% | 90.00% | 98.00% |
| Ours | **100.00%** | **92.00%** | **94.33%** | **100.00%** | **96.83%** | **100.00%** | **100.00%** | **93.33%** | **100.00%** |

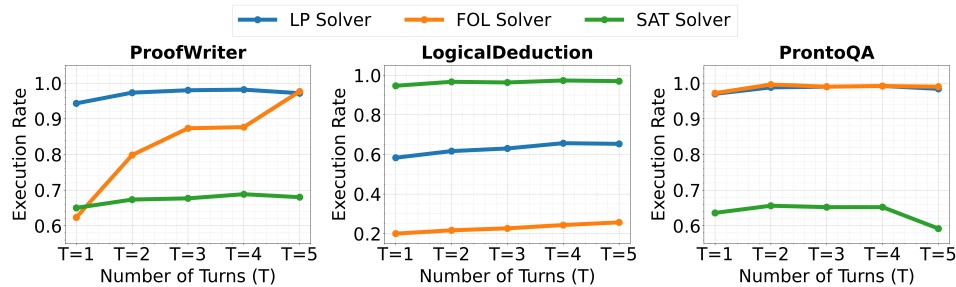

Figure 2: Relation between debate rounds and solver execution rate (GPT-4). Execution rate peaks at 2-3 rounds then declines.

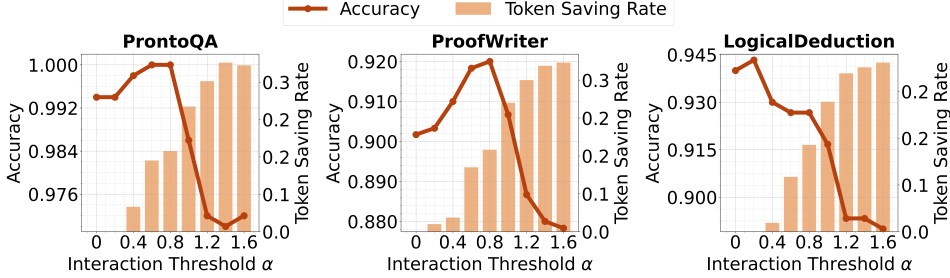

Figure 3: Effect of communication gating threshold on accuracy and token saving rate on GPT-4.

the reported results for multi-agent baseline methods adopt our proposed NL–SL hybrid reasoning stage; they differ from our method only in the debate topology. For real-world setting, we include the strongest symbolic reasoning (SymbCoT and LogicLM) and the strongest multi-agent baseline (CortexDebate). The details of baseline methods and implementations are presented in Appendix B.

**Evaluation Metrics.** We report **Accuracy**, the percentage of correctly answered logical questions.

## 5.2 MAIN RESULTS

We set *temperature* as 0 enabling deterministic reproducibility. Table 1 presents results across three logical reasoning benchmarks, and Table 2 presents results across three real-world reasoning benchmarks. experiments on Qwen2.5-7B-Instruct is in Table 3. To assess the robustness of our method, we conduct a controlled variance study by generating three semantically equivalent prompt paraphrases (using GPT-5) for each experiment. We report mean and standard deviation across the three paraphrases in Table 4. Improvements marked with **\*** are statistically significant compared to strongest competing baseline. Results for gpt-4o-mini follow the same experimental protocol and are reported in Appendix H.

Table 6: Effect of agent diversity and composition

| | | GPT-4 | | | Claude 3.7 Sonnet | | | DeepSeek-V3 | | |
|---|---|---|---|---|---|---|---|---|---|---|
| SL reasoning | NL reasoning | ProntoQA | ProofWriter | LogiDeduct | ProntoQA | ProofWriter | LogiDeduct | ProntoQA | ProofWriter | LogiDeduct |
| FOL | COT | 97.00% | 85.50% | 81.67% | 99.20% | 93.83% | 97.67% | 98.00% | 91.00% | 92.00% |
| SAT+FOL | COT | 97.20% | 86.17% | 93.00% | 99.60% | 94.00% | 99.67% | 98.40% | 92.50% | 99.67% |
| SAT+FOL+LP | COT | **100.00%** | 91.67% | 94.00% | **100.00%** | 96.17% | **100.00%** | 99.60% | 92.83% | **100.00%** |
| SAT+FOL+LP | COT+P&S | **100.00%** | **92.00%** | **94.33%** | **100.00%** | **96.83%** | **100.00%** | **100.00%** | **93.33%** | **100.00%** |

Table 7: Sensitivity to $\lambda$ in the sparse gate. "Tok" denotes token saving rate (%).

| | GPT-4 | | | | | | DeepSeek-V3 | | | | | |
|---|---|---|---|---|---|---|---|---|---|---|---|---|
| | ProofWriter | | ProntoQA | | LogicalDeduction | | ProofWriter | | ProntoQA | | LogicalDeduction | |
| $\lambda$ | Acc | Tok | Acc | Tok | Acc | Tok | Acc | Tok | Acc | Tok | Acc | Tok |
| 0 | 90.17% | 8.52% | 99.20% | 11.45% | 92.67% | 5.63% | 92.17% | 3.88% | 99.60% | 18.01% | 98.33% | 17.29% |
| 0.5 | **92.17%** | 13.62% | 99.60% | 18.31% | 94.00% | 10.29% | 93.00% | 6.02% | 99.80% | 28.82% | 98.67% | 27.66% |
| 1.0 | 92.00% | 17.03% | **100.00%** | 22.89% | **94.33%** | 12.35% | 93.33% | 13.15% | **100.00%** | 36.02% | **100.00%** | 34.57% |
| 1.5 | 91.50% | 18.73% | 99.80% | 25.18% | 93.67% | 13.02% | **93.67%** | 15.15% | 99.60% | 36.53% | 99.33% | 38.44% |
| 2.0 | 91.50% | 19.59% | **100.00%** | 26.32% | 93.33% | 13.31% | 93.33% | 15.34% | **100.00%** | 36.87% | 99.67% | 39.31% |

**Overall Performance.** Our method with sparse debate consistently outperforms all baselines across benchmarks and models. Compared to single-agent methods, we achieve substantial improvements over LogicLM, LINC, Aristotle, SymbCOT, CR, and DetermLR. Against multi-agent baselines, we surpass both SparseMAD and CortexDebate while maintaining computational efficiency (detailed token cost comparisons are provided in Appendix C).

**Sparse vs. Full Communication.** Notably, our sparse variant consistently outperforms the fully-connected version, indicating that selective communication filtering not only reduces computational costs but also mitigates noise from redundant agent interactions, leading to more effective debates. The improvements across diverse base models demonstrate the robustness of our approach.

## 5.3 ABLATION STUDY

To understand the contribution of each component in our framework, we conduct comprehensive ablation studies on both the debate stages and the agent composition.

**Impact of Debate Stages.** Table 5 ablates three debate components: (1) translation debate during NL-to-SL conversion, (2) symbolic reasoning agents, and (3) natural language reasoning agents. Removing symbolic reasoning causes the largest performance drop, followed by translation debate, confirming that formal logical reasoning is most critical while accurate symbolic translation and natural language reasoning provide complementary benefits—validating our multi-stage debate design.

**Impact of Agent Diversity.** Table 6 examines how different combinations of reasoning agents affect performance. We progressively add agents: starting from a single FOL agent with COT reasoning, we incrementally incorporate SAT, LP, and Plan&Solve agents. The results reveal improvements with each addition, demonstrating that both symbolic reasoning diversity (FOL, SAT, LP) and natural language reasoning diversity (COT, Plan&Solve) are essential for robust logical reasoning.

## 5.4 HYPERPARAMETER ANALYSIS

**Translation Debate.** Figure 2 shows executable rates of translated symbolic expressions peak at 2-3 debate rounds before degrading—a pattern consistent across all models (see Appendices K and L for other models). This degradation beyond round 3 indicates excessive debate introduces noise through over-correction of initially accurate translations. The finding validates our choice of $D = 3$ rounds. We further conducted a translation-quality study in Appendix 5.5, where we quantify symbolic translation error rates and evaluate our FOL translations against gold formulas, confirming that the translation-debate stage reliably improves NL→SL quality.

**Accuracy-Communication Sparsity Trade-off.** We investigate the impact of the communication threshold $\alpha$ on both accuracy and computational efficiency, measured as token saving rate: $(\text{Tokens}_{\text{w/o sparse}} - \text{Tokens}_{\text{w/ sparse}})/\text{Tokens}_{\text{w/o sparse}}$. Higher $\alpha$ values enforce stricter communication filtering, resulting in sparser interaction graphs. Figure 3 illustrates this trade-off for GPT-4 (see Appendices K and L for other models). A notable pattern emerges: as $\alpha$ increases, accuracy of-

Table 8: Translation common error rate T-CER$_n$ for three SL agents (LP/FOL/SAT). Values are probabilities (lower is better).

| #SL agents | GPT-4 | | | DeepSeek-V3 | | |
|---|---|---|---|---|---|---|
| | ProofWriter | ProntoQA | LogicDeduct | ProofWriter | ProntoQA | LogiDeduct |
| 1 agent | 18.50% | 10.33% | 44.56% | 17.61% | 14.53% | 53.56% |
| 2 agents | 2.50% | 1.20% | 19.33% | 1.44% | 2.87% | 13.78% |
| 3 agents | 0.33% | 0.20% | 4.00% | 0.17% | 1.00% | 5.67% |

Table 9: FOL translation quality on FOLIO. Numbers are LLM-judged semantic correctness of the FOL translations (%).

| Model | w/o translation debate | w/ translation debate |
|---|---|---|
| GPT-4o-mini | 68.14% | 75.49% |
| GPT-4 | 76.47% | 84.80% |
| DeepSeek-V3 | 75.98% | 88.24% |

ten improves while simultaneously reducing token costs by 10-30%. This suggests that moderate sparsity filters out redundant inter-agent communications that can harm reasoning quality.

**Reasoning Debate.** Figure 4 shows accuracy saturates after 2-3 debate rounds across three benchmarks, then plateauing or slightly degrading. This pattern suggests agents quickly reach consensus on logical problems, with further rounds introducing noise through overthinking or redundant arguments. The consistent 3-round optimum across datasets validates our choice of $D = 4$, balancing reasoning quality with computational efficiency.

**Sensitivity to the Sparsity Hyperparameter** $\lambda$**.** We examine the effect of the sparsity coefficient $\lambda$ in our communication gate (Table 7). Results show that the accuracy of our method remains high and stable once $\lambda \geq 0.5$ (fluctuations within $\approx 1$ pp).

## 5.5 TRANSLATION QUALITY ANALYSIS

To assess the quality of our NL $\rightarrow$ SL translations, we provide two complementary analyses:

**Translation Common Error Rate.** For each of the three SL agents and each question, we mark the translated program as correct or incorrect. For any subset of $n$ SL agents, we define T-CER$_n$ as the probability that all $n$ agents are wrong on the same question. Table 8 reports T-CER$_n$ on the three main benchmarks. In all cases, T-CER$_3$ is very small, showing that all three symbolic translators rarely fail simultaneously, which supports combining multiple heterogeneous SL agents.

**Direct Validation on FOLIO.** We further evaluate on **FOLIO**, which is one of the few datasets that provide human-annotated FOL formulas aligned with natural-language premises and hypotheses. For each example, we compare our translated FOL formula with the gold one and use an LLM judge to decide semantic equivalence, reporting the percentage of translations judged correct. As shown in Table 9, across all three base models the translation-debate stage consistently improves FOL translation accuracy, confirming that debate enhances NL $\rightarrow$ SL translation quality.

## 5.6 SOLVER TIMING ANALYSIS

Table 10 shows, for each dataset and solver (Pyke for LP, Prover9 for FOL, Z3 for SAT), the average solving time on executable instances and the timeout rate under a fixed threshold. Overall, symbolic solving rarely times out and does not dominate the computational cost of our method.

## 5.7 CASE STUDIES: MULTI-AGENT DEBATE DYNAMICS

To illustrate how our multi-agent debate framework achieves consensus through collaborative reasoning debate, we present a case study from the ProofWriter dataset shown in Table 11. The problem requires determining whether "The lion visits the lion" is true (A), false (B), or unknown (C) based on given logical rules and facts, with ground truth answer being (A). The debate showcases effective peer correction: agents with incorrect initial answers recognize their logical oversights through examining others' reasoning chains and converge to the correct solution, validating multi-

Table 10: Average solving time (seconds) and timeout rate (%) of symbolic solvers (measured on executable samples only). "–" indicates that the solver is not used on that dataset.

| Dataset | Pyke avg time | Pyke timeout | Prover9 avg time | Prover9 timeout | Z3 avg time | Z3 timeout |
|---|---|---|---|---|---|---|
| ProntoQA | 0.031 | 0.00% | 0.097 | 0.00% | 0.065 | 0.00% |
| ProofWriter | 0.028 | 0.00% | 0.102 | 0.00% | 0.065 | 0.00% |
| LogicalDeduction | 0.031 | 0.00% | 3.762 | 2.23% | 0.072 | 0.00% |
| FOLIO | – | – | 0.568 | 0.49% | 0.063 | 0.00% |
| AR-LSAT | – | – | 4.419 | 3.40% | 0.061 | 0.00% |
| Chinese LogiQA-V2 | – | – | 1.487 | 0.00% | 0.063 | 0.00% |

Table 11: Case Study of agent reasoning across debate rounds. In this case, A is the correct answer, C is the incorrect answer.

| Round | Agent | Answer | Key Reasoning |
|---|---|---|---|
| 1 | LP | A | From `Needs(Lion,Bear)` and Rule 2: `Needs($x,Bear)`→`Needs($x,Lion)`, we deduce `Needs(Lion,Lion)`. Then Rule 3: `Needs($x,Lion)`→`Visits($x,Lion)` gives us `Visits(Lion,Lion)`. |
| | FOL | C | Both proving and denying `Visits(Lion,Lion)` yield no contradiction. Neither can be conclusively established as true or false. Problem is underdefined. |
| | SAT | C | Z3 solver finds no explicit fact that implies `Visits(Lion,Lion)`. Rules do not derive it. Status: Unknown/indeterminate. |
| | P&S | A | Lion needs bear (Fact 6). By Rule 2: lion needs lion. By Rule 3: lion visits lion. Clear logical chain establishes truth. |
| | COT | C | No direct rule or statement establishes whether lion visits itself. No chain of reasoning leads to this conclusion. |
| 2 | LP | A | **FOL's claim of "underdefined" is incorrect. Rules 2→3 are deterministic, not "underdefined".** |
| | FOL | C | Still claiming Unknown despite clear logical chain shown by others. Insisting problem structure yields no definitive resolution. |
| | SAT | A | **After reviewing Plan&Solve's argument, I agree. The deduction chain is valid. Changing from Unknown to True.** |
| | P&S | A | Standing by deduction. Facts and rules suffice to establish truth. The logical steps are valid and complete. |
| | COT | A | **I acknowledging my oversight before. Plan&Solve's Rule 2+3 chain is valid.Changing answer to True.** |
| 3 | LP | A | Chain of reasoning is airtight. Therefore, lion visits itself. |
| | FOL | A | **After reassessing, I concur. The Rule 2→Rule 3 chain is valid and well-supported. Changing from Unknown to True.** |
| | SAT | A | Maintaining revised conclusion. |
| | P&S | A | Reasoning follows structured approach: Fact 6 + Rule 2 + Rule 3 = definitive True. |
| | COT | A | Agreement confirmed. Chain of logic using Fact 6, Rule 2, and Rule 3 is sound. |
| 4 | | | **All agents reach consensus: Answer A - "The lion visits the lion" is conclusively true** |

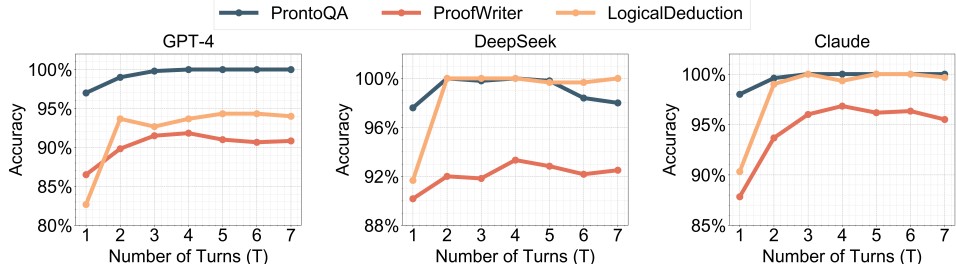

Figure 4: Relation between turns and final accuracy.

agent debate's error-correction capability. Full question and dialogues for this case can be found in Appendix Q.3. Case study for translation debate is presented in Appendix Q.2.

## 6 CONCLUSION

This paper mitigates the important limitations of large language models (LLMs) in complex logical reasoning. To our best knowledge, we are the first to propose a multi-agent approach, which enables the absorption of advantages from multiple reasoning methods through debate. Additionally, we propose a sparse communication mechanism to optimize the efficiency and cost of these multi-agent interactions. Extensive experiments on three datasets show that our method enhances logical QA performance while reducing computational cost. A limitation of this work, also serves as a future research direction, motivates from an observation that the LLM's logical reasoning performance drops significantly when handling newly released and out-of-distribution datasets (Liu et al., 2023a), thus it is crucial to extend our approach to accommodate out-of-distribution scenarios.

ACKNOWLEDGMENTS

Yisen Wang is supported by Beijing Natural Science Foundation (L257007), Beijing Major Science and Technology Project (Z251100008425006), National Natural Science Foundation of China (92370129, 62376010), Beijing Nova Program (20230484344, 20240484642), and State Key Laboratory of General Artificial Intelligence. Fenrong Liu is supported by Beijing Natural Science Foundation (L257007) and Tsinghua University's Initiative for Advancing First-Class and World-Leading Disciplines in the Humanities and Social Sciences. Haoxuan Li is supported by the Institute for Decentralized AI, a project of the Cosmos Institute funded by the AI Safety Fund.

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

## USAGE OF AI

In this work, we made limited use of LLMs as an assistive writing tool. Specifically, we used LLMs to replace synonyms, restructure sentences, and brainstorm alternative ways of expressing ideas within paragraphs. All conceptual contributions, research design, experiments, analyses, and final writing decisions were made by the authors. The authors take full responsibility for the accuracy and originality of the content.

## A  RELATED WORK

**Logical Question Answering.** The field of logical question answering seeks to enhance the reasoning capabilities of large language models and is generally pursued through three main approaches: solver-based, fine-tuning, and prompt-based methods (Cheng et al., 2025). Solver-based methods operate by converting natural language queries into formal symbolic expressions before utilizing specialized solvers for inference (Lyu et al., 2023; Olausson et al., 2023; Ye et al., 2023; Ryu et al., 2025). Fine-tuning techniques employ a dual strategy of creating synthetic datasets with explicit reasoning processes and augmenting training corpora with structured logical knowledge to embed reasoning abilities directly within model parameters (Feng et al., 2024; Morishita et al., 2024; Wan et al., 2024). Prompt-based methods explore a variety of strategies, with some generating explicit reasoning chains to guide inference (Wei et al., 2022; Yao et al., 2023; Besta et al., 2024; Zhang et al., 2024), while others direct models to produce symbolic forms for stepwise verification (Li et al., 2024a; Wang et al., 2024b; Xu et al., 2024; Liu et al., 2025b; Xu et al., 2025a). Recent work has also explored multimodal symbolic logical reasoning and step-wise logical supervision for reasoning improvement (Xu et al., 2025b; 2026). While existing research has predominantly focused on single-agent systems, our work introduces a multi-agent debate framework to synergize the complementary advantages of both SL and NL reasoning.

**Multi-Agent Debate in LLMs.** Within this domain, multi-agent debate (MAD) (Du et al., 2023) is a strategy where agents engage in iterative rounds of discussion to improve their final responses through a process of collective refinement. Research on agent roles has explored distinct reasoning modes and functional assignments, such as a proposer, a critic, a planner, and an executor, to increase diversity and reliability (Li et al., 2023; Park et al., 2023; Liang et al., 2024). The inclusion of an independent judge has been shown to enhance the factual accuracy and stability of results across tasks (Chan et al., 2023; Du et al., 2023; Estornell & Liu, 2024; Khan et al., 2024). Additionally, collaboration among heterogeneous models aims for a more robust consensus through opinion aggregation, with methods like Reconcile adding confidence-weighted voting to integrate varying viewpoints (Chen et al., 2024; Wang et al., 2024a). To address the inherent cost of these frameworks, some methods, such as SparseMAD, reduce communication by pruning the topology to a static sparse graph where agents read from fixed neighbors (Li et al., 2024b), while CortexDebate constructs a sparse debate graph with equal participation and learns edge weights using the McKinsey Trust Formula (Sun et al., 2025). Our work builds on these efforts by proposing a multi-agent debate framework that combines both symbolic and natural language reasoning, and we introduce a novel adaptive sparse communication mechanism to significantly enhance efficiency.

## B  IMPLEMENTATION DETAILS.

**Backbone models.** Our main experiments use three widely adopted and highly capable LLMs—GPT-4 (OpenAI, 2023), Claude 3.7 Sonnet (Anthropic, 2025), and DeepSeek-V3 (Wu et al., 2024). To further assess the scalability and applicability of our method to smaller and ordinary models, we also include Qwen2.5-7B-Instruct (Team, 2025) and GPT-4o-mini (OpenAI, 2024).

**Datasets.** We evaluate our method on three **synthetic** benchmarks: (1) **ProntoQA** (Saparov & He, 2022), a dataset for testing deductive reasoning over ontological knowledge with 500 test examples; (2) **ProofWriter** (Tafjord et al., 2021), We use the test set following (Pan et al., 2023), which is a set of randomly sampled 600 examples from the most challenging depth-5 subset; and (3) **LogicalDeduction** (Srivastava et al., 2023), a dataset from BIG-Bench focusing on complex deductive reasoning with ordering constraints, containing 300 test examples. These benchmarks assess different aspects of logical reasoning, from basic syllogistic inference to complex multi-hop deduction and constraint-based reasoning. In addition, we further evaluate our framework on three **real-world** benchmarks: (4) **AR-LSAT** (Zhong et al., 2021), consisting of reasoning questions from official

Table 12: Per-dataset cost-effectiveness comparison. Tokens are prefill tokens per question ($\downarrow$), accuracy is in % ($\uparrow$).

| Model | Method | ProntoQA | | ProofWriter | | LogicalDeduct | |
|---|---|---|---|---|---|---|---|
| | | Tokens $\downarrow$ | Acc (%) $\uparrow$ | Tokens $\downarrow$ | Acc (%) $\uparrow$ | Tokens $\downarrow$ | Acc (%) $\uparrow$ |
| GPT-4 | SparseMAD | 37,784 | 99.80 | 41,678 | 89.50 | **43,635** | 88.67 |
| | CortexDebate | 35,973 | 99.60 | **37,554** | 90.83 | 45,487 | 92.33 |
| | Ours (w/o sparse) | 46,502 | 99.40 | 51,358 | 90.17 | 54,857 | 94.00 |
| | **Ours (w/ sparse)** | **35,854** | **100.00** | 42,617 | **92.00** | 54,171 | **94.33** |
| Claude 3.7 | SparseMAD | 79,456 | 99.80 | 48,190 | 92.83 | **44,245** | 99.83 |
| | CortexDebate | 68,023 | 99.80 | **41,897** | 96.17 | 45,962 | 99.67 |
| | Ours (w/o sparse) | 106,015 | **100.00** | 63,105 | **97.00** | 68,636 | 99.67 |
| | **Ours (w/ sparse)** | **52,923** | **100.00** | 62,204 | 96.83 | 47,121 | **100.00** |
| DeepSeek-V3 | SparseMAD | 40,200 | 98.00 | **18,527** | 92.50 | 53,257 | 95.33 |
| | CortexDebate | **35,381** | 99.80 | 19,349 | 93.00 | 47,388 | 99.67 |
| | Ours (w/o sparse) | 57,366 | 99.80 | 25,059 | 92.83 | 70,464 | **100.00** |
| | **Ours (w/ sparse)** | 36,702 | **100.00** | 24,115 | **93.33** | **46,107** | **100.00** |

LSAT examinations; (5) **FOLIO** (Han et al., 2022), a human-authored natural language dataset annotated with first-order-logic formulas; and (6) **Chinese LogiQA-V2** (Liu et al., 2023b), a Chinese logical reasoning benchmark adapted from real civil-service examination questions.

**Baselines.** We compare against nine representative methods that span different approaches: (1) *Solver-based methods*: LogicLM (Pan et al., 2023) and LINC (Olausson et al., 2023), which translate natural language into symbolic forms for external solver processing; (2) *Prompt-based methods*: one-shot COT (Wei et al., 2022), Aristotle (Xu et al., 2025a), SymbCOT (Xu et al., 2024), CR (Cumulative Reasoning) (Zhang et al., 2023), and DetermLR (Sun et al., 2024); (3) *Multi-agent methods*: SparseMAD (Li et al., 2024b), and CortexDebate (Sun et al., 2025). To isolate the effect of debate topology, the reported results for SparseMAD and CortexDebate adopt our proposed NL–SL hybrid reasoning stage; they differ from our method only in the debate topology. For real-world setting, we include the strongest symbolic reasoning (SymbCoT and LogicLM) and the strongest multi-agent debate baseline (CortexDebate). The details of implementations are presented in Appendix B.

Our framework employs five agents in the reasoning debate stage (three symbolic reasoning agents using LP, FOL, and SAT solvers respectively, plus two natural language reasoning agents using COT and Plan-and-Solve prompting). We set the debate rounds $D = 3$ for translation and $D = 4$ for reasoning stages based on our parameter analysis (Sections 5.4). The hyperparameter $\lambda$ for balancing confidence and information gain is set to 1.0. When symbolic solvers fail to execute, we employ the "Simulate" strategy (detailed in Appendix D) where agents fall back to LLM reasoning while maintaining their symbolic perspective. The complete prompt used is detailed in the Appendix P. We use Sentence-BERT (Reimers & Gurevych, 2019) to encode agent outputs into dense embeddings for computing cosine similarity.

## C  Cost-Effectiveness Analysis

We evaluate the cost-effectiveness of our sparse communication approach by measuring token consumption and accuracy across three LLMs and three benchmarks. Following our evaluation protocol, we report *prefill tokens per question* as a reproducible cost proxy and accuracy as effectiveness; lower tokens are better ($\downarrow$), higher accuracy is better ($\uparrow$). We do not report wall-clock time due to API jitter; tokens serve as a stable, reproducible proxy for runtime and dollar cost.

In our experiments, *Ours (w/o sparse)* approximates a fully-connected debate topology where all agents communicate in each round, while *Ours (w/ sparse)* uses our adaptive sparse communication gate to selectively prune interactions based on confidence and information gains.

Our adaptive sparse gate achieves the highest accuracy while keeping token costs comparable to strong baselines. As shown in Table 12 and Table 13, our sparse method consistently outperforms the fully-connected baseline on all three models, achieving both higher accuracy (+0.92pp on GPT-

Table 13: Aggregate performance across three benchmarks. Token Saving and $\Delta$Acc are relative to *Ours (w/o sparse)*.

| Model | Method | Avg. Acc (%) ↑ | Avg. Tokens ↓ | Token Saving (%) ↑ | $\Delta$Acc (pp) ↑ |
|---|---|---|---|---|---|
| GPT-4 | SparseMAD | 92.66 | 41,032 | 19.40 | −1.87 |
| | CortexDebate | 94.39 | **39,671** | **22.07** | −0.14 |
| | Ours (w/o sparse) | 94.52 | 50,906 | 0.00 | +0.00 |
| | **Ours (w/ sparse)** | **95.44** | 44,214 | 13.15 | **+0.92** |
| Claude 3.7 | SparseMAD | 97.49 | 57,297 | 27.70 | −1.40 |
| | CortexDebate | 98.61 | **51,961** | **34.44** | −0.28 |
| | Ours (w/o sparse) | 98.89 | 79,252 | 0.00 | +0.00 |
| | **Ours (w/ sparse)** | **98.94** | 54,082 | 31.76 | **+0.05** |
| DeepSeek-V3 | SparseMAD | 95.28 | 37,328 | 26.75 | −2.27 |
| | CortexDebate | 97.49 | **34,039** | **33.21** | −0.05 |
| | Ours (w/o sparse) | 97.54 | 50,963 | 0.00 | +0.00 |
| | **Ours (w/ sparse)** | **97.78** | 35,641 | 30.06 | **+0.24** |

Table 14: Average token cost and accuracy on GPT-4.

| Method | ProntoQA | | ProofWriter | | LogicalDeduction | | AR-LSAT | | FOLIO | | Chinese LogiQA-V2 | |
|---|---|---|---|---|---|---|---|---|---|---|---|---|
| | Tokens | Acc | Tokens | Acc | Tokens | Acc | Tokens | Acc | Tokens | Acc | Tokens | Acc |
| Direct Answer | 252 | 75.40% | 315 | 53.50% | 286 | 59.00% | 144 | 32.90% | 100 | 65.20% | 355 | 62.27% |
| CoT | 303 | 81.20% | 498 | 67.17% | 261 | 69.67% | 747 | 35.06% | 177 | 70.59% | 554 | 65.22% |
| Ours (w/o sparse) | 46,502 | 99.40% | 51,358 | 90.17% | 54,857 | 94.00% | 35,012 | 50.42% | 23,167 | 84.31% | 19,015 | 74.01% |
| **Ours (w/ sparse)** | 35,854 | **100.00%** | 42,617 | **92.00%** | 43,264 | **94.67%** | 28,044 | **53.25%** | 19,023 | **86.27%** | 14,733 | **74.76%** |

4, +0.05pp on Claude 3.7, +0.24pp on DeepSeek-V3) and substantial token savings (13–36%). Remarkably, it also surpasses existing multi-agent baselines (SparseMAD and CortexDebate) in accuracy while maintaining competitive token efficiency. This demonstrates that our confidence-based pruning mechanism not only reduces computational overhead but also improves reasoning quality by filtering redundant inter-agent communications.

Token and accuracy comparison against Direct Reasoning and COT is provided in Table 14 and Table 15

## D  HANDLING SYMBOLIC SOLVER FAILURES

During the symbolic reasoning stage, solvers may occasionally fail to execute the translated logical expressions due to syntax errors, incompatible formula structures, or computational timeouts. Since our multi-agent framework relies on symbolic solvers (Pyke, Prover9, and Z3) to provide formal reasoning, handling these execution failures appropriately is crucial for maintaining system robustness.

Table 16 presents the impact of different failure handling strategies on final accuracy across three benchmarks using GPT-4. We evaluate three strategies:

- **Random**: When a solver fails, the agent randomly selects an answer from the available options. This serves as a baseline strategy.
- **Discard**: Failed solver agents are excluded from the debate, and only successfully executed agents participate in subsequent rounds and final voting.
- **Simulate**: When a solver fails, we prompt the corresponding agent to simulate the solver's reasoning process using the LLM's inherent logical capabilities, effectively falling back to natural language reasoning while maintaining the agent's role in the debate.

The results demonstrate that the *Simulate* strategy consistently achieves the best performance across all benchmarks. This approach leverages the LLM's ability to approximate symbolic reasoning when formal execution fails, maintaining full agent participation while providing reasonable fallback reasoning. The *Discard* strategy performs better than random selection but loses valuable perspectives from failed agents. These findings suggest that maintaining agent diversity through simulation is

Table 15: Average token cost and accuracy on DeepSeek-V3.

| Method | ProntoQA | | ProofWriter | | LogicalDeduction | | AR-LSAT | | FOLIO | | Chinese LogiQA-V2 | |
|---|---|---|---|---|---|---|---|---|---|---|---|---|
| | Tokens | Acc | Tokens | Acc | Tokens | Acc | Tokens | Acc | Tokens | Acc | Tokens | Acc |
| Direct Answer | 165 | 79.20% | 184 | 68.33% | 371.85 | 85.33% | 2,324 | 36.80% | 632 | 66.18% | 420 | 74.33% |
| CoT | 257 | 85.00% | 296 | 71.83% | 497.55 | 83.00% | 2,453 | 45.45% | 651 | 76.96% | 488 | 77.97% |
| Ours (w/o sparse) | 57,366 | 99.80% | 25,059 | 92.83% | 70,464 | 100.00% | 38,973 | 76.79% | 24,410 | 89.22% | 19,334 | 85.68% |
| **Ours (w/ sparse)** | 36,702 | **100.00%** | 24,115 | **93.33%** | 46,107 | **100.00%** | 30,152 | **79.65%** | 19,832 | **90.67%** | 15,547 | **86.93%** |

Table 16: Final accuracy (%) under different handling strategies when a symbolic solver fails (GPT-4).

| Strategy | ProntoQA | ProofWriter | LogicalDeduction |
|---|---|---|---|
| Random | 99.20% | 89.83% | 91.33% |
| Discard | 99.80% | 91.33% | 93.67% |
| Simulate | **100.00%** | **92.00%** | **94.33%** |

more beneficial than excluding agents, even when their primary symbolic reasoning mechanism fails.

## E  THEORETICAL ANALYSIS OF MAJORITY VOTE

**Problem Setup and Assumptions**

- **Setting**: We focus on Logical QA, which is a multiclass classification task. For simplicity, we denote the input space as $\mathcal{X}$ and output space $\mathcal{Y} = \{c_1, c_2, ..., c_k\}$ ($k \geq 2$), where $y \in \mathcal{Y}$ denotes the ground-truth label.
- We have a collection of $m$ agents $\mathcal{H} = \{h_1, h_2, ..., h_m\}$. For any agent $h_i$, we assume it is better than random guess, i.e., the overall accuracy $p = \mathbb{P}(h_i(x) = y) > 1/k$. For simplicity, assume uniform error answer distribution (relaxable to non-uniform with minor adjustments).
- For any two distinct agents $h_i, h_j$ ($i \neq j$), we define the **average pairwise class-wise correlation** $\rho \in [0, 1)$:

$$\rho = \frac{1}{\binom{k}{2}} \sum_{1 \leq a < b \leq k} \rho_{ab},$$

where $\rho_{ab} = \text{Cov}(Z_{i,ab}, Z_{j,ab})/\sqrt{\text{Var}(Z_{i,ab})\text{Var}(Z_{j,ab})}$, and $Z_{i,ab} = \mathbb{I}(h_i(x) = a) - \mathbb{I}(h_i(x) = b)$ (binary indicator for answering $a$ vs. $b$ for learner $h_i$). This captures how often two learners agree on answer pairs.
- The majority vote yields an ensemble learner

$$H(x) = \arg\max_{c \in \mathcal{Y}} \sum_{i=1}^{m} \mathbb{I}(h_i(x) = c).$$

In case of a tie, random selection is applied.

**Theorem (Accuracy Lower Bound for Majority Vote Ensemble).** Under the above setting, let $\delta = p - \frac{1-p}{k-1}$ and note that $\delta > 0$. For any incorrect class $c \neq y$, define $T_i = \mathbb{I}(h_i(x) = y) - \mathbb{I}(h_i(x) = c)$ and assume $\text{Var}(T_i) = \sigma^2$ and $\text{Cov}(T_i, T_j) = \rho\sigma^2$ for $i \neq j$, where $\rho$ is the average pairwise class-wise correlation defined above. Then the accuracy of the majority vote ensemble satisfies:

$$\mathbb{P}(H(x) = y) \geq 1 - (k - 1) \cdot \frac{\sigma^2[1 + (m-1)\rho]}{m\delta^2}.$$

In particular:

1. If $\rho = 0$, then $\lim_{m \to \infty} \mathbb{P}(H(x) = y) = 1$.
2. If $\rho > 0$, then as $m \to \infty$, the accuracy lower bound converges to $1 - (k-1)\frac{\rho\sigma^2}{\delta^2}$.
3. For any $\epsilon > 0$, if $\rho < \frac{\delta^2}{(k-1)\sigma^2}$, then there exists $m_0$ such that for all $m > m_0$, $\mathbb{P}(H(x) = y) > 1 - \epsilon$.

**Proof.**

Let:

- $S = \sum_{i=1}^{m} \mathbb{I}(h_i(x) = y)$: number of agents predicting the correct class.
- For each $c \neq y$, $S_c = \sum_{i=1}^{m} \mathbb{I}(h_i(x) = c)$: number of agents predicting class $c$.

The ensemble $H$ predicts correctly if and only if $S > S_c$ for all $c \neq y$.

We compute expectations:

- $\mathbb{E}[S] = mp$.
- Due to uniform error distribution, $\mathbb{E}[S_c] = m \cdot \frac{1-p}{k-1}$ for each $c \neq y$.

Define $\delta = p - \frac{1-p}{k-1}$. Since $p > 1/k$, we have:

$$p > \frac{1}{k} \Rightarrow kp > 1 \Rightarrow p > \frac{1-p}{k-1} \Rightarrow \delta > 0.$$

Therefore, for each $c \neq y$:

$$\mathbb{E}[S - S_c] = m\delta > 0.$$

For a fixed $c \neq y$, define $T_i = \mathbb{I}(h_i(x) = y) - \mathbb{I}(h_i(x) = c)$, so $S - S_c = \sum_{i=1}^{m} T_i$.

Compute the statistics of $T_i$:

- $\mathbb{E}[T_i] = p - \frac{1-p}{k-1} = \delta$.
- $\mathbb{E}[T_i^2] = p \cdot 1^2 + \frac{1-p}{k-1} \cdot (-1)^2 + \left(1 - p - \frac{1-p}{k-1}\right) \cdot 0 = p + \frac{1-p}{k-1}$.
- $\mathrm{Var}(T_i) = \mathbb{E}[T_i^2] - (\mathbb{E}[T_i])^2 = p + \frac{1-p}{k-1} - \delta^2 = \sigma^2$.

By assumption, for $i \neq j$, $\mathrm{Cov}(T_i, T_j) = \rho \sigma^2$.

Therefore, the variance of $S - S_c$ is:

$$\mathrm{Var}(S - S_c) = \sum_{i=1}^{m} \mathrm{Var}(T_i) + \sum_{i \neq j} \mathrm{Cov}(T_i, T_j) = m\sigma^2 + m(m-1)\rho\sigma^2 = m\sigma^2[1 + (m-1)\rho].$$

Using Chebyshev's inequality:

$$\mathbb{P}(S \leq S_c) = \mathbb{P}(S - S_c \leq 0) = \mathbb{P}((S - S_c) - m\delta \leq -m\delta)$$
$$\leq \mathbb{P}(|S - S_c - m\delta| \geq m\delta) \leq \frac{\mathrm{Var}(S - S_c)}{(m\delta)^2} = \frac{\sigma^2[1 + (m-1)\rho]}{m\delta^2}.$$

The ensemble errs if there exists some $c \neq y$ such that $S \leq S_c$. By the union bound:

$$\mathbb{P}(H(x) \neq y) \leq \sum_{c \neq y} \mathbb{P}(S \leq S_c) = (k-1) \cdot \frac{\sigma^2[1 + (m-1)\rho]}{m\delta^2}.$$

Thus:

$$\mathbb{P}(H(x) = y) \geq 1 - (k-1) \cdot \frac{\sigma^2[1 + (m-1)\rho]}{m\delta^2}.$$

We then analyze the asymptotic properties.

**1. Case $\rho = 0$.**

$$\mathbb{P}(H(x) = y) \geq 1 - (k-1) \cdot \frac{\sigma^2}{m\delta^2} \to 1 \quad \text{as } m \to \infty.$$

Therefore, $\lim_{m \to \infty} \mathbb{P}(H(x) = y) = 1$.

**2. Case $\rho > 0$.**

$$\mathbb{P}(H(x) = y) \geq 1 - (k-1) \cdot \frac{\sigma^2[1 + (m-1)\rho]}{m\delta^2}$$
$$= 1 - (k-1)\frac{\rho\sigma^2}{\delta^2} - (k-1)\frac{\sigma^2(1-\rho)}{m\delta^2}.$$

As $m \to \infty$, the lower bound converges to:

$$1 - (k-1)\frac{\rho\sigma^2}{\delta^2}.$$

**3. Arbitrary accuracy guarantee.** For any $\epsilon > 0$, if $\rho < \frac{\delta^2}{(k-1)\sigma^2}$, then:

$$1 - (k-1)\frac{\rho\sigma^2}{\delta^2} > 0,$$

and there exists $m_0$ such that for all $m > m_0$:

$$1 - (k-1) \cdot \frac{\sigma^2[1 + (m-1)\rho]}{m\delta^2} > 1 - \epsilon.$$

Hence, $\mathbb{P}(H(x) = y) > 1 - \epsilon$.

For completeness, we provide an explicit expression for $\sigma^2$:

$$\sigma^2 = p + \frac{1-p}{k-1} - \delta^2 = p + \frac{1-p}{k-1} - \left(p - \frac{1-p}{k-1}\right)^2.$$

This expression can be further simplified but is not essential for the theorem statement or proof.

**Interpretation and Corollaries** The above result yields the following important insights, which we highlight for clarity:

- **(i) If errors are independent** $\rho = 0$, the lower bound goes to 1 as $m \to \infty$.
- **(ii) If errors are positively but moderately correlated** $\rho > 0$, the bound converges to

$$1 - (k-1)\frac{\rho\sigma^2}{\delta^2} \quad \text{as } m \to \infty,$$

demonstrating that the majority vote remains well-behaved unless agents are highly correlated. This formalizes a key intuition in our system: **since our agents come from distinct SL/NL reasoning paradigms, their error correlation is substantially below the regime that leads to the failure mode of high spurious agreement.**

## F  CONSENSUS ANALYSIS VIA VOTE ENTROPY

Since we introduced a preference score to prune communication edges, here we assess whether sparse pruning affects the level of consensus reached by the agents.

For each question $q$, let $\mathcal{Y}$ be the set of answer options, $n$ the number of agents in the debate, and $c_y$ the number of agents voting for option $y$. We define the normalized vote entropy

$$H_{\text{norm}}(q) = -\frac{1}{\log|\mathcal{Y}|} \sum_{y \in \mathcal{Y}} \frac{c_y}{n} \log \frac{c_y}{n} \in [0, 1],$$

where $0$ corresponds to perfect agreement (all agents choose the same option) and $1$ corresponds to maximally split votes. We report the average $H_{\text{norm}}$ over all questions at the final debate round.

Table 17 compares the average normalized vote entropy between a fully connected debate graph (**w/o sparse**) and our sparse communication graph (**w/ sparse**) for GPT-4 and DeepSeek-V3. Sparse pruning does not much change vote entropy across datasets, and in several cases even slightly reduces it, indicating that our sparse topology preserves the consensus behavior of the debate in practice.

Table 17: Average normalized vote entropy $H_{\text{norm}} \in [0, 1]$ across questions (lower = stronger consensus), measured at the final debate round.

| Model | Method | ProofWriter | ProntoQA | LogicalDeduction | AR-LSAT | FOLIO | Chinese LogiQA-V2 |
|---|---|---|---|---|---|---|---|
| GPT-4 | w/o sparse | 0.0224 | 0.0014 | 0.0055 | 0.0881 | 0.0655 | 0.0710 |
| GPT-4 | w/ sparse | 0.0243 | 0.0014 | 0.0054 | 0.0893 | 0.0687 | 0.0733 |
| DeepSeek-V3 | w/o sparse | 0.2359 | 0.0540 | 0.0034 | 0.0346 | 0.2832 | 0.0142 |
| DeepSeek-V3 | w/ sparse | 0.2334 | 0.0555 | 0.0034 | 0.0340 | 0.2732 | 0.0150 |

Table 18: Sensitivity to aggregation rules (Accuracy %)

| Dataset | GPT-4o-mini | | | GPT-4 | | | DeepSeek-V3 | | |
|---|---|---|---|---|---|---|---|---|---|
| | Majority | Conf-Weighted | LLM-as-Judge | Majority | Conf-Weighted | LLM-as-Judge | Majority | Conf-Weighted | LLM-as-Judge |
| ProofWriter | 76.33% | 75.50% | 76.00% | 92.00% | 91.60% | 92.00% | 93.33% | 93.50% | 93.50% |
| ProntoQA | 89.60% | 88.00% | 90.40% | 100.00% | 100.00% | 100.00% | 100.00% | 100.00% | 100.00% |
| LogiDeduct | 82.33% | 82.67% | 83.33% | 94.33% | 93.67% | 95.00% | 100.00% | 99.80% | 100.00% |

## G    SENSITIVITY TO AGGREGATION RULES

To show that our results are not due to a specific voting rule, we added a sensitivity study on three aggregation rules (Table 18): (i) Majority vote (ii) Confidence-weighted vote (iii) "LLM-as-judge" (agents debate, and an independent LLM reads all rationales and produces final prediction). Across 3 base LLMs × 3 datasets, the gap between all three methods is within 1 pp. This indicates that:

- Our improvements are not due to a particular voting method.
- The gains primarily come from the SL+NL multi-agent debate and sparse communication, while the final aggregator is easily changeable.

## H    ADDITIONAL EXPERIMENTS ON SMALL / ORDINARY MODELS

To evaluate the generalization of our framework to smaller and more accessible LLMs, we conduct experiments on two compact models: Qwen2.5-7B-Instruct and GPT-4o-mini. Despite their significantly lower parameter counts, our sparse multi-agent debate framework continues to yield consistent gains across all six benchmarks on both models as shown in Table 3 and Table 19, demonstrating that our approach is not limited to large frontier LLMs. CortexDebate (w/ NL-SL) reuses our translation stage, solver stage, and agent roles—thus differing from our method only in the communication graph topology. CortexDebate (w/o NL-SL) corresponds to the original pure-NL version as used in its original work.

## I    ABLATION ON SL–NL CROSS-PARADIGM AND SPARSE COMMUNICATION

To disentangle the contributions of the SL–NL cross-paradigm design and the sparse communication topology, we provide a comprehensive ablation study. The variants are grouped into two families: (A) SL–NL cross-paradigm ablations that manipulate symbolic vs. natural language reasoning components, and (B) sparse-communication/topology ablations that vary the debate graph while keeping the number of agents fixed.

**(A) SL–NL cross-paradigm ablations.**    We consider the following variants:

- **COT + P&S only (NL-only).** Remove all symbolic translators and solvers. Only two NL agents (Chain-of-Thought and Plan-and-Solve) participate in the debate.
- **LP + FOL + SAT only (SL-only).** Remove all NL agents. Keep only the three solver-based agents (LP/Pyke, FOL/Prover9, SAT/Z3).
- **No SL–NL interaction in debate.** Keep all 5 agents, but force SL agents to debate only with SL agents and NL agents only with NL agents (two disjoint debates).
- **Translation debate rounds = 0.** Disable the translation-stage debate ($D_{\text{trans}} = 0$). SL translations are generated once and used as-is by solvers.
- **5-agent direct vote (no debate).** All 5 agents (SL and NL) answer once independently; the final answer is decided by a single majority vote without any iterative debate.

**(B) Sparse topology / communication ablations.**    We next investigate different communication topologies:

Table 19: Performance of different methods on GPT-4o-mini.

| Method | ProofWriter | ProntoQA | LogiDeduct | AR-LSAT | FOLIO | Chinese LogiQA-V2 |
|---|---|---|---|---|---|---|
| Direct Answer | 53.50% | 62.60% | 56.00% | 19.91% | 59.80% | 57.31% |
| CoT | 43.67% | 75.00% | 70.33% | 19.04% | 61.76% | 54.30% |
| LogicLM | 58.67% | 76.00% | 73.00% | 22.94% | 34.80% | 25.00% |
| SymbCoT | 70.33% | 82.80% | 75.33% | 26.00% | 69.10% | 58.87% |
| CortexDebate (w/o NL–SL) | 60.87% | 80.40% | 70.67% | 21.21% | 62.75% | 62.02% |
| CortexDebate (w/ NL–SL) | 75.17% | 89.00% | 82.33% | 34.20% | 75.00% | 64.03% |
| Ours (w/o sparse) | 74.00% | 89.40% | 82.33% | 33.81% | 74.02% | 65.91% |
| **Ours (w/ sparse)** | **76.33%** | **90.60%** | **84.67%** | **34.20%** | **76.47%** | **67.29%** |

Table 20: Ablations on (A) SL–NL cross-paradigm reasoning and (B) sparse communication strategies. Accuracy (%). Columns correspond to GPT-4 / DeepSeek-V3 on ProofWriter (PW), ProntoQA (PQA), and LogicalDeduction (LD).

| Setting / Variant | GPT4-PW | GPT4-PQA | GPT4-LD | DS-PW | DS-PQA | DS-LD |
|---|---|---|---|---|---|---|
| **(A) SL–NL Cross-Paradigm Ablations** | | | | | | |
| COT + P&S only (NL-only) | 79.33% | 95.60% | 84.67% | 86.17% | 96.00% | 93.00% |
| LP + FOL + SAT only (SL-only) | 90.67% | 99.20% | 94.00% | 90.00% | 99.20% | 98.00% |
| No SL–NL interaction in debate | 90.83% | 99.20% | 93.00% | 90.17% | 99.20% | 97.33% |
| Translation debate rounds = 0 | 89.17% | 99.40% | 90.00% | 92.67% | 99.60% | 97.33% |
| 5-agent direct vote (no debate) | 86.50% | 97.00% | 82.67% | 90.00% | 97.60% | 91.67% |
| **(B) Sparse Topology / Communication Ablations (5-agent)** | | | | | | |
| Pure NL 5-agent chat, fully-connected | 73.00% | 91.40% | 84.00% | 82.83% | 93.00% | 88.33% |
| Pure NL 5-agent chat + SparseMAD | 72.83% | 91.00% | 85.00% | 80.17% | 93.20% | 87.67% |
| Pure NL 5-agent chat + CortexDebate | 73.50% | 89.00% | 84.33% | 83.17% | 94.00% | 90.00% |
| Pure NL 5-agent chat + our sparse gate | 75.67% | 90.20% | 85.33% | 84.33% | 94.00% | 91.33% |
| Replace our sparse gate w/ SparseMAD | 89.50% | 99.80% | 88.67% | 92.50% | 98.00% | 95.33% |
| Replace our sparse gate w/ CortexDebate | 90.83% | 99.60% | 92.33% | 93.00% | 99.80% | 98.33% |
| **Ours (full SL+NL, full debate, our gate)** | **92.00%** | **100.00%** | **94.33%** | **93.33%** | **100.00%** | **100.00%** |

- **Pure NL 5-agent chat, fully-connected.** Remove all SL agents. Use 5 identical NL agents; every agent reads all others (fully connected graph).
- **Pure NL 5-agent chat + SparseMAD.** Same pure-NL setup, but replace the communication graph with SparseMAD's static neighbor topology.
- **Pure NL 5-agent chat + CortexDebate.** Same pure-NL setup, but use CortexDebate's trust-weighted sparse graph.
- **Pure NL 5-agent chat + our sparse gate.** Same pure-NL setup, but apply our confidence + information-gain based sparse gate.
- **Replace our sparse gate with SparseMAD (full SL+NL pipeline).** Use our full SL+NL pipeline (translators, solvers, NL agents), but replace our gate with the SparseMAD topology.
- **Replace our sparse gate with CortexDebate (full SL+NL pipeline).** Same full SL+NL pipeline, but use CortexDebate's learned trust graph.
- **Ours (full SL+NL, full debate, our gate).** The complete proposed method.

Table 20 summarizes the accuracy of all ablations under GPT-4 and DeepSeek-V3 across the three benchmarks. The results show that: (i) symbolic and natural language reasoning are complementary—removing either side or their interaction harms accuracy; and (ii) within both pure-NL and full SL+NL pipelines, our adaptive sparse gate outperforms static sparse topologies such as SparseMAD and CortexDebate.

## J  ABLATION ON THE CONFIDENCE TERM IN THE SPARSE GATE

Our design of confidence scores follows a variety of multi-agent works, where LLMs generate an explicit confidence score that is then used for confidence-weighted voting or debate control. Instead of using absolute probabilities, our sparse gate uses self-reported confidence only as a relative ranking signal, via ratios such as $C_i/C_j$.

To test the necessity and robustness of the self-reported confidence scores in our sparse communication gate, we perform an ablation where we disable the confidence term in sparse-gating. Table 21 compares accuracy with and without the confidence term on Qwen2.5-7B-Instruct and GPT-4 over three datasets (ProofWriter, FOLIO, Chinese LogiQA-V2). Using confidence is consistently better, suggesting that self-reported confidence always provides a useful additional signal for sparse gating.

Table 21: Accuracy (%) with and without the confidence term in the sparse gate.

| Model | Variant | ProofWriter | FOLIO | Chinese LogiQA-V2 |
|---|---|---|---|---|
| Qwen2.5-7B-Instruct | w/o conf | 75.33% | 64.23% | 67.92% |
| Qwen2.5-7B-Instruct | w/ conf | 76.50% | 65.68% | 68.11% |
| GPT-4 | w/o conf | 90.87% | 84.80% | 74.26% |
| GPT-4 | w/ conf | 92.00% | 86.27% | 74.76% |

# K  ADDITIONAL EXPERIMENTAL RESULTS ON DEEPSEEK-V3

This section presents additional experimental results for DeepSeek-V3 that show similar patterns to the GPT-4 results discussed in the main paper.

## K.1  COMMUNICATION THRESHOLD ANALYSIS

Figure 5 shows the effect of communication gating threshold on accuracy and token saving rate for DeepSeek-V3. The results demonstrate patterns consistent with GPT-4, achieving token reduction while maintaining high accuracy.

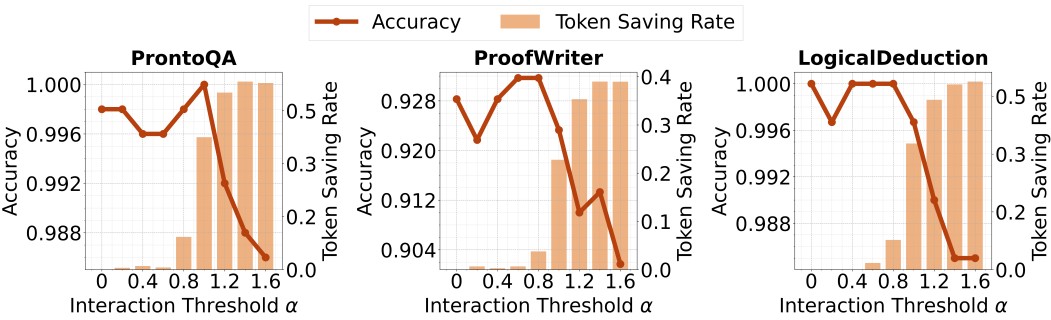

Figure 5: Effect of communication gating threshold on accuracy and token saving rate on DeepSeek-V3.

## K.2  TRANSLATION QUALITY ANALYSIS

Figure 6 shows the relationship between debate rounds and solver execution rates for DeepSeek-V3. Consistent with our GPT-4 findings, the execution rate increases during the first 1-2 rounds and then shows diminishing returns.

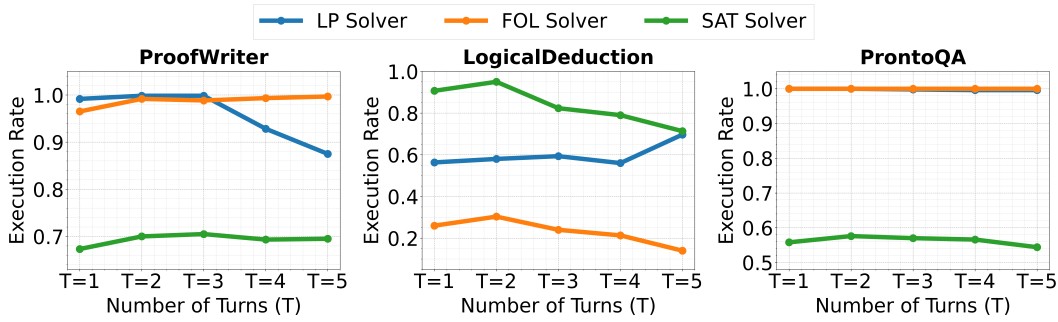

Figure 6: Relation between debate rounds and solver execution rate for DeepSeek-V3.

# L  ADDITIONAL EXPERIMENTAL RESULTS ON CLAUDE 3.7 SONNET

This section provides supplementary experimental results for Claude 3.7 Sonnet.

### L.1 COMMUNICATION THRESHOLD ANALYSIS

Figure 7 presents the accuracy-efficiency trade-off for Claude 3.7 Sonnet. Similar to GPT-4 and DeepSeek-V3, Claude 3.7 maintains high accuracy while achieving significant token savings through sparse communication.

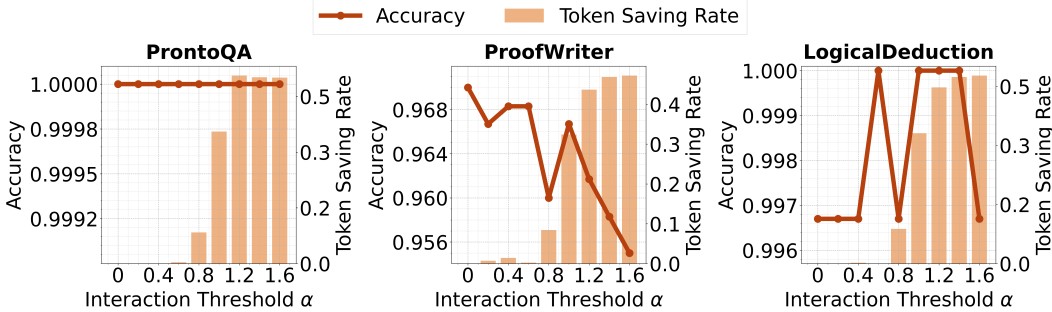

Figure 7: Effect of communication gating threshold on accuracy and token saving rate on Claude 3.7 Sonnet.

### L.2 TRANSLATION QUALITY ANALYSIS

Figure 8 illustrates the translation quality dynamics for Claude 3.7 Sonnet. The pattern is consistent with other models: execution rates improve significantly within the first 2-3 debate rounds, validating our multi-agent debate approach for translation refinement.

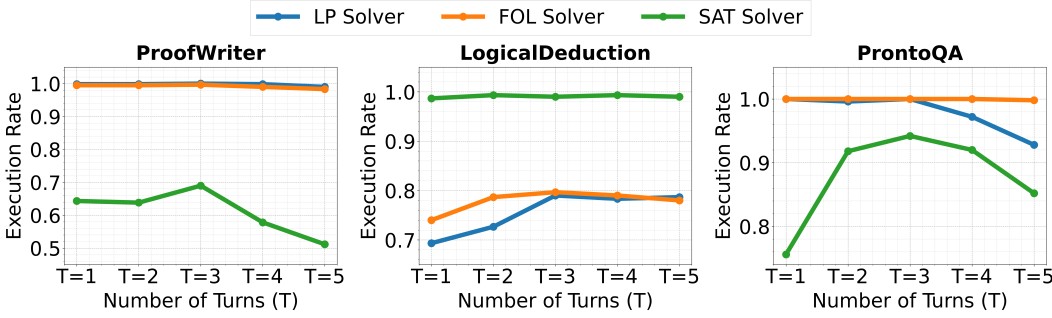

Figure 8: Relation between debate rounds and solver execution rate for Claude 3.7 Sonnet.

## M  EMPIRICAL QUANTIFICATION OF SHARED MISTAKES AMONG AGENTS

To measure the probability of agents sharing similar mistakes in our system, we introduce the **Common Error Rate** $\mathrm{CER}_n$. For any subset $S$ of $n$ agents, let the answer given by agent $a$ be $y_{a,q}$, the indicator of correctness of answer $y_{a,q}$ be $c_{a,q} \in \{0,1\}$,

$$\mathrm{CER}_S = \frac{1}{|Q|} \big| \{q \in Q : \ \forall a \in S, \ c_{a,q} = 0 \text{ and } y_{a,q} \text{ are identical}\} \big|.$$

That is, it measures the fraction of questions on which all agents in subset $S$ pick the same wrong option. We then average over all $\binom{5}{n}$ subsets to obtain $\mathrm{CER}_n$.

As show in Table 22:

- $\mathrm{CER}_n$ drops rapidly as $n$ increases.
- Our sparse gating further largely reduces these correlated mistakes.

This shows that "many agents choosing the same wrong label" is already rare, and our sparse gating mechanism further reduces such shared mistakes.

Table 22: Common Error Rate ($\text{CER}_n$) across agent subsets (lower is better).

| #Agents | Variant | GPT-4 | | | DeepSeek-V3 | | |
|---|---|---|---|---|---|---|---|
| | | ProofWriter | ProntoQA | LogicalDeduction | ProofWriter | ProntoQA | LogicalDeduction |
| 1 | w/o sparse | 0.1017 | 0.0024 | 0.0700 | 0.2593 | 0.0016 | 0.0060 |
| | w/ sparse | 0.1060 | 0.0016 | 0.0855 | 0.2475 | 0.0008 | 0.0047 |
| 2 | w/o sparse | 0.0900 | 0.0020 | 0.0555 | 0.0955 | 0.0002 | 0.0033 |
| | w/ sparse | 0.0503 | 0.0002 | 0.0446 | 0.0643 | 0.0000 | 0.0003 |
| 3 | w/o sparse | 0.0653 | 0.0020 | 0.0426 | 0.0537 | 0.0000 | 0.0033 |
| | w/ sparse | 0.0183 | 0.0000 | 0.0212 | 0.0207 | 0.0000 | 0.0000 |
| 4 | w/o sparse | 0.0425 | 0.0020 | 0.0402 | 0.0367 | 0.0000 | 0.0033 |
| | w/ sparse | 0.0117 | 0.0000 | 0.0101 | 0.0125 | 0.0000 | 0.0000 |
| 5 | w/o sparse | 0.0200 | 0.0020 | 0.0383 | 0.0283 | 0.0000 | 0.0033 |
| | w/ sparse | **0.0067** | **0.0000** | **0.0097** | **0.0095** | **0.0000** | **0.0000** |

# N  MULTI-TURN INTERACTION ALGORITHM FOR SPARSE COMMUNICATION

## N.1  MULTI-TURN DYNAMIC INTERACTION PREFERENCE BETWEEN LLMS

We establish a sparse communication topology to improve the efficiency in multi-turn interactions through a dynamic pruning mechanism, which allows source agent $i$ to communicate its output to the receiving agent $j$ at round $d$. Specifically, we propose a preference score quantifying the potential utility of the information in the communication, which is defined as:

$$\text{Pre}_{i \to j}^{d} = \frac{C_i^d}{C_j^d} + \lambda(1 - \cos(A_j^d, A_i^d || A_j^d)).$$

This score comprises two key components. The first is $C_i^d/C_j^d$, representing the ratio of confidence scores between the source agent $i$ and the receiving agent $j$ at round $d$. The second is $1 - \cos(A_j^d, A_i^d)$, measuring the difference between the two outputs, regarded as information gain.

To guarantee efficiency, we propose a dynamic strategy to determine with which agent to communicate. Specifically, in round $d$, we use this average preference score $\overline{\text{Pre}_{i \to j}^{d-1}}$ as the adaptive threshold. We define a binary communication gate $O_{i \to j}^d$. Communication from $i$ to $j$ is permitted only if the current preference score is greater than or equal to the historical average, indicating that the current interaction is at least as beneficial as the average past interaction between this pair. The indicator of whether agent $i$ benefits agent $j$ at round $d$ is formally defined as:

$$O_{i \to j}^{d} = \begin{cases} 1, & \text{Pre}_{i \to j}^d \geq \alpha \cdot \overline{\text{Pre}_{i \to j}^{d-1}} \\ 0, & \text{Pre}_{i \to j}^d < \alpha \cdot \overline{\text{Pre}_{i \to j}^{d-1}} \end{cases}.$$

## N.2  MULTI-TURN INTERACTION ALGORITHM FOR ENHANCING LLMS' REASONING

The sparse communication mechanism directly informs how each agent updates its internal state or memory across debate rounds. Each agent maintains a personalized memory that aggregates valuable insights from others. At the beginning of the first round ($d = 1$), all agents start with an empty memory $M_s^1 \leftarrow \varnothing$ and communication is fully connected ($O_{i \to j}^d = 1$ for all pairs). From the second round, the sparse communication gate $O_{i \to j}^d$ is activated. At the end of each round $d$, every agent $s$ updates its memory for the next round $M_s^{d+1}$ by selectively incorporating the outputs $A_i^d$ from only those agents $i$ for which the communication channel was open (i.e., $O_{i \to j}^d = 1$). After the memory is updated, agent $s$ generates its output for the next round $A_i^{d+1}$, by querying the symbolic question and $i$'s newly updated, personalized memory. After $D$ rounds of debate, the final outputs from all agents $A_1^{D+1}, \ldots, A_n^{D+1}$, are aggregated via a majority vote to determine the final answer.

# O  SENSITIVITY TO THE SIMILARITY METRIC

Our sparse gate uses a similarity measure between agent rationales to estimate information gain. In the main experiments we use cosine similarity over Sentence-BERT embeddings, but other text similarity metrics are also possible. To test robustness, we compare cosine similarity against ROUGE-L as the similarity metric inside the gate.

---

**Algorithm 1:** Multi-Turn Interaction Algorithm for Enhancing LLMs' Logical Reasoning

---

**Input:** Communication rounds $D$, Agent number $n$, hyper-parameter $\lambda$;

1   Translate raw logical question $Q$ to symbolic expression $\mathrm{Sym}(Q)$;

2   $M_1^{d=1}, \ldots, M_n^{d=1} \leftarrow \varnothing$;

3   **for** $d \in \{1, \ldots, D\}$ **do**

4      $O_{i \rightarrow j}^d = 1$ for all $i, j \in \{1, \ldots, n\}$;

5      Compute $\mathrm{Pre}_{i \rightarrow j}^d = \frac{C_i^d}{C_j^d} + \lambda(1 - \cos(A_j^d, A_i^d))$ for all $i \neq j$;

6      Compute $\overline{\mathrm{Pre}_{i \rightarrow j}^d} = \frac{1}{d}(\overline{\mathrm{Pre}_{i \rightarrow j}^{d-1}} \cdot (d - 1) + \frac{C_i^d}{C_j^d} + \lambda(1 - \cos(A_j^d, A_i^d)))$ for all $i \neq j$;

7      **if** $\mathrm{Pre}_{i \rightarrow j}^d < \alpha \cdot \overline{\mathrm{Pre}_{i \rightarrow j}^{d-1}}$ **then**

8         $O_{i \rightarrow j}^d = 0$;

9      **for** $s \in \{1, \ldots, n\}$ **do**

        `// Memory update of the s-th agent at round d`

10         $M_s^{d+1} \leftarrow M_s^d \cup \{A_i^d \mid i \in \{1, \ldots, n\}, O_{i \rightarrow s}^d = 1\}$;

        `// Output of the s-th agent at round d using personalized`
        `memory`

11         $A_s^{d+1} \leftarrow \mathrm{LLM}_s(\mathrm{Sym}(Q) \| M_s^{d+1})$;

12   Majority vote among the $n$ agents $A_1^{D+1}, \ldots, A_n^{D+1}$;

---

Table 23: Sensitivity to the similarity metric. Accuracies (%) for cosine similarity vs. ROUGE-L.

| Metric | Model | PW Acc | PQA Acc | LD Acc |
|--------|-------|--------|---------|--------|
| Cosine | GPT-4 | 92.00% | 100.00% | 94.33% |
| ROUGE-L | GPT-4 | 91.00% | 97.80% | 93.33% |
| Cosine | DeepSeek-V3 | 93.33% | 100.00% | 100.00% |
| ROUGE-L | DeepSeek-V3 | 92.00% | 98.40% | 98.33% |

Table 23 reports accuracies when using cosine vs. ROUGE-L for GPT-4 and DeepSeek-V3 across the three main benchmarks. The differences are within 1–2 percentage points, indicating that our framework is not sensitive to the specific choice of similarity metric.

## P    PROMPT TEMPLATES

### P.1    TRANSLATION DEBATE

---

**Translation Prompt**

**Task**. You are given a logic problem in natural language including a context and a question as follows:
Context: `${context}`
Question: `${question}`

**Discussion Rules**

1. **Syntax Verification**: Carefully review previous discussions to understand others' translations. While maintaining *your own* symbolic language system, check and correct any syntax errors in your translation (e.g., unclosed parentheses, malformed expressions).

2. **Completeness Check**: Review others' translations to understand their interpretation of the natural language problem. While keeping *your own* symbolic language system, verify and correct the information completeness of your translation (no missing/extra facts, rules, predicates, or statements from the original problem).

3. **Language Independence**: When referencing others' translations, you *must* maintain your own symbolic language system. *Do not* adopt symbols or syntax from other languages.

**Discussion history**
`${chat_history}`

---

---

**Role-specific description**

```
${role_description}
```
Now it's your turn to speak. Please speak as concisely and clearly as possible

---

**Role-specific description — LP translator**

Your task is to translate the logic problem in natural language into LP logic formulas:
1. define all the predicates in the problem
2. parse the problem into logic rules based on the defined predicates
3. write all the facts mentioned in the problem
4. parse the question into the logic form (Use `&&` to represent AND, and you cannot use `NOT` or other negations in LP)

**Example**

```
Context: Each jompus is fruity.
(... more context here ...)
Rompuses are zumpuses. Alex is a tumpus.

Question: True or false: Alex is not shy.

Predicates:
Jompus($x, bool) ::: Does x belong to Jompus?
(... more predicates here ...)
Zumpus($x, bool) ::: Does x belong to Zumpuses?

Facts:
Tumpuses(Alex, True)

Rules:
Jompus($x, True) >>> Fruity($x, True)
(... more rules here ...)
Dumpus($x, True) >>> Rompus($x, True)

Query:
Shy(Alex, False)
```

---

**Role-specific description — FOL translator**

Your task is to translate the logic problem in natural language into first-order logic formulas. The grammar of first-order logic is defined as follows:

| | |
|---|---|
| logical conjunction: | $expr_1 \wedge expr_2$ |
| logical disjunction: | $expr_1 \vee expr_2$ |
| logical exclusive disjunction: | $expr_1 \oplus expr_2$ |
| logical negation: | $\neg expr_1$ |
| $expr_1$ implies $expr_2$: | $expr_1 \rightarrow expr_2$ |
| $expr_1$ iff $expr_2$: | $expr_1 \leftrightarrow expr_2$ |
| logical universal quantification: | $\forall x$ |
| logical existential quantification: | $\exists x$ |

Output format: `logic form ::: description`

**Example**

```
Context: All people who regularly drink coffee are
dependent on caffeine.
(... more context here ...)
If Rina is not a person dependent on caffeine and a
student, then Rina is either a person dependent
on caffeine and a student, or a person dependent
on caffeine nor a student, or neither a person
dependent on caffeine nor a student.
```

```
Question: Based on the above information, is the
following statement true, false, or uncertain?
Rina is either a person who jokes about being
addicted to caffeine or is unaware that caffeine
is a drug.

Predicates:
Dependent(x) ::: x is a person dependent on caffeine
(... more predicates here ...)
Student(x) ::: x is a student

Premises:
$\forall x$ (Drinks(x) $\rightarrow$ Dependent(x)) ::: All people who
regularly drink coffee are dependent on caffeine.
(... more premises here ...)
$\forall x$ (Jokes(x) $\rightarrow$ $\neg$Unaware(x)) ::: No one who
    jokes
about being addicted to caffeine is unaware that
caffeine is a drug.

Conclusion:
Jokes(rina)  Unaware(rina) ::: Rina is either a person who jokes about
    being addicted to caffeine
or is unaware that caffeine is a drug.
```

---

### Role-specific description — SAT translator

Your task is to parse the logic problem in natural language as a SAT problem using Z3 syntax, defining declarations, constraints, and options.

1. Always include all three section headers in order: `# Declarations`, `# Constraints`, `# Options`
2. Declarations must follow exact patterns:
   - `name = EnumSort([items, ...])` for non-numeric items
   - `name = IntSort([numbers, ...])` for numeric items
   - `name = Function([types] -> [return_type])`
3. Constraints support:
   - Direct expressions with `==, !=, <=, >=, <, >, Implies(), And(), Or(), Not()`
   - `ForAll([var:type, ...], expr)` and `Exists([var:type, ...], expr)`
   - `Count([var:type], condition)`
   - `Distinct([var:type], expr)`
4. Options must use predefined functions:
   - `is_valid(), is_sat(), is_unsat()`
5. Add explanation with `:::`
6. Avoid:
   - Add # in any other places apart from three section headers
   - Add any other unnecessary comment or dashes

### Example

```
Context: Bob is cold. Bob is quiet. Bob is red. Bob is smart. Charlie
    is kind. Charlie is quiet. Charlie is red. Charlie is rough. Dave
    is cold. Dave is kind. Dave is smart. Fiona is quiet. If something
     is quiet and cold then it is smart. Red, cold things are round.
    If something is kind and rough then it is red. All quiet things
    are rough. Cold, smart things are red. If something is rough then
    it is cold. All red things are rough. If Dave is smart and Dave is
     kind then Dave is quiet.
Question: True or false: Charlie is kind.
```

```
# Declarations
objects = EnumSort([Bob, Charlie, Dave, Fiona])
attributes = EnumSort([cold, quiet, red, smart, kind, rough, round])
has_attribute = Function([objects, attributes] -> [bool])

# Constraints
has_attribute(Bob, cold) == True ::: Bob is cold.
has_attribute(Bob, quiet) == True ::: Bob is quiet.
has_attribute(Bob, red) == True ::: Bob is red.
has_attribute(Bob, smart) == True ::: Bob is smart.
has_attribute(Charlie, kind) == True ::: Charlie is kind.
has_attribute(Charlie, quiet) == True ::: Charlie is quiet.
has_attribute(Charlie, red) == True ::: Charlie is red.
has_attribute(Charlie, rough) == True ::: Charlie is rough.
has_attribute(Dave, cold) == True ::: Dave is cold.
has_attribute(Dave, kind) == True ::: Dave is kind.
has_attribute(Dave, smart) == True ::: Dave is smart.
has_attribute(Fiona, quiet) == True ::: Fiona is quiet.
ForAll([x:objects], Implies(And(has_attribute(x, quiet) == True,
    has_attribute(x, cold) == True), has_attribute(x, smart) == True))
     ::: If something is quiet and cold then it is smart.
ForAll([x:objects], Implies(And(has_attribute(x, red) == True,
    has_attribute(x, cold) == True), has_attribute(x, round) == True))
     ::: Red, cold things are round.
ForAll([x:objects], Implies(And(has_attribute(x, kind) == True,
    has_attribute(x, rough) == True), has_attribute(x, red) == True))
     ::: If something is kind and rough then it is red.
ForAll([x:objects], Implies(has_attribute(x, quiet) == True,
    has_attribute(x, rough) == True)) ::: All quiet things are rough.
ForAll([x:objects], Implies(And(has_attribute(x, cold) == True,
    has_attribute(x, smart) == True), has_attribute(x, red) == True))
     ::: Cold, smart things are red.
ForAll([x:objects], Implies(has_attribute(x, rough) == True,
    has_attribute(x, cold) == True)) ::: If something is rough then it
     is cold.
ForAll([x:objects], Implies(has_attribute(x, red) == True,
    has_attribute(x, rough) == True)) ::: All red things are rough.
Implies(And(has_attribute(Dave, smart) == True, has_attribute(Dave,
    kind) == True), has_attribute(Dave, quiet) == True) ::: If Dave is
     smart and Dave is kind then Dave is quiet.

# Options
is_valid(has_attribute(Charlie, kind) == True) ::: Charlie is kind is
    True (A).
is_unsat(has_attribute(Charlie, kind) == True) ::: Charlie is kind is
    False (B).
```

## P.2 REASONING DEBATE

---

**Final Debate Prompt**

You are given a logic problem that contains a context, a question, and options:
Context: ${context}
Question: ${question}
Options: ${options}
Role description: ${Role-specific description}

**Your initial answer is** ${predict}.
**Your initial reasoning is:** ${reasoning}.

You are now in a collaborative debate with other reasoning agents. Your goal is to reach the correct answer through discussion.

**Important Debate Rules**

---

1. Review other agents' arguments in the discussion history first.
2. Identify specific points of agreement or disagreement.
3. Challenge weak reasoning with concrete counterexamples.
4. No need to repeat your whole reasoning if your argument remains unchanged.
5. Acknowledge other arguments when you find them correct, even if they contradict your initial position.
6. If you change your answer, **always** explain why you changed.
7. Be willing to change your answer if convinced by other arguments.
8. Reference specific agents and their arguments when responding.
9. Be interactive and engaging with other agents!

**Discussion history**
${chat_history}

**Turn-specific instruction**
${turn_specific_instruction}

---

**Role-specific description — LP supporter**

You are a supporter of the Logic Programming (LP) approach. **Strengths:**
- Systematic rule-based reasoning with clear steps
- Handle complex relations via predicates and rules
- Transparent reasoning process verifiable step by step
- Strong foundation in formal logic and theorem proving

**In the debate, you should:**
- Emphasize rigor and reliability of LP reasoning
- Highlight systematic application of logical rules
- Defend transparency and verifiability
- Challenge others when lacking formal logical foundation

---

**Role-specific description — FOL supporter**

You are a supporter of First-Order Logic (FOL). **Strengths:**
- Mathematical precision with quantifiers and operators
- Express complex relationships precisely
- Sound theoretical foundation
- Handle nested structures and implications

**In the debate, you should:**
- Emphasize rigor and expressiveness of FOL
- Highlight formal completeness and soundness
- Defend against criticisms of complexity
- Challenge others when lacking precision

---

**Role-specific description — SAT supporter**

You are a supporter of the SAT/SMT (Z3) approach. **Strengths:**
- Formal representation using Z3 syntax
- Complete and sound reasoning
- Ability to prove validity/satisfiability
- Handle quantifiers and complex formulas

**In the debate, you should:**
- Emphasize formal correctness and completeness
- Highlight Z3's power in constraint solving
- Defend clarity of declarative specification
- Challenge others on missed edge cases

---

**Role-specific description — Plan-and-Solve supporter**

You are a supporter of the Plan-and-Solve approach. **Strengths:**
- First devise a plan, then execute it
- Structured reasoning framework

**In the debate, you should:**
• Defend the reasoning process and solution steps
• Justify the plan structure and accuracy

---

**Role-specific description — COT Solver supporter**

You are the Chain-of-Thought (CoT) supporter. **Strengths:**
• Step-by-step reasoning from facts to conclusions
**In the debate, you should:**
• Defend solutions generated via CoT prompting
• Validate each inference in the reasoning chain
• Argue for soundness of the conclusion

## Q CASE STUDIES

### Q.1 REASONINGS EXTRACTED FROM SYMBOLIC SOLVERS

### Case Study of Reasonings Extracted

**Context.** The cat eats the squirrel. The cat sees the squirrel. The cow eats the squirrel. The cow sees the cat. The rabbit is round. The rabbit sees the cat. The squirrel eats the rabbit. The squirrel is cold. The squirrel needs the rabbit. The squirrel sees the cat. If someone sees the cat and they are not green then they see the cow. If the rabbit is kind and the rabbit sees the squirrel then the squirrel needs the rabbit. Rough people are cold. If someone sees the rabbit then they are not round. If someone sees the squirrel and they are not green then they need the squirrel. If someone eats the cow then they see the rabbit. If someone eats the squirrel then they are rough. If someone is cold then they eat the cow.

**Question.** Based on the above information, is the following statement true, false, or unknown? *The cat is not round.*

**Options.** A) True    B) False    C) Unknown

---

**LP Reasoning Extracted from Pyke Solver:**

```
We first define following predicates and corresponding natural language explanations:
  Eats($x, $y, bool) ::: Does x eat y?
  Sees($x, $y, bool) ::: Does x see y?
  Round($x, bool) ::: Is x round?
  Cold($x, bool) ::: Is x cold?
  Needs($x, $y, bool) ::: Does x need y?
  Green($x, bool) ::: Is x green?
  Kind($x, bool) ::: Is x kind?
  Rough($x, bool) ::: Is x rough?
We have following known facts from the context:
  Eats(cat, squirrel, True)
  Sees(cat, squirrel, True)
  Eats(cow, squirrel, True)
  Sees(cow, cat, True)
  Round(rabbit, True)
  Sees(rabbit, cat, True)
  Eats(squirrel, rabbit, True)
  Cold(squirrel, True)
  Needs(squirrel, rabbit, True)
  Sees(squirrel, cat, True)
We have following known rules from the context:
  rule1: Sees($x, cat, True) && Green($x, False) >>> Sees($x, cow, True)
  rule2: Kind(rabbit, True) && Sees(rabbit, squirrel, True) >>> Needs(squirrel, rabbit,
      True)
  rule3: Rough($x, True) >>> Cold($x, True)
  rule4: Sees($x, rabbit, True) >>> Round($x, False)
  rule5: Sees($x, squirrel, True) && Green($x, False) >>> Needs($x, squirrel, True)
  rule6: Eats($x, cow, True) >>> Sees($x, rabbit, True)
  rule7: Eats($x, squirrel, True) >>> Rough($x, True)
  rule8: Cold($x, True) >>> Eats($x, cow, True)
Now begin reasoning to obtain all implied facts:
Use rule1: Sees($x, cat, True) && Green($x, False) >>> Sees($x, cow, True)
Bind $x to 'cow'
Unbind $x
```

```
Bind $x to 'rabbit'
Unbind $x
Bind $x to 'squirrel'
Unbind $x
Finish implied with rule1
Use rule2: Kind(rabbit, True) && Sees(rabbit, squirrel, True) >>> Needs(squirrel,
     rabbit, True)
Finish implied with rule2
Use rule3: Rough($x, True) >>> Cold($x, True)
Finish implied with rule3
Use rule4: Sees($x, rabbit, True) >>> Round($x, False)
Finish implied with rule4
Use rule5: Sees($x, squirrel, True) && Green($x, False) >>> Needs($x, squirrel, True)
Bind $x to 'cat'
Unbind $x
Finish implied with rule5
Use rule6: Eats($x, cow, True) >>> Sees($x, rabbit, True)
Finish implied with rule6
Use rule7: Eats($x, squirrel, True) >>> Rough($x, True)
Bind $x to 'cat'
Obtain a new implied fact: Rough('cat', True)
Reuse rule3: Rough($x, True) >>> Cold($x, True)
Bind $x to 'cat'
Obtain a new implied fact: Cold('cat', True)
Use rule8: Cold($x, True) >>> Eats($x, cow, True)
Finish implied with rule8
Finish implied with rule3
Reuse rule3: Rough($x, True) >>> Cold($x, True)
Bind $x to 'cow'
Obtain a new implied fact: Cold('cow', True)
Reuse rule8: Cold($x, True) >>> Eats($x, cow, True)
Finish implied with rule8
Finish implied with rule3
Finish implied with rule7
Reuse rule8: Cold($x, True) >>> Eats($x, cow, True)
Bind $x to 'squirrel'
Obtain a new implied fact: Eats('squirrel', 'cow', True)
Reuse rule6: Eats($x, cow, True) >>> Sees($x, rabbit, True)
Bind $x to 'squirrel'
Obtain a new implied fact: Sees('squirrel', 'rabbit', True)
Reuse rule1: Sees($x, cat, True) && Green($x, False) >>> Sees($x, cow, True)
Bind $x to 'squirrel'
Finish implied with rule1
Reuse rule2: Kind(rabbit, True) && Sees(rabbit, squirrel, True) >>> Needs(squirrel,
     rabbit, True)
Finish implied with rule2
Reuse rule4: Sees($x, rabbit, True) >>> Round($x, False)
Bind $x to 'squirrel'
Obtain a new implied fact: Round('squirrel', False)
Finish implied with rule4
Reuse rule5: Sees($x, squirrel, True) && Green($x, False) >>> Needs($x, squirrel, True)
Bind $x to 'squirrel'
Finish implied with rule5
Finish implied with rule6
Reuse rule7: Eats($x, squirrel, True) >>> Rough($x, True)
Bind $x to 'squirrel'
Finish implied with rule7
Reuse rule6: Eats($x, cow, True) >>> Sees($x, rabbit, True)
Bind $x to 'cat'
Obtain a new implied fact: Sees('cat', 'rabbit', True)
Reuse rule1: Sees($x, cat, True) && Green($x, False) >>> Sees($x, cow, True)
Bind $x to 'cat'
Finish implied with rule1
Reuse rule2: Kind(rabbit, True) && Sees(rabbit, squirrel, True) >>> Needs(squirrel,
     rabbit, True)
Finish implied with rule2
Reuse rule4: Sees($x, rabbit, True) >>> Round($x, False)
Bind $x to 'cat'
Obtain a new implied fact: Round('cat', False)
Finish implied with rule4
Reuse rule5: Sees($x, squirrel, True) && Green($x, False) >>> Needs($x, squirrel, True)
Bind $x to 'cat'
Finish implied with rule5
Finish implied with rule6
Reuse rule7: Eats($x, squirrel, True) >>> Rough($x, True)
Bind $x to 'cat'
Finish implied with rule7
Reuse rule6: Eats($x, cow, True) >>> Sees($x, rabbit', True)
```

```
Bind $x to 'cow'
Obtain a new implied fact: Sees('cow', 'rabbit', True)
Reuse rule1: Sees($x, cat, True) && Green($x, False) >>> Sees($x, cow, True)
Bind $x to 'cow'
Finish implied with rule1
Reuse rule2: Kind(rabbit, True) && Sees(rabbit, squirrel, True) >>> Needs(squirrel,
    rabbit, True)
Finish implied with rule2
Reuse rule4: Sees($x, rabbit, True) >>> Round($x, False)
Bind $x to 'cow'
Obtain a new implied fact: Round('cow', False)
Finish implied with rule4
Reuse rule5: Sees($x, squirrel, True) && Green($x, False) >>> Needs($x, squirrel, True)
Bind $x to 'cow'
Finish implied with rule5
Finish implied with rule6
Reuse rule7: Eats($x, squirrel, True) >>> Rough($x, True)
Bind $x to 'cow'
Finish implied with rule7
Finish implied with rule8
All newly implied Facts: Cold('cat', True), Cold('cow', True), Eats('squirrel', 'cow',
    True), Rough('cat', True), Round('cat', False), Round('cow', False), Round('
    squirrel', False), Sees('cat', 'rabbit', True), Sees('cow', 'rabbit', True), Sees
    ('squirrel', 'rabbit', True)
Finish reasoning
```

**FOL Reasoning Extracted from Prover9 Solver:**

```
prove original conclusion:
3 (all x (Rough(x) -> Cold(x))).  [assumption].
4 (all x (Sees(x,Rabbit) -> -Round(x))).  [assumption].
6 (all x (Eats(x,Cow) -> Sees(x,Rabbit))).  [assumption].
7 (all x (Eats(x,Squirrel) -> Rough(x))).  [assumption].
8 (all x (Cold(x) -> Eats(x,Cow))).  [assumption].
9 -Round(Cat).  [goal].
10 -Eats(x,Cow) | Sees(x,Rabbit).  [clausify(6)].
11 Eats(Cat,Squirrel).  [assumption].
14 -Eats(x,Squirrel) | Rough(x).  [clausify(7)].
15 -Cold(x) | Eats(x,Cow).  [clausify(8)].
16 -Sees(x,Rabbit) | -Round(x).  [clausify(4)].
18 Round(Cat).  [deny(9)].
19 -Cold(x) | Sees(x,Rabbit).  [resolve(15,b,10,a)].
21 -Rough(x) | Cold(x).  [clausify(3)].
27 -Sees(Cat,Rabbit).  [resolve(18,a,16,b)].
29 Sees(x,Rabbit) | -Rough(x).  [resolve(19,a,21,b)].
31 Rough(Cat).  [resolve(14,a,11,a)].
33 -Rough(Cat).  [resolve(29,a,27,a)].
34 $F.  [resolve(33,a,31,a)].
```

## Q.2 TRANSLATE DEBATE

We present a case study from LogicalDeduction where agents collaboratively translate a fruit pricing problem into symbolic languages. Table 24 shows how through three debate rounds, agents identify and correct critical errors: SAT discovers its inverted ranking (treating 1 as most expensive rather than cheapest), LP clarifies ranking ambiguity, and FOL adds missing ordering axioms. This peer review process yields syntactically and semantically accurate translations across all three symbolic languages, demonstrating how multi-agent debate enhances translation quality.

---

### Case Study from logical_deduction

**Context:** A fruit stand sells five fruits: mangoes, kiwis, plums, pears, and watermelons. The kiwis are less expensive than the plums. The pears are the third-most expensive. The kiwis are the second-cheapest. The watermelons are the most expensive.
**Question:** Which of the following is true?
A) The mangoes are the third-most expensive.
B) The kiwis are the third-most expensive.
C) The plums are the third-most expensive.
D) The pears are the third-most expensive.
E) The watermelons are the third-most expensive.

---

Table 24: Case Study of Translation Debate: Agents collaboratively refine their NL-to-SL translations through debate, ✓ indicates correct translation, ✗ indicates incorrect translation.

| Round | Agent Translation | Key Points during Debate |
|---|---|---|
| 1 | **LP ✗:** Predicates: `Fruit($x)`, `MoreExpensive($x,$y)`, `LessExpensive($x,$y)`, `ThirdMostExpensive($x)`… Facts: `LessExpensive(kiwi,plum,True)`, `ThirdMostExpensive(pear,True)` | Initial translation with multiple predicates for comparison. Uses separate predicates for each ranking position. |
| | **FOL ✗:** `Rank(fruit,pos)` where pos ∈ {one,two,three,four,five} `Cheaper(x,y)`. Premises: `Rank(watermelon,one)`, `Rank(pears,three)`, `Rank(kiwis,four)`… | Uses positional ranking with `one`=most expensive. Encodes ordering relationships between fruits. |
| | **SAT ✗:** `fruits=EnumSort([...])` `price_rank=IntSort([1,2,3,4,5])` `pos=Function([fruits]->[price_rank])`… Constraints: `pos(Kiwis)==2…,pos(Pears)==3,pos(Watermelons)==5` | Models prices as integer positions 1-5. Uses constraints like `pos(Kiwis)<pos(Plums)`. |
| 2 | **LP ✗:** Modified to `Rank($x,$n,bool)` where $n: 5=most expensive, 1=least expensive Added rules for deriving complete ordering | **"I need to be more precise about the ranking system." Realizes ambiguity in ranking direction needs clarification.** Adds missing derivation rules for complete ordering. |
| | **FOL ✗:** Same predicates, but added ordering axioms: $\forall X \forall Y$ `(Rank(X,one) ∧ Rank(Y,two) → Cheaper(Y,X))` Plus completeness: each fruit must have some rank | **"I need to add the ordering relationships between ranks." Adds 10 ordering axioms to fully specify rank relations.** |
| | **SAT ✗:** Same structure but notes critical error Realizes rank 1 should be cheapest, not most expensive | **"I made a critical error in my ranking system." Identifies that ranking direction was inverted.** |
| 3 | **LP ✓:** Final version with both `LessExpensive` and `MoreExpensive` Complete rules for rank-based comparisons `Rank($x,$n,True) && Rank($y,$m,True) && $n<$m »> LessExpensive(...)` | Maintains own symbolic system while incorporating insights. Final translation is syntactically correct and complete. |
| | **FOL ✓:** Complete with all 10 ordering axioms Asymmetry constraint: $\forall X \forall Y$ `(Cheaper(X,Y) → ¬Cheaper(Y,X))` Each fruit and rank uniqueness constraints maintained | Final version includes all necessary constraints. Ensures logical consistency of ordering relations. |
| | **SAT ✓:** Corrected ranking: 1=cheapest, 5=most expensive `Distinct([f:fruits], pos(f))` for unique ranks All constraints properly oriented: `pos(Kiwis)<pos(Plums)` | **Successfully corrected the ranking direction.** Final translation aligns with problem semantics. Maintains Z3 syntax requirements. |

**Answer:** D

**Round 1**

*LP translator:*
I'll translate this logic problem about fruit prices into LP format:
**Predicates:**
Fruit($x, bool) ::: $x is one of the five fruits
MoreExpensive($x, $y, bool) ::: Fruit $x is more expensive than fruit $y
LessExpensive($x, $y, bool) ::: Fruit $x is less expensive than fruit $y
MostExpensive($x, bool) ::: Fruit $x is the most expensive
SecondMostExpensive($x, bool) ::: Fruit $x is the second-most expensive
ThirdMostExpensive($x, bool) ::: Fruit $x is the third-most expensive
FourthMostExpensive($x, bool) ::: Fruit $x is the fourth-most expensive (second-cheapest)
LeastExpensive($x, bool) ::: Fruit $x is the least expensive (cheapest)
**Facts:**
Fruit(mango, True)
Fruit(kiwi, True)
Fruit(plum, True)
Fruit(pear, True)
Fruit(watermelon, True)
LessExpensive(kiwi, plum, True) ::: Kiwis are less expensive than plums
ThirdMostExpensive(pear, True) ::: Pears are the third-most expensive
FourthMostExpensive(kiwi, True) ::: Kiwis are the second-cheapest (fourth-most expensive)
MostExpensive(watermelon, True) ::: Watermelons are the most expensive
**Rules:**
LessExpensive($x, $y, True) »> MoreExpensive($y, $x, True)
MoreExpensive($x, $y, True) »> LessExpensive($y, $x, True)
**Query:**
ThirdMostExpensive(mango, True) ::: Option A
ThirdMostExpensive(kiwi, True) ::: Option B
ThirdMostExpensive(plum, True) ::: Option C

ThirdMostExpensive(pear, True) ::: Option D
ThirdMostExpensive(watermelon, True) ::: Option E

*FOL translator:*
I'll translate this logical deduction problem into first-order logic.
**Predicates:**
Rank(fruit, pos) ::: fruit has price position pos, where pos $\in \{one, two, three, four, five\}$;
one = most expensive, five = cheapest
Cheaper(x, y) ::: x is cheaper than y
**Premises:**
Rank(watermelon, one) ::: Watermelons are the most expensive
Rank(pears, three) ::: Pears are the third-most expensive
Rank(kiwis, four) ::: Kiwis are the second-cheapest (fourth-most expensive)
Cheaper(kiwis, plums) ::: Kiwis are less expensive than plums
$\forall F \forall P \forall Q\big((Rank(F, P) \land Rank(F, Q)) \to (P = Q)\big)$ ::: Each fruit has exactly one rank
$\forall P \forall F \forall G\big((Rank(F, P) \land Rank(G, P)) \to (F = G)\big)$ ::: Each rank has exactly one fruit
**Conclusion:**
Rank(mangoes, three) ::: Option A
Rank(kiwis, three) ::: Option B
Rank(plums, three) ::: Option C
Rank(pears, three) ::: Option D
Rank(watermelon, three) ::: Option E

*SAT translator:*
I'll formulate this problem by modeling the fruit prices as positions in an ordering.
**# Declarations**
fruits = EnumSort([Mangoes, Kiwis, Plums, Pears, Watermelons])
price_rank = IntSort([1, 2, 3, 4, 5])
pos = Function([fruits] -> [price_rank])
**# Constraints**
Distinct([f:fruits], pos(f)) ::: Each fruit has a unique price rank (1=cheapest, 5=most expensive)
pos(Kiwis) < pos(Plums) ::: The kiwis are less expensive than the plums
pos(Pears) == 3 ::: The pears are the third-most expensive
pos(Kiwis) == 2 ::: The kiwis are the second-cheapest (rank 2)
pos(Watermelons) == 5 ::: The watermelons are the most expensive (rank 5)
**# Options**
is_valid(pos(Mangoes) == 3) ::: A) The mangoes are the third-most expensive
is_valid(pos(Kiwis) == 3) ::: B) The kiwis are the third-most expensive
is_valid(pos(Plums) == 3) ::: C) The plums are the third-most expensive
is_valid(pos(Pears) == 3) ::: D) The pears are the third-most expensive
is_valid(pos(Watermelons) == 3) ::: E) The watermelons are the third-most expensive

**Round 2**
*LP translator:*

Looking at the other translations, I notice some inconsistencies in my approach that need correction:
1. Price ranking clarification: I need to be more precise about the ranking system. In my translation, I should clarify that rank 1 is least expensive and rank 5 is most expensive (or vice versa) to avoid ambiguity.
2. Missing derivation rules: I should add rules that allow us to derive the complete ordering of fruits based on the given information.
Let me correct my translation:
**Predicates:**
Fruit($x, bool) ::: $x is one of the five fruits
Rank($x, $n, bool) ::: Fruit $x has price rank $n (5=most expensive, 1=least expensive)
LessExpensive($x, $y, bool) ::: Fruit $x is less expensive than fruit $y
**Facts:**

Fruit(mango, True)
Fruit(kiwi, True)
Fruit(plum, True)
Fruit(pear, True)
Fruit(watermelon, True)
LessExpensive(kiwi, plum, True) ::: Kiwis are less expensive than plums
Rank(pear, 3, True) ::: Pears are the third-most expensive
Rank(kiwi, 2, True) ::: Kiwis are the second-cheapest (rank 2)
Rank(watermelon, 5, True) ::: Watermelons are the most expensive (rank 5)
**Rules:**
LessExpensive($x, $y, True) && Rank($y, $n, True) »> Rank($x, $m, True) && $m < $n
::: If $x is less expensive than $y, then $x's rank is lower than $y's rank
Rank($x, $n, True) && Rank($y, $m, True) && $n < $m »> LessExpensive($x, $y, True)
::: If $x's rank is lower than $y's, then $x is less expensive than $y
**Query:**
Rank(mango, 3, True) ::: Option A
Rank(kiwi, 3, True) ::: Option B
Rank(plum, 3, True) ::: Option C
Rank(pear, 3, True) ::: Option D
Rank(watermelon, 3, True) ::: Option E

*FOL translator*
I'll translate this problem into first-order logic, focusing on the ordering of fruits by price.
**Predicates:**
Rank(fruit, pos) ::: fruit has price position pos, where pos $\in \{one, two, three, four, five\}$;
one = most expensive, five = cheapest
Cheaper(x, y) ::: x is cheaper (less expensive) than y
**Premises:**
Rank(watermelon, one) ::: Watermelons are the most expensive
Rank(pears, three) ::: Pears are the third-most expensive
Rank(kiwis, four) ::: Kiwis are the second-cheapest
Cheaper(kiwis, plums) ::: Kiwis are cheaper than plums
$\forall F \forall P \forall Q \big((Rank(F, P) \land Rank(F, Q)) \rightarrow (P = Q)\big)$ ::: One rank per fruit
$\forall P \forall F \forall G \big((Rank(F, P) \land Rank(G, P)) \rightarrow (F = G)\big)$ ::: One fruit per rank
Rank(mangoes, one) $\lor$ Rank(mangoes, two) $\lor$ Rank(mangoes, three) $\lor$ Rank(mangoes, four) $\lor$ Rank(mangoes, five) ::: Mangoes must have some rank
Rank(plums, one) $\lor$ Rank(plums, two) $\lor$ Rank(plums, three) $\lor$ Rank(plums, four) $\lor$ Rank(plums, five) ::: Plums must have some rank
I need to add the ordering relationships between ranks:
$\forall X \forall Y (Rank(X, one) \land Rank(Y, two) \rightarrow Cheaper(Y, X))$ ::: "higher rank → more expensive"
$\forall X \forall Y (Rank(X, two) \land Rank(Y, three) \rightarrow Cheaper(Y, X))$
$\forall X \forall Y (Rank(X, three) \land Rank(Y, four) \rightarrow Cheaper(Y, X))$
$\forall X \forall Y (Rank(X, four) \land Rank(Y, five) \rightarrow Cheaper(Y, X))$
$\forall X \forall Y (Cheaper(X, Y) \rightarrow \neg Cheaper(Y, X))$ ::: "cheaper" is asymmetric
**Conclusion:**
Rank(mangoes, three) ::: Option A
Rank(kiwis, three) ::: Option B
Rank(plums, three) ::: Option C
Rank(pears, three) ::: Option D
Rank(watermelon, three) ::: Option E

*SAT translator*
I need to correct my SAT translation. Looking at the problem more carefully:
**# Declarations**
fruits = EnumSort([Mangoes, Kiwis, Plums, Pears, Watermelons])
price_rank = IntSort([1, 2, 3, 4, 5])
pos = Function([fruits] -> [price_rank])

# Constraints
Distinct([f:fruits], pos(f)) ::: Each fruit has a unique price rank
pos(Kiwis) < pos(Plums) ::: The kiwis are less expensive than the plums
pos(Pears) == 3 ::: The pears are the third-most expensive
pos(Kiwis) == 2 ::: The kiwis are the second-cheapest
pos(Watermelons) == 5 ::: The watermelons are the most expensive
# Options
is_valid(pos(Mangoes) == 3) ::: A) The mangoes are the third-most expensive
is_valid(pos(Kiwis) == 3) ::: B) The kiwis are the third-most expensive
is_valid(pos(Plums) == 3) ::: C) The plums are the third-most expensive
is_valid(pos(Pears) == 3) ::: D) The pears are the third-most expensive
is_valid(pos(Watermelons) == 3) ::: E) The watermelons are the third-most expensive
I made a critical error in my ranking system. In the problem, rank 1 should be cheapest and
rank 5 most expensive, not the other way around. Let me fix this.
**Round 3**

*LP translator:*

**Predicates:**
Fruit($x, bool) ::: $x is one of the five fruits
Rank($x, $n, bool) ::: Fruit $x has price rank $n (5=most expensive, 1=least expensive)
LessExpensive($x, $y, bool) ::: Fruit $x is less expensive than fruit $y
MoreExpensive($x, $y, bool) ::: Fruit $x is more expensive than fruit $y
**Facts:**
Fruit(mango, True)
Fruit(kiwi, True)
Fruit(plum, True)
Fruit(pear, True)
Fruit(watermelon, True)
LessExpensive(kiwi, plum, True) ::: Kiwis are less expensive than plums
Rank(pear, 3, True) ::: Pears are the third-most expensive
Rank(kiwi, 2, True) ::: Kiwis are the second-cheapest (rank 2)
Rank(watermelon, 5, True) ::: Watermelons are the most expensive (rank 5)
**Rules:**
LessExpensive($x, $y, True) »> MoreExpensive($y, $x, True) ::: If $x is less expensive
than $y, then $y is more expensive than $x
MoreExpensive($x, $y, True) »> LessExpensive($y, $x, True) ::: If $x is more expensive
than $y, then $y is less expensive than $x
Rank($x, $n, True) && Rank($y, $m, True) && $n < $m »> LessExpensive($x, $y, True)
::: If $x's rank is lower than $y's, then $x is less expensive than $y
LessExpensive($x, $y, True) && Rank($y, $n, True) && $n > 1 »> Rank($x, $m, True)
&& $m < $n ::: If $x is less expensive than $y and $y has rank greater than 1, then $x has a
lower rank than $y
**Query:**
Rank(mango, 3, True) ::: Option A
Rank(kiwi, 3, True) ::: Option B
Rank(plum, 3, True) ::: Option C
Rank(pear, 3, True) ::: Option D
Rank(watermelon, 3, True) ::: Option E

*FOL translator:*
**Predicates:**
Rank(fruit, pos) ::: fruit has price position pos, where pos $\in \{one, two, three, four, five\}$;
one = most expensive, five = cheapest
Cheaper(x, y) ::: x is cheaper (less expensive) than y
**Premises:**
Rank(watermelon, one) ::: Watermelons are the most expensive
Rank(pears, three) ::: Pears are the third-most expensive
Rank(kiwis, four) ::: Kiwis are the second-cheapest

Cheaper(kiwis, plums) ::: Kiwis are cheaper than plums

$\forall F \forall P \forall Q \big((Rank(F,P) \land Rank(F,Q)) \to (P=Q)\big)$ ::: One rank per fruit

$\forall P \forall F \forall G \big((Rank(F,P) \land Rank(G,P)) \to (F=G)\big)$ ::: One fruit per rank

Rank(mangoes, one) $\lor$ Rank(mangoes, two) $\lor$ Rank(mangoes, three) $\lor$ Rank(mangoes, four) $\lor$ Rank(mangoes, five) ::: each still-unknown fruit occupies some rank

Rank(plums, one) $\lor$ Rank(plums, two) $\lor$ Rank(plums, three) $\lor$ Rank(plums, four) $\lor$ Rank(plums, five) ::: each still-unknown fruit occupies some rank

$\forall X \forall Y \, (Rank(X, one) \land Rank(Y, two) \to Cheaper(Y,X))$ ::: "higher rank $\to$ more expensive" (10 ordered pairs)

$\forall X \forall Y \, (Rank(X, one) \land Rank(Y, three) \to Cheaper(Y,X))$ ::: "higher rank $\to$ more expensive" (10 ordered pairs)

$\forall X \forall Y \, (Rank(X, one) \land Rank(Y, four) \to Cheaper(Y,X))$ ::: "higher rank $\to$ more expensive" (10 ordered pairs)

$\forall X \forall Y \, (Rank(X, one) \land Rank(Y, five) \to Cheaper(Y,X))$ ::: "higher rank $\to$ more expensive" (10 ordered pairs)

$\forall X \forall Y \, (Rank(X, two) \land Rank(Y, three) \to Cheaper(Y,X))$ ::: "higher rank $\to$ more expensive" (10 ordered pairs)

$\forall X \forall Y \, (Rank(X, two) \land Rank(Y, four) \to Cheaper(Y,X))$ ::: "higher rank $\to$ more expensive" (10 ordered pairs)

$\forall X \forall Y \, (Rank(X, two) \land Rank(Y, five) \to Cheaper(Y,X))$ ::: "higher rank $\to$ more expensive" (10 ordered pairs)

$\forall X \forall Y \, (Rank(X, three) \land Rank(Y, four) \to Cheaper(Y,X))$ ::: "higher rank $\to$ more expensive" (10 ordered pairs)

$\forall X \forall Y \, (Rank(X, three) \land Rank(Y, five) \to Cheaper(Y,X))$ ::: "higher rank $\to$ more expensive" (10 ordered pairs)

$\forall X \forall Y \, (Rank(X, four) \land Rank(Y, five) \to Cheaper(Y,X))$ ::: "higher rank $\to$ more expensive" (10 ordered pairs)

$\forall X \forall Y \, (Cheaper(X,Y) \to \neg Cheaper(Y,X))$ ::: "cheaper" is asymmetric

**Conclusion:**

Rank(mangoes, three) ::: Option A

Rank(kiwis, three) ::: Option B

Rank(plums, three) ::: Option C

Rank(pears, three) ::: Option D

Rank(watermelon, three) ::: Option E

*SAT translator:*

**# Declarations**

fruits = EnumSort([Mangoes, Kiwis, Plums, Pears, Watermelons])

price_rank = IntSort([1, 2, 3, 4, 5])

pos = Function([fruits] -> [price_rank])

**# Constraints**

Distinct([f:fruits], pos(f)) ::: Each fruit has a unique price rank (1=cheapest, 5=most expensive)

pos(Kiwis) < pos(Plums) ::: The kiwis are less expensive than the plums

pos(Pears) == 3 ::: The pears are the third-most expensive

pos(Kiwis) == 2 ::: The kiwis are the second-cheapest

pos(Watermelons) == 5 ::: The watermelons are the most expensive

**# Options**

is_valid(pos(Mangoes) == 3) ::: A) The mangoes are the third-most expensive

is_valid(pos(Kiwis) == 3) ::: B) The kiwis are the third-most expensive

is_valid(pos(Plums) == 3) ::: C) The plums are the third-most expensive

is_valid(pos(Pears) == 3) ::: D) The pears are the third-most expensive

is_valid(pos(Watermelons) == 3) ::: E) The watermelons are the third-most expensive

Q.3 REASONING DEBATE

---

## Case Study from ProofWriter Dataset

**Context.** The bear is round. The bear likes the cow. The bear visits the lion. The cow is big. The cow is rough. The lion needs the bear. The mouse likes the lion. If someone visits the lion then they are round. If someone needs the bear then they need the lion. If someone needs the lion then they visit the lion. If someone visits the lion and the lion is round then they like the lion. If someone is rough then they need the lion. If someone likes the lion and they need the lion then the lion visits the mouse. If someone is rough then they visit the lion. If someone likes the lion and the lion is big then they are kind.

**Question.** Based on the above information, is the following statement true, false, or unknown? *The lion visits the lion.*
**Options.** A) True    B) False    C) Unknown

---

**Round 1**
*LP supporter:*
After reviewing the reasoning and deductions, I conclude that the statement "The lion visits the lion" is **True**. From rule 2 (`Needs($x, Bear) »> Needs($x, Lion)`) and the given fact `Needs(Lion, Bear)`, we deduce `Needs(Lion, Lion)`. Applying rule 3 (`Needs($x, Lion) »> Visits($x, Lion)`), this implies `Visits(Lion, Lion)`. Therefore, the statement holds true. **Key Points:**
1. `Needs(Lion, Bear)` is a given fact.
2. Rule 2 extends this to `Needs(Lion, Lion)`.
3. Rule 3 further concludes `Visits(Lion, Lion)`.
Thus, my answer remains **True**, and my confidence in this answer is **1.0**, based on the rigorous application of logical rules. **Confidence**: 1.0

*FOL supporter:*
Based on my analysis using First-Order Logic (FOL), I stand by my initial conclusion of **Unknown**. Here's why:
1. **Logic Consistency**: Both attempting to prove `Visits(Lion, Lion)` and its negation resulted in no contradiction from the given predicates. This means neither can be conclusively established as true or false.
2. **Expressiveness of FOL**: FOL captured all the nuanced dependencies and conditions in the problem (such as implications involving `Needs`, `Visits`, and `Likes`). However, the information provided still leaves the truth of `Visits(Lion, Lion)` indeterminate.
3. **Evidence Limitations**: Although key facts like `Rough(Cow) -> Visits(Cow, Lion)` and conditions involving `Needs` and `Likes` were derived, none directly establish the behavior of `Visits(Lion, Lion)`, nor does denying `Visits(Lion, Lion)` provide a definitive contradiction.
In summary, this is a clear example of the limitations of the provided context, not of FOL itself. The problem is underdefined based on the assumptions and given predicates, leading to the result of **Unknown**.
**Confidence**: 1.0

*SAT supporter:*
To analyze whether "The lion visits the lion" is true, false, or unknown, I model the logical context using Z3. Here's a formal representation of the problem with the relevant predicates:
1. Individuals: {`bear, cow, lion, mouse`}
2. Predicates:
   - `Round(x)`
   - `Big(x)`
   - `Rough(x)`
   - `Likes(x, y)`
   - `Visits(x, y)`
   - `Needs(x, y)`
   - `Kind(x)`

**Facts encoded**: `Round(bear)`, `Likes(bear, cow)`, `Visits(bear, lion)`, `Big(cow)`, `Rough(cow)`, `Needs(lion, bear)`, `Likes(mouse, lion)`
**Rules encoded**: `(Visits(x, lion) → Round(x))`, `(Needs(x, bear) → Needs(x, lion))`, `(Needs(x, lion) → Visits(x, lion))`, `(Visits(x, lion) ∧ Round(lion) → Likes(x, lion))`, `(Rough(x) → Needs(x, lion))`, `(Likes(x, lion) ∧ Needs(x, lion) → Visits(lion, mouse))`, `(Rough(x) → Visits(x, lion))`, `(Likes(x, lion) ∧ Big(lion) → Kind(x))`
To verify "The lion visits the lion", I encode: `?Visits(lion, lion)`.
Solving this using Z3 for satisfiability under the constraints:
- No explicit fact implies `Visits(lion, lion)`.
- Rules do not derive `Visits(lion, lion)`.
- Unknown/indeterminate due to lack of evidence.

Thus, **the statement remains UNKNOWN**.
**Confidence**: 1.0

*Plan-and-Solve supporter:*
The Plan-and-Solve approach confidently concludes that **"The lion visits the lion"** is true. This conclusion hinges on two key deductions:
1. **The lion needs the bear (Fact 6).**
   Using **Rule 2** ("If someone needs the bear, then they need the lion"), we deduce that the lion also needs the lion.
2. **The lion needs the lion.**
   Applying **Rule 3** ("If someone needs the lion, then they visit the lion"), it follows directly that the lion visits the lion.

This reasoning is logically consistent and follows the facts and rules provided. My confidence in this conclusion remains high unless another agent proves an inconsistency or overlooked factor.
**Confidence**: 1.0

*COT Solver supporter:*
I conclude that the answer is **C (Unknown)** because there is no information in the context that establishes whether the lion visits itself. None of the given rules or statements necessarily imply that the lion visits itself. While others visiting the lion or liking/needing the lion are explained, no direct condition or chain of reasoning leads to the conclusion about the lion visiting itself. Therefore, this remains undetermined based on the given information.
**Confidence**: 0.9

---

**Round 2**
*LP supporter:*
After reviewing FOL supporter's argument, I must directly address the critical error in their reasoning.
1. FOL state that `Needs(Lion, Lion)` "*could*" imply `Visits(Lion, Lion)` via Rule 3. There is no "could" here - Rule 3 is a universal implication: `Needs($x, Lion) → Visits($x, Lion)`. This is deterministic, not probabilistic.
2. FOL's claim of "no explicit or derived fact ensures `Visits(Lion, Lion)` conclusively" ignores that we do derive this fact through the logical chain: `Needs(Lion, Bear) → Needs(Lion, Lion) → Visits(Lion, Lion)`.

The statement "The lion visits the lion" is conclusively **True**.
**Confidence**: 1.0

*FOL supporter:*
I maintain that the answer is **C (Unknown)**. My reasoning is as follows:
1. **Logical Consistency:** Attempts to both prove `Visits(Lion, Lion)` and its negation resulted in no contradictions, which indicates that the problem is underdefined with the current rules and facts.
2. **Dependence:** While `Needs(Lion, Bear)` implies `Needs(Lion, Lion)` (via Rule 2), and `Needs(Lion, Lion)` *could* imply `Visits(Lion, Lion)` (via Rule

3), this chain of reasoning isn't fully supported because no explicit or derived fact ensures `Visits(Lion, Lion)` conclusively.

3. **No Contradiction Found:** The problem's structure does not yield a definitive resolution to the truth of the statement.

Thus, the status remains **Unknown** based on the provided information and logical framework.

**Confidence**: 1.0

*SAT supporter:*
After reviewing the arguments provided by others, I align with **Plan-and-Solve Supporter**'s reasoning. The chain of deductions from **Rule 2** (`Needs(Lion, Bear)` → `Needs(Lion, Lion)`) and then **Rule 3** (`Needs(Lion, Lion)` → `Visits(Lion, Lion)`) unequivocally establishes that the lion visits itself. This reasoning directly satisfies the logical derivation that I would also construct using Z3.

I now change my answer from **Unknown** to **True**, as the required logical steps are valid and complete.

**Confidence**: 1.0

*Plan-and-Solve supporter:*
I stand by my original answer that **"The lion visits the lion"** is **True**, supported by valid logical deductions from the provided rules and facts:

1. **Fact 6:** "The lion needs the bear."
   Applying **Rule 2** ("If someone needs the bear, then they need the lion"), we deduce that **the lion needs the lion**.

2. **Rule 3:** "If someone needs the lion, then they visit the lion."
   Since **the lion needs the lion**, it follows that **the lion visits the lion**.

This argument addresses the claim that we lack sufficient information. The facts and rules provided suffice to logically establish the truth of the statement.

**Confidence**: 1.0

*COT Solver supporter:*
After reviewing the ongoing arguments, I acknowledge that the argument provided by **Plan-and-Solve supporter** is valid: "The lion needs the bear," and by applying **Rule 2** and **Rule 3**, it follows logically that "The lion visits the lion." This was an oversight in my earlier evaluation.

Thus, I change my answer to **A (True)** as the reasoning is clear and follows directly from the rules and the given facts.

**Confidence**: 1.0

---

**Round 3**

*LP supporter:*
I remain confident that the statement **"The lion visits the lion"** is **True**. The reasoning provided earlier, and supported by others, systematically deduces this conclusion:

1. From **Fact 6**, `Needs(Lion, Bear)` is given.
2. Using **Rule 2**, this implies `Needs(Lion, Lion)`.
3. By **Rule 3**, `Needs(Lion, Lion)` directly leads to `Visits(Lion, Lion)`.

This chain of reasoning is airtight and rigorously validated. Therefore, the lion indeed visits itself.

**Confidence**: 1.0

*FOL supporter:*
After carefully reassessing the arguments, I now concur with **Plan-and-Solve Supporter** and others who assert that **"The lion visits the lion"** is **True**. The reasoning that:

1. **Fact 6:** "The lion needs the bear" leads to **"The lion needs the lion"** via **Rule 2**.
2. **The lion needing the lion** implies **"The lion visits the lion"** through **Rule 3**.

This chain of reasoning is both valid and well-supported by the premises. I change my initial conclusion from **Unknown** to **True**.

**Confidence**: 1.0

*SAT supporter:*
I maintain my revised conclusion with confidence: **"The lion visits the lion" is True.** Following the reasoning provided:
1. Fact: `Needs(Lion, Bear)` is given.
2. Rule 2: `Needs(Lion, Bear)` → `Needs(Lion, Lion)`, so `Needs(Lion, Lion)` is deduced.
3. Rule 3: `Needs(Lion, Lion)` → `Visits(Lion, Lion)`, so `Visits(Lion, Lion)` is established.

This reasoning aligns perfectly with the constraints I would also encode in Z3. Therefore, the statement is logically valid and proven to be **True**.
**Confidence**: 1.0

*Plan-and-Solve supporter:*
Thank you. My position remains that the statement **"The lion visits the lion"** is **True**. The reasoning follows directly from the structured Plan-and-Solve approach:
1. **Fact 6:** The lion needs the bear.
2. **Rule 2:** If someone needs the bear, they need the lion.
   From Fact 6, this gives **Needs(Lion, Lion)**.
3. **Rule 3:** If someone needs the lion, they visit the lion.
   Using **Needs(Lion, Lion)**, this directly implies **Visits(Lion, Lion)**.

Thus, the statement is logically derived as **True** based on a clear application of the given rules. No conflicting interpretations invalidate this deduction.
**Confidence**: 1.0

*COT Solver supporter:*
After reviewing the provided arguments, I now agree with the reasoning by **Plan-and-Solve supporter** and others who concluded that the statement is **True**. The chain of logic using **Fact 6**, **Rule 2**, and **Rule 3** is sound, showing that "The lion visits the lion" must hold. I initially missed this chain of reasoning.
**Revised Answer: A (True)**
**Confidence**: 1.0

---

**Round 4 (Final answers)**
*LP supporter:* <answer>A</answer>
*FOL supporter:* <answer>A</answer>
*SAT supporter:* <answer>A</answer>
*Plan-and-Solve supporter:* <answer>A</answer>
*COT Solver supporter:* <answer>A</answer>

---

## Case Study from LogicalDeduction Dataset

**Context.** The following paragraphs each describe a set of five objects arranged in a fixed order. The statements are logically consistent within each paragraph.
In an antique car show, there are five vehicles: a minivan, a limousine, a sedan, a tractor, and a hatchback. The tractor is the second-newest. The limousine is newer than the hatchback. The limousine is older than the sedan. The minivan is newer than the sedan.

**Question.** Which of the following is true?
**Options.** A) The minivan is the second-oldest.     B) The limousine is the second-oldest.
C) The sedan is the second-oldest.     D) The tractor is the second-oldest.     E) The hatchback is the second-oldest.

---

**Round 1**
*LP supporter:*
Using logic programming (LP), I argue that the correct option is **A) the minivan is the second-oldest**. My reasoning is as follows. First, the tractor is explicitly stated to be the *second-newest*, so exactly one vehicle is newer than the tractor. Next, from the pairwise constraints we have $S > L > H$ (i.e., the sedan is newer than the limousine, which is

newer than the hatchback), and we also have $M > S$ (the minivan is newer than the sedan). Intuitively, this pushes the trio $(H, L, S)$ toward the older side in that order while keeping the minivan ahead of the sedan; with the tractor fixed near the new end (second-newest), the remaining older-side slot that best satisfies all relations is occupied by the **minivan**, making it the *second-oldest*. Therefore, I select **A**. Confidence: 0.8

*FOL supporter:*
Let's formalize the problem using First-Order Logic (FOL) for precision:
**Definitions:**
- Let `Newer(x, y)` mean "x is newer than y."
- Objects: `Minivan (M), Limousine (L), Sedan (S), Tractor (T), Hatchback (H)`
**Given information:**
1. `Newer(S, L)` (The sedan is newer than the limousine.)
2. `Newer(L, H)` (The limousine is newer than the hatchback.)
3. `T is the second-newest.`
4. `Newer(M, S)` (The minivan is newer than the sedan.)
**Logical Analysis:** We need to determine the ordering from newest to oldest, consistent with the above statements.
**Step 1: Identify the newest vehicle.** From `T` being second-newest, exactly one vehicle is newer than `T`. Since `M > S` and `S > L > H`, neither `S`, `L`, nor `H` can be newest. Thus, the **minivan is the newest**.
**Step 2: Order the limousine, sedan, and hatchback.** From statement 1: `Newer(S, L)` $\Rightarrow S > L$. From statement 2: `Newer(L, H)` $\Rightarrow L > H$. Thus, the order for these three is $S > L > H$.
**Step 3: Position the tractor.** From statement 3, the tractor must be the second-newest vehicle. Thus, it follows the minivan: $M > T$.
**Final order (newest → oldest):** M > T > S > L > H.
**Evaluation of "second-oldest":** The second-oldest is the vehicle fourth in the order. From the final order, this is **Limousine**.
**Answer: B) The limousine is the second-oldest.**
Confidence: 1.0

*SAT supporter:*
I will represent the problem using Z3 to rigorously verify the correct ordering.
The five vehicles (minivan, limousine, sedan, tractor, hatchback) are given, and we define their ages using integer variables. Let:
- `minivan = x1`
- `limousine = x2`
- `sedan = x3`
- `tractor = x4`
- `hatchback = x5`
We impose the following constraints from the context:
1. `x4 = second newest` implies there are three vehicles older than `tractor` and one newer.
2. `x2 > x5`: The limousine is newer than the hatchback.
3. `x2 < x3`: The limousine is older than the sedan.
4. `x1 > x3`: The minivan is newer than the sedan.
To solve this:
- Define constraints for ages to ensure strict ordering, i.e., no two vehicles have the same age.
- Enumerate all possibilities for which vehicle can be "second-oldest" and validate each against the constraints.
Upon solving: The second-oldest is indeed the **limousine**, satisfying all conditions.
**Agreement/Disagreement with Reasoning:** No counterarguments provided yet.
**Conclusion:** The correct answer is **B) The limousine is the second-oldest.**
Confidence: 1.0

*Plan-and-Solve supporter:*
I have reviewed the arguments provided. The Plan-and-Solve method has clearly analyzed the four clues systematically, arranged the vehicles in order of their relative ages, and identified the second-oldest vehicle as the limousine. All steps were justified and consistent with the given information. No challenges or alternative reasoning have been introduced that contradict this deduction. Since the solution is logically sound, supported by verification of all clues, and accurately identifies the second-oldest vehicle, I maintain that the answer is **B) The limousine is the second-oldest.**
Confidence: 1.0

*COT Solver supporter:*
I stand by my initial answer of C (the sedan is the second-oldest). Here's my reasoning in summary:
1. The **tractor** is the second-newest, so it cannot be the second-oldest.
2. The **limousine** is older than the sedan and newer than the hatchback, making it older than at least two vehicles, but not the second-oldest.
3. The **minivan** is newer than the sedan, so it cannot be the second-oldest.
4. The **hatchback** is the oldest or close to it since the limousine is newer than it, thus eliminating it as the second-oldest.
5. The **sedan** is older than the minivan but not the oldest, making it the second-oldest.
Hence, the sedan fits the criteria for being the second-oldest. My confidence remains high as the logic follows directly from the constraints provided.
Confidence: 0.9

**Round 2**
*LP supporter:*
After reviewing the arguments in this collaborative debate, I realize a critical discrepancy in my initial reasoning. The process of logical deduction and the formal order articulated by the **FOL supporter**, **SAT supporter**, and **Plan-and-Solve supporter** has established the following sequence of ages:
**Newest $\rightarrow$ Oldest:** $M > T > S > L > H$.
This makes the **limousine** the second-oldest vehicle. My earlier labeling of multiple second-oldest vehicles contradicted the constraints and was erroneous. I now agree that the correct option is **B) The limousine is the second-oldest**.
Confidence: 1.0

*FOL supporter:*
After carefully reviewing the arguments from all agents and reapplying First-Order Logic, I disagree with **COT_Solver_supporter**'s deduction of the sedan as the second-oldest. According to the formal logical constraints derived, the correct ordering from newest to oldest is: $M > T > S > L > H$. Thus, the second-oldest vehicle is clearly **Limousine**, not the sedan.
This conclusion is consistent with all constraints: the tractor is second-newest, and the ordering ensures all relative relationships hold true.
**Final Answer: B) The limousine is the second-oldest.**
Confidence: 1.0

*SAT supporter:*
I disagree with COT_Solver_supporter claiming the **sedan** is the second-oldest. Z3's SMT solving clearly resolves all constraints without ambiguity and confirms the order as $M > T > S > L > H$. Hence, the **limousine** is fourth in the sequence, or, equivalently, the "second-oldest." The claim that the sedan is the second-oldest is incorrect because it is explicitly positioned third in the correct sequence (newer than limousine and hatchback but older than tractor and minivan).
Thus, the correct answer remains **B) The limousine is the second-oldest.**
Confidence: 1.0

*Plan-and-Solve supporter:*
After reviewing the discussion and the constraints again, my initial reasoning and solution remain valid. The Plan-and-Solve approach systematically deduced that the order from **oldest**

**to newest** is hatchback > limousine > sedan > tractor > minivan. Therefore, the second-oldest vehicle is the **limousine**, consistent with all clues. The argument for the sedan being the second-oldest is incorrect because it conflicts with the clue stating that the limousine is older than the sedan. This directly eliminates the sedan as a contender for second-oldest.
**Final Answer: B) The limousine is the second-oldest.**
Confidence: 1.0

*COT Solver supporter:*
I now realize that I made an error in my earlier reasoning. Upon reviewing the arguments and steps from the other agents, it is evident that the proper ordering is M > T > S > L > H (newest to oldest). From this, the **limousine** is the second-oldest vehicle, not the sedan. Thus, I agree with the corrected logical deductions provided by the other agents. The correct answer is **B) The limousine is the second-oldest.**
Confidence: 1.0

---

**Round 3**
*LP supporter:*
Thank you. After reviewing all the reasoning provided by the agents and scrutinizing my initial position, I fully align with the consensus reached by the **FOL supporter**, **SAT supporter**, and others. The logical deductions clearly establish that the correct order is M > T > S > L > H, making the **limousine** the second-oldest vehicle. My initial mislabeling has been corrected.
**Final Answer: B) The limousine is the second-oldest.**
Confidence: 1.0

*FOL supporter:*
Given the thorough reasoning and agreement across multiple approaches, including clear formal deductions, the correct ordering is M > T > S > L > H (newest to oldest). Therefore, the second-oldest is the **limousine**.
**Final Answer: B) The limousine is the second-oldest.**
Confidence: 1.0

*SAT supporter:*
I maintain that the correct answer is **B) The limousine is the second-oldest**. Using Z3 to represent the constraints ensures a complete and rigorous solution, confirming the order: M > T > S > L > H. No contradictions remain in this conclusion, and all logical relationships are satisfied.
Confidence: 1.0

*Plan-and-Solve supporter:*
The correct answer remains **B) The limousine is the second-oldest.** The Plan-and-Solve approach has systematically deduced the correct ordering and verified it against all constraints. Other agents have corroborated this conclusion, and alternative claims have been refuted. The reasoning is consistent, logical, and validated.
Confidence: 1.0

*COT Solver supporter:*
After reviewing and assessing the arguments presented by other agents, I acknowledge my earlier error. The systematic deductions using constraints demonstrate that the correct order is M > T > S > L > H. From this order, the **limousine** is undoubtedly the second-oldest vehicle. **Final Answer: B) The limousine is the second-oldest.**
Confidence: 1.0

---

**Round 4 (Final answers)**
*LP supporter:* <answer>B</answer>
*FOL supporter:* <answer>B</answer>
*SAT supporter:* <answer>B</answer>
*Plan-and-Solve supporter:* <answer>B</answer>
*COT Solver supporter:* <answer>B</answer>

