# OpenReview forum: "MAD-Logic: Multi-Agent Debate Enhances Symbolic Translation and Reasoning"
_ICLR.cc/2026/Conference — ICLR 2026 Poster_

### Official Review · Reviewer_Q5K8 · 2025-10-16

**Soundness:** 1
**Presentation:** 2
**Contribution:** 2
**Rating:** 4
**Confidence:** 5

**Summary:**

The manuscript proposes a sparse multi-agent debate framework that first translates natural-language logical questions into three symbolic languages (LP, FOL, SAT) via agent debate, then lets symbolic and natural-language agents debate the answers, finally voting on the result. An adaptive communication gate prunes low-value inter-agent messages. Experiments on three logical-QA benchmarks report accuracy improvements over a leaderboard that has already reached a degree of saturation.

**Strengths:**

Table 1 shows +0.92 pp average gain over the fully-connected multi-agent baseline on GPT-4, demonstrating that merging symbolic and NL reasoning is more accurate than either alone.

Table 3 ablation adding each solver (LP→SAT→FOL) steadily lifts accuracy (88.06 % → 95.44 %), confirming that diversity of symbolic systems matters.

Algorithm 1 (Appendix E) gives reproducible pseudo-code, enhancing technical soundness.

**Weaknesses:**

The experiments are confined to three English synthetic datasets—all of which have already reached saturation. Their small size further undermines the significance of the findings, and no evidence is provided for performance on natural text, whether from specialized domains (e.g., legal, medical) or multilingual contexts.

Table 1 presents performance percentages without accompanying variance estimates (e.g., standard deviation, confidence intervals). Given the dataset’s size, small discrepancies in reported performance could easily be attributed to random noise rather than meaningful differences.

No token cost comparison is provided against two critical baselines: human-only reasoning, or single-agent Chain-of-Thought (CoT) operating within the same budget. This makes the "token saving" claim questionable, as the reference point for comparison remains unspecified.

No timing analysis: how often do Pyke/Prover9/Z3 timeout?

Eq. (1) introduces a preference score but no lemma guarantees that pruning preserves consensus or convergence. No ablation on λ beyond a single value (1.0).

I'm concerned with the prompt sensitivity. Appendix F shows one fixed prompt per role; no robustness check with alternative prompt wordings.

GPT-4 and Claude may have seen ProofWriter during pre-training; no sanitisation check is mentioned.

**Questions:**

Suggestions:
Run pipeline on FOLIO multilingual, Chinese LogiQA-V2 and LogiEval-Hard; report delta and failure rates.
Repeat main experiments with three prompt seeds; report mean and std-dev in Table 1.

---

> ### Author Response · Authors · 2025-11-24
>
> We thank Reviewer Q5K8 for the insightful comments and valuable feedback. We carefully address each of the concerns as follows:
>
> > W1: "The experiments are confined to three English synthetic datasets—all of which have already reached saturation. Their small size further undermines the significance of the findings, and no evidence is provided for performance on natural text, whether from specialized domains (e.g., legal, medical) or multilingual contexts."
>
> **Response:** Thank you for pointing out the issue! Following your suggestion, we have added **three more challenging benchmarks that include natural text from specialized domains and multilingual contexts:**
>
> - **AR-LSAT**: analytical reasoning questions from **real LSAT exams in the legal domain**, requiring multi-step reasoning over constraints.
> - **FOLIO**: **natural-language** premise–hypothesis pairs annotated with first-order-logic formulas to test logically complex entailment.
> - **Chinese LogiQA-V2**: Chinese **reading-comprehension questions from civil-service exams** that target fine-grained logical reasoning in **Chinese**.
>
> The full results across four base models are reported in **Tables R1.1–R1.4, where we include the strongest symbolic reasoning baselines** (SymbCoT and LogicLM) and the **strongest multi-agent debate baseline** (CortexDebate). Note that **CortexDebate (w/ NL-SL)** reuses our translation stage, solver stage, and agent roles, and thus differs from our method only in the communication graph topology, while **CortexDebate (w/o NL-SL)** corresponds to the original pure-NL setup used in prior work.
>
> **On these settings, the tasks are clearly far from saturated,** and our approach achieves substantial improvements over direct LLM reasoning, symbolic reasoning, and multi-agent baselines, which we believe makes the effectiveness of our method more convincing.
>
>
>
> > W2: "Table 1 presents performance percentages without accompanying variance estimates (e.g., standard deviation, confidence intervals). Given the dataset’s size, small discrepancies in reported performance could easily be attributed to random noise rather than meaningful differences."
>
> **Response:** **Our original setting used temperature = 0 for deterministic reproducibility**. Following the suggestion, we repeated main experiments using **three semantically equivalent prompt rewrites** (via GPT-5) to introduce controlled variance. We now report **mean ± std** **(Table R2)**, with several improvements statistically significant to the most competitive baseline (marked *).
>
> Note that **CortexDebate** reuses our translation stage, solver stage, and agent roles, and thus differs from our method only in the communication graph topology

---

> ### Author Response · Authors · 2025-11-24
>
> **Table R1.1: Performance of different methods on Qwen2.5-7B-Instruct.**
>
> | Method                   | ProofWriter | ProntoQA  | LogiDeduct | AR-LSAT   | FOLIO     | Chinese LogiQA-V2 |
> | ------------------------ | ----------- | --------- | ---------- | --------- | --------- | ----------------- |
> | Direct Answer            | 40.17       | 55.20     | 42.67      | 24.24     | 32.35     | 62.96             |
> | CoT                      | 40.50       | 75.60     | 40.33      | 29.87     | 53.24     | 59.32             |
> | LogicLM                  | 61.63       | 73.80     | 63.00      | 14.29     | 59.61     | 25.33             |
> | SymbCoT                  | 70.17       | 80.60     | 60.00      | 32.03     | 57.39     | 65.44             |
> | CortexDebate (w/o NL-SL) | 47.50       | 78.20     | 39.67      | 19.91     | 53.92     | 62.96             |
> | CortexDebate (w/ NL-SL)  | 75.83       | 84.00     | 67.00      | 35.93     | 63.73     | 66.92             |
> | Ours (w/o sparse)        | 74.67       | 85.20     | 68.33      | 35.06     | 62.24     | 67.98             |
> | **Ours (w/ sparse)**     | **76.50**   | **86.40** | **67.67**  | **37.20** | **65.68** | **68.11**         |
> ||
>
>
> **Table R1.2: Performance of different methods on GPT-4o-mini.**
>
> | Method                   | ProofWriter | ProntoQA  | LogiDeduct | AR-LSAT   | FOLIO     | Chinese LogiQA-V2 |
> | ------------------------ | ----------- | --------- | ---------- | --------- | --------- | ----------------- |
> | Direct Answer            | 53.50       | 62.60     | 56.00      | 19.91     | 59.80     | 57.31             |
> | CoT                      | 43.67       | 75.00     | 70.33      | 19.04     | 61.76     | 54.30             |
> | LogicLM                  | 58.67       | 76.00     | 73.00      | 22.94     | 34.80     | 25.00             |
> | SymbCoT                  | 70.33       | 82.80     | 75.33      | 26.00     | 69.10     | 58.87             |
> | CortexDebate (w/o NL-SL) | 60.87       | 80.40     | 70.67      | 21.21     | 62.75     | 62.02             |
> | CortexDebate (w/ NL-SL)  | 75.17       | 89.00     | 82.33      | **34.20** | 75.00     | 64.03             |
> | Ours (w/o sparse)        | 74.00       | 89.40     | 82.33      | 33.81     | 74.02     | 65.91             |
> | **Ours (w/ sparse)**     | **76.33**   | **90.60** | **84.67**  | **34.20** | **76.47** | **67.29**         |
> ||
>
> **Table R1.3: Performance of different methods on GPT-4.**
>
> | Method                   | ProofWriter | ProntoQA    | LogiDeduct | AR-LSAT   | FOLIO      | Chinese LogiQA-V2 |
> | ------------------------ | ----------- | ----------- | ---------- | --------- | ---------- | ----------------- |
> | Direct Answer            | 53.50      | 75.40      | 59.00     | 32.90     | 65.20      | 62.27            |
> | CoT                      | 67.17      | 81.20      | 69.67     | 35.06     | 70.59      | 65.22            |
> | LogicLM                  | 79.17      | 93.40      | 87.00     | 40.86     | 76.96      | 25.99            |
> | SymbCoT                  | 82.33      | 96.00      | 86.33     | 42.86     | 80.39      | 70.57            |
> | CortexDebate (w/o NL-SL) | 73.50      | 89.00      | 84.33     | 34.20     | 75.00     | 68.17            |
> | CortexDebate (w/ NL-SL)  | 90.83      | 99.60      | 92.33     | 51.08     | 84.80     | 74.13            |
> | Ours (w/o sparse)        | 90.17      | 99.40      | 94.00     | 50.42     | 84.31     | 74.01            |
> | **Ours (w/ sparse)**     | **92.00**  | **100.00** | **94.33** | **53.25** | **86.27** | **74.76**        |
> ||
>
>
> **Table R1.4: Performance of different methods on DeepSeek-V3.**
>
> | Method                   | ProofWriter | ProntoQA   | LogiDeduct | AR-LSAT   | FOLIO     | Chinese LogiQA-V2 |
> | ------------------------ | ----------- | ---------- | ---------- | --------- | --------- | ----------------- |
> | Direct Answer            | 68.33       | 79.20      | 85.33      | 36.80     | 66.18     | 74.33             |
> | CoT                      | 71.83       | 85.00      | 83.00      | 45.45     | 76.96     | 77.97             |
> | LogicLM                  | 80.50       | 83.20      | 93.33      | 43.72     | 78.92     | 28.63             |
> | SymbCoT                  | 85.83       | 98.00      | 94.00      | 47.02     | 81.37     | 81.98             |
> | CortexDebate (w/o NL-SL) | 83.17       | 94.00      | 90.00      | 44.16     | 79.90     | 78.97             |
> | CortexDebate (w/ NL-SL)  | 93.00       | 99.80      | 99.67      | 74.03     | 88.73     | 83.04             |
> | Ours (w/o sparse)        | 92.83       | 99.80      | **100.00** | 73.62     | 89.22     | 85.68             |
> | **Ours (w/ sparse)**     | **93.33**   | **100.00** | **100.00** | **75.76** | **90.67** | **86.93**         |
> ||

---

> ### Author Response · Authors · 2025-11-24
>
> **Table R2: Mean ± standard deviation over 3 prompt paraphrases. `*`  means statistical significance using pairwise t-test
> with p-value <0.05.**
>
> | Model       | Method       | ProofWriter       | ProntoQA         | LogiDeduct        | AR-LSAT           | FOLIO             | Chinese LogiQA-V2 |
> | ----------- | ------------ | ----------------- | ---------------- | ----------------- | ----------------- | ----------------- | ----------------- |
> | GPT-4       | w/o sparse   | 89.78 ± 0.35      | 99.20 ± 0.20     | 93.78 ± 0.19      | 50.29 ± 0.12      | 84.47 ± 0.28      | 74.11 ± 0.19      |
> | GPT-4       | CortexDebate | 90.78 ± 0.09      | 99.67 ± 0.31     | 92.44 ± 0.51      | 51.42 ± 0.98      | 85.13 ± 0.57      | 73.95 ± 0.22      |
> | GPT-4       | w/ sparse    | **91.83 ± 0.17*** | **99.87 ± 0.12** | **94.61 ± 0.26*** | **53.17 ± 0.14*** | **86.60 ± 0.43*** | **74.66 ± 0.14*** |
> | DeepSeek-V3 | w/o sparse   | 92.61 ± 0.38      | 99.80 ± 0.20     | 99.11 ± 0.51      | 73.31 ± 0.27      | 89.22 ± 0.49      | 85.75 ± 0.08      |
> | DeepSeek-V3 | CortexDebate | 92.83 ± 0.17      | 99.80 ± 0.20     | 99.67 ± 0.33      | 73.88 ± 0.25      | 89.18 ± 0.44      | 83.04 ± 0.27      |
> | DeepSeek-V3 | w/ sparse    | **93.50 ± 0.14*** | **99.93 ± 0.12** | **99.89 ± 0.19**  | **75.76 ± 0.44*** | **90.85 ± 0.28*** | **86.55 ± 0.48*** |
> ||
>
> > W3: "No token cost comparison is provided against two critical baselines: human-only reasoning, or single-agent Chain-of-Thought (CoT) operating within the same budget. This makes the "token saving" claim questionable, as the reference point for comparison remains unspecified."
>
> **Response:** Thank you for pointing out this issue. We apologize for our ambiguity.
>
> - Regarding token cost, we clarify that our “token saving’’ claim is defined **within the family of multi‑agent debate methods**: **the reference point is a fully connected topology (“Ours w/o sparse’’)** with the same agents, NL–SL translation stage, and solver components as our method, and our sparse variant only prunes communication edges.
>
> - Tables R3.1–R3.2 report the average token cost and accuracy for **Direct Answer, CoT, fully connected debate, and our sparse debate** on six datasets with GPT‑4 and DeepSeek‑V3. Across all datasets, sparse communication reduces the token cost of multi‑agent debate by roughly **20–40%** relative to “Ours w/o sparse’’ while maintaining or slightly improving accuracy. Direct Answer and CoT indeed use far fewer tokens but also achieve much lower accuracy; our goal is **to make multi‑agent debate itself more efficient**, rather than to claim that it is more token‑efficient than single‑agent CoT.
>
> - A “human‑only’’ baseline is left for future work, since there is currently no standardized way to measure human “tokens’’
>
> **Table R3.1: Average token cost and accuracy on GPT-4.**
>
> |                      | ProntoQA |             | ProofWriter |            | LogiDeduct |            | AR-LSAT |            | FOLIO  |            | Chinese LogiQA-V2 |            |
> | -------------------- | -------- | ----------- | ----------- | ---------- | ---------- | ---------- | ------- | ---------- | ------ | ---------- | ----------------- | ---------- |
> | **Method**           | token    | acc         | token       | acc        | token      | acc        | token   | acc        | token  | acc        | token             | acc        |
> | Direct               | 252      | 75.40%      | 315         | 53.50%     | 286        | 59.00%     | 144     | 32.90%     | 100    | 65.20%     | 355               | 62.27%     |
> | CoT                  | 303      | 81.20%      | 498         | 67.17%     | 261        | 69.67%     | 747     | 35.06%     | 177    | 70.59%     | 554               | 65.22%     |
> | Ours (w/o sparse)    | 46,502   | 99.40%      | 51,358      | 90.17%     | 54,857     | 94.00%     | 35,012  | 50.42%     | 23,167 | 84.31%     | 19,015            | 74.01%     |
> | **Ours (w/ sparse)** | 35,854   | **100.00%** | 42,617      | **92.00%** | 43,264     | **94.67%** | 28,044  | **53.25%** | 19,023 | **86.27%** | 14,733            | **74.76%** |
> ||

---

> ### Author Response · Authors · 2025-11-24
>
> **Table R3.2: Average token cost and accuracy on DeepSeek-V3.**
>
> |                      | ProntoQA |             | ProofWriter |            | LogiDeduct |             | AR-LSAT |            | FOLIO  |            | Chinese LogiQA-V2 |            |
> | -------------------- | -------- | ----------- | ----------- | ---------- | ---------- | ----------- | ------- | ---------- | ------ | ---------- | ----------------- | ---------- |
> | **Method**           | token    | acc         | token       | acc        | token      | acc         | token   | acc        | token  | acc        | token             | acc        |
> | Direct               | 165      | 79.20%      | 184         | 68.33%     | 371.85     | 85.33%      | 2,324   | 36.80%     | 632    | 66.18%     | 420               | 74.33%     |
> | CoT                  | 257      | 85.00%      | 296         | 71.83%     | 497.55     | 83.00%      | 2,453   | 45.45%     | 651    | 76.96%     | 488               | 77.97%     |
> | Ours (w/o sparse)    | 57,366   | 99.80%      | 25,059      | 92.83%     | 70,464     | 100.00%     | 38,973  | 76.79%     | 24,410 | 89.22%     | 19,334            | 85.68%     |
> | **Ours (w/ sparse)** | 36,702   | **100.00%** | 24,115      | **93.33%** | 46,107     | **100.00%** | 30,152  | **79.65%** | 19,832 | **90.67%** | 15,547            | **86.93%** |
> ||
>
>
> > W4: "No timing analysis: how often do Pyke/Prover9/Z3 timeout?"
>
> **Response:**
>
> - Regarding timing, **for overall end-to-end cost**, **we primarily report token usage rather than wall-clock time**, because **API latency for commercial LLMs is highly unstable** and depends on external factors (server load, batching, etc.), making direct runtime comparisons across methods noisy and hard to reproduce.
>
> - For the **symbolic solvers** (Pyke, Prover9, Z3), we provide a timing analysis in **Table R4**. We measure, on executable instances only, the average solving time (in seconds) and the timeout rate under a fixed timeout threshold. These results indicate that symbolic solving rarely times out and does not dominate the overall computational cost of our method.
>
> **Table R4: Average solving time (seconds) and timeout rate (%) of symbolic solvers (measured on executable samples only). “–” means the solver is not used on that dataset.**
>
> | Dataset           | Pyke avg time | Pyke timeout | Prover9 avg time | Prover9 timeout | Z3 avg time | Z3 timeout |
> | ----------------- | ------------- | ------------ | ---------------- | --------------- | ----------- | ---------- |
> | ProntoQA          | 0.031         | 0.00%        | 0.097            | 0.00%           | 0.065       | 0.00%      |
> | ProofWriter       | 0.028         | 0.00%        | 0.102            | 0.00%           | 0.065       | 0.00%      |
> | LogiDeduct        | 0.031         | 0.00%        | 3.762            | 2.23%           | 0.072       | 0.00%      |
> | FOLIO             | –             | –            | 0.568            | 0.49%           | 0.063       | 0.00%      |
> | AR-LSAT           | –             | –            | 4.419            | 3.40%           | 0.061       | 0.00%      |
> | Chinese LogiQA-V2 | –             | –            | 1.487            | 0.00%           | 0.063       | 0.00%      |
> ||

---

> ### Author Response · Authors · 2025-11-24
>
> > W5: "Eq. (1) introduces a preference score but no lemma guarantees that pruning preserves consensus or convergence."
>
> **Response:** Thank you for your insightful concern! We agree that Eq. (1) does not provide a formal convergence guarantee. Deriving a lemma that links the learned preference scores to consensus or convergence is highly non-trivial and is beyond the scope of our work. Instead, we provide **empirical evidence** that our sparse pruning does not harm consensus in practice.
>
> - Concretely, we measure **vote entropy** at the final debate round. For each question $q$, let $\mathcal{Y}$ be the set of answer options, $n$ be the total number of agents participating debate, $c_y$ the number of agents voting for option $y$. **We define the normalized vote entropy**
>   **$$H_{\text{norm}}(q) = -\frac{1}{\log|\mathcal{Y}|}\sum_{y\in\mathcal{Y}}\frac{c_y}{n}\log\frac{c_y}{n}\in[0,1],$$**
>   **where 0 means perfect agreement and 1 means maximally split votes**.
> - **Table R5** reports the average $H_{\text{norm}}$ over questions for GPT‑4 and DeepSeek‑V3, comparing a fully connected debate graph (“w/o sparse”) and our sparse communication graph (“w/ sparse”). We find that **applying sparse communication** **does not much change the vote entropy** across all datasets, indicating that the consensus behavior of the debate remains stable. In fact, the **entropy** under the sparse topology is **almost identical** to that of the fully connected graph—and in several cases **even** **slightly lower**—showing that pruning generally preserves convergence.
>
>
>
> **Table R5: Average normalized vote entropy $H_{\text{norm}} \in [0,1]$ across questions (lower = stronger consensus). Results are reported for the final debate round.**
>
> | Model       | Method     | ProofWriter | ProntoQA | LogiDeduct | AR-LSAT | FOLIO  | Chinese LogiQA-V2 |
> | ----------- | ---------- | ----------- | -------- | ---------- | ------- | ------ | ----------------- |
> | GPT-4       | w/o sparse | 0.0224      | 0.0014   | 0.0055     | 0.0881  | 0.0655 | 0.0710            |
> | GPT-4       | w/ sparse  | 0.0243      | 0.0014   | 0.0054     | 0.0893  | 0.0687 | 0.0733            |
> | DeepSeek-V3 | w/o sparse | 0.2359      | 0.0540   | 0.0034     | 0.0346  | 0.2832 | 0.0142            |
> | DeepSeek-V3 | w/ sparse  | 0.2334      | 0.0555   | 0.0034     | 0.0340  | 0.2732 | 0.0150            |
> ||
>
>
> > W6: "No ablation on λ beyond a single value (1.0)."
>
> **Response:** We added a sensitivity analysis for the sparsity hyperparameter λ in **Table R6**. Experiment shows that accuracy is stable once λ ≥ 0.5 (variations within ≈1 pp), while the token saving rate increases monotonically; so we choose λ = 1.0 as an accuracy–efficiency trade-off.
>
> **Table R6. Sensitivity of accuracy and token saving rate to λ in the sparse gate.**
> All values are percentages (%). “Tok” = token saving rate.
>
> | λ    | Model    | PW Acc    | PW Tok    | PQA Acc    | PQA Tok   | LD Acc     | LD Tok    |
> | ---- | -------- | --------- | --------- | ---------- | --------- | ---------- | --------- |
> | 0    | GPT-4    | 90.17     | 8.52      | 99.20      | 11.45     | 92.67      | 5.63      |
> | 0.5  | GPT-4    | **92.17** | 13.62     | 99.60      | 18.31     | 94.00      | 10.29     |
> | 1.0  | GPT-4    | 92.00     | 17.03     | **100.00** | 22.89     | **94.33**  | 12.35     |
> | 1.5  | GPT-4    | 91.50     | 18.73     | 99.80      | 25.18     | 93.67      | 13.02     |
> | 2.0  | GPT-4    | 91.50     | **19.59** | **100.00** | **26.32** | 93.33      | **13.31** |
> |      |          |           |           |            |           |            |           |
> | 0    | DeepSeek | 92.17     | 3.88      | 99.60      | 18.01     | 98.33      | 17.29     |
> | 0.5  | DeepSeek | 93.00     | 6.02      | 99.80      | 28.82     | 98.67      | 27.66     |
> | 1.0  | DeepSeek | 93.33     | 13.15     | 100.00     | 36.02     | **100.00** | 34.57     |
> | 1.5  | DeepSeek | **93.67** | 15.15     | 99.60      | 36.53     | 99.33      | 38.44     |
> | 2.0  | DeepSeek | 93.33     | **15.34** | **100.00** | **36.87** | 99.67      | **39.31** |
> ||
>
>
> > W7: "I'm concerned with the prompt sensitivity. Appendix F shows one fixed prompt per role; no robustness check with alternative prompt wordings."
>
> **Response:** We have repeated main experiments using **three semantically equivalent prompt rewrites** (via GPT-5) to perform robustness check with alternative prompt wordings in **Table R2** as in **response to W2.**

---

> ### Author Response · Authors · 2025-11-24
>
> > W8: "GPT-4 and Claude may have seen ProofWriter during pre-training; no sanitisation check is mentioned."
>
> **Response:**
>
> - We appreciate the reviewer’s concern. Since the pre-training corpora of closed-source models such as GPT-4 and Claude are not publicly disclosed, we cannot strictly guarantee that ProofWriter has never appeared in their training data. This limitation is shared by most recent neuro-symbolic reasoning works that evaluate GPT-4/Claude on ProofWriter (e.g., LogicLM, SymbCoT, DetermLR, Aristotle), and we follow the same standard evaluation protocol; we will explicitly mention this as a limitation in the revised version.
>
> - **However, our conclusions do not rely on ProofWriter alone**. In the revision, we evaluate our method on **four different base models** (Qwen2.5-7B-Instruct, GPT-4o-mini, GPT-4, DeepSeek-V3) and **six datasets**, **including more challenging and realistic benchmarks** such as **AR-LSAT, FOLIO, and Chinese LogiQA-V2** **(see Tables R1.1–R1.4)**. Across all these settings our approach consistently outperforms strong symbolic baselines and multi-agent debate baseline.
>
> - **After carefully reading a very related paper titled "Evaluating the Logical Reasoning Ability of ChatGPT and GPT-4" (Liu et al., 2023), we further add the limitation discussion in our "Conclusion" section.** The base idea motivates from an observation that the LLM's logical reasoning performance drops significantly when handling newly released and out-of-distribution datasets, thus it is crucial to extend our approach to accommodate out-of-distribution scenarios.
>
> Going forward, we plan to explore incorporating recent techniques for black-box LLM contamination checks in future versions of this line of work.
>
> > Q1: "Suggestions: Run pipeline on FOLIO multilingual, Chinese LogiQA-V2 and LogiEval-Hard; report delta and failure rates. Repeat main experiments with three prompt seeds; report mean and std-dev in Table 1."
>
> **Response:** Thank you very much for your valuable suggestions!
>
> - Following your suggestion to test on more challenging and multilingual benchmarks, we have added **AR-LSAT**, **FOLIO** (English), and **Chinese LogiQA-V2**, as described in our response to **W1**. The full results are reported in **Tables R1.1–R1.4**.
> - Regarding the requested datasets, we were unfortunately not able to locate an official release of **“FOLIO multilingual”**, and **LogiEval-Hard** is, to the best of our knowledge, still under review for ICLR 2026 and not yet publicly available. We will be happy to evaluate on these datasets once they are released.
> - We follow the reviewer's suggestion to repeat our main experiments with **three prompt paraphrases** and report the **mean ± standard deviation** in **Table R2**, as discussed in **W2**.
> - Furthermore, we **provide detailed translation-quality analyses via Translation Common Error Rate (T-CER_n) and direct validation against gold FOL formulas on FOLIO** in Section 5.5 (Tables 8–9).
>
>
> ------
>
> We are eager to hear your feedback! We’d deeply appreciate it if you could let us know whether your concerns have been addressed.

---

> > ### Author Response · Authors · 2025-11-28
> >
> > Dear reviewer Q5K8,
> >
> > Thank you once again for your detailed review and the valuable time you have dedicated to our work! We have uploaded our revised PDF incorporating *all* mentioned changes, which highlighted in "$\textcolor{blue}{blue}$" for facilitating checking.
> >
> > FYI, we made the following changes:
> >
> > - In response to ```W1``` and ```Q1(a)```, we **add extend evaluation to real-world datasets (FOLIO, Chinese LogiQA-V2, and AR-LSAT)** in Table 2.
> >
> > - In response to ```W2```, ```W7``` and ```Q1(c)```, we **perform significance testing and prompt-sensitivity analysis by rewriting prompts into three paraphrases** in Table 4.
> >
> > - In response to ```W3```, we **compare token cost and accuracy between Direct Answer, CoT, and our method**, clarifying that our token-saving claim is defined relative to a fully connected multi-agent debate baseline in Tables 14–15.
> >
> > - In response to ```W4```, we **add timing analysis for symbolic solvers** to show that solver timeouts are rare and do not dominate overall cost in Table 10.
> >
> > - In response to ```W5```, we **introduce vote-entropy–based consensus analysis** to show that sparse pruning preserves (and sometimes slightly improves) consensus in multi-agent debate in Appendix F.
> >
> > - In response to ```W6```, we **add sensitivity experiments on $\lambda$** in Table 7.
> >
> > - In response to ```W8```, we **add experiments using more backbones and benchmarks other than ProofWriter** and **discuss the limitation** in our "Conclusion" section.
> >
> > - In response to ```Q1(b)```, we **provide detailed translation-quality analyses via Translation Common Error Rate (T-CER_n) and direct validation against gold FOL formulas on FOLIO** in Section 5.5 (Tables 8–9).
> >
> > We sincerely hope the reviewer will re-evaluate our paper based on the additional experiments and explanations provided above. We look forward to your feedback if you have any further questions!
> >
> > Thanks for your time,
> >
> > Authors of #14334

---

### Official Review · Reviewer_awiG · 2025-10-17

**Soundness:** 3
**Presentation:** 2
**Contribution:** 2
**Rating:** 4
**Confidence:** 2

**Summary:**

This paper proposes a sparse multi-agent debate framework  to improve the performance of large language models in complex symbolic logic reasoning tasks. This approach combines the advantages of symbolic language (SL) and natural language (NL) reasoning: multiple agents translate the question into different formal logics (LP, FOL, SAT) and optimize the translation through debate. The SL and NL agents then debate and vote on the final answer over multiple rounds. To improve efficiency, the authors design an adaptive sparse communication mechanism, which reduces token consumption while improving accuracy.

My opinion may be negative at this stage, if the author can address my concerns, I will increase my score.

**Strengths:**

1. Achieved very good performance on multiple data sets, with many indicators reaching 100%.

2. The experiments are very substantial, and a large number of experiments are done to verify the performance and efficiency of the proposed method.

**Weaknesses:**

1. The paper repeatedly states that LLMs struggle with complex logical reasoning. However, the metrics shown in Table 1 are all very high, which appears inconsistent with this claim. Moreover, I am curious about the results of directly testing the LLMs—i.e., having the LLM answer the questions without any additional framework—as such baseline results might better support the authors’ argument.

2. While most of the metrics in Table 1 are high, the proposed method does not achieve significant improvements over the baseline. The numerical metrics indicate that all three datasets appear simple, with 100% accuracy achieved multiple times, making the experiment less than convincing.

3. The paper uses the latest powerful baseline models. Does this mean that the method proposed in this paper has high requirements for the base model? Is it applicable to those ordinary models or smaller models such as qwen 2.5 7B?

**Questions:**

1. Is there a more challenging dataset now? This is necessary to verify the effectiveness of the method in this paper.

2. How is the confidence score obtained in sparse communication?

---

> ### Author Response · Authors · 2025-11-24
>
> Thank you very much for your helpful comments! Below, we address your questions and indicate the changes we’ve made thanks to your valuable suggestions.
>
> To provide a clearer and more comprehensive evaluation, **we added** **two smaller/ordinary base models** **(Qwen2.5-7B instruct and GPT-4o-mini) and three more challenging datasets (AR-LSAT, FOLIO, and Chinese LogiQA-V2), with full results reported in Tables R1.1–R1.4.** We included the **strongest symbolic reasoning baselines** (SymbCoT and LogicLM) and the **strongest multi-agent debate baseline** (CortexDebate). We clarify that **CortexDebate (w/ NL-SL)** reuses our translation stage, solver stage, and agent roles—thus differing from our method only in the communication graph topology. **CortexDebate (w/o NL-SL)** corresponds to the original pure-NL version as used in its original work.
>
> > W1: "The paper repeatedly states that LLMs struggle with complex logical reasoning. However, the metrics shown in Table 1 are all very high, which appears inconsistent with this claim. Moreover, I am curious about the results of directly testing the LLMs—i.e., having the LLM answer the questions without any additional framework—as such baseline results might better support the authors’ argument."
>
> **Response:** Thank you for your helpful suggestion! **We have** **added direct LLM reasoning baselines (Direct Answer)** for all four base models (Qwen2.5-7B, GPT-4o-mini, GPT-4, and DeepSeek-V3) in **Tables R1.1–R1.4**. These direct-baseline results clearly show that, without any reasoning framework, LLMs still make many errors on complex logical datasets (e.g., only 20–63% accuracy for Qwen2.5-7B and GPT-4o-mini, and 30–80% even for GPT-4 and DeepSeek-V3). This supports our statement that “LLMs struggle with complex logical reasoning”.
>
>
>
> > W2: "While most of the metrics in Table 1 are high, the proposed method does not achieve significant improvements over the baseline. The numerical metrics indicate that all three datasets appear simple, with 100% accuracy achieved multiple times, making the experiment less than convincing."
>
> **Response:**
>
> - We apologize for the ambiguity in our original paper writing. In **Table 1** of our paper, the reported results for **SparseMAD** and **CortexDebate** already adopt our proposed **NL–SL hybrid reasoning stage**; thus, they differ from our method mainly in the debate topology. This choice was intended to isolate the effect of debate topology structure, but **it also makes these baselines stronger than their original implementations**. To compare fairly, we now report **“CortexDebate (w/o NL–SL)”** in **Tables R1.1–R1.4**.
> - Moreover, we have expanded our evaluation to **additional base models** and **more challenging benchmarks** with full results in **Tables R1.1–R1.4**. On these settings, the tasks are clearly far from saturated, and our approach achieves substantial improvements over direct LLM reasoning, symbolic reasoning, and multi-agent baselines, which we believe makes the effectiveness of our method more convincing.
>
> > W3: "The paper uses the latest powerful baseline models. Does this mean that the method proposed in this paper has high requirements for the base model? Is it applicable to those ordinary models or smaller models such as qwen 2.5 7B?"
>
> **Response:** Thank you for raising this concern. As shown in **Tables R1.1–R1.4** where **we added small (Qwen2.5-7B-Instruct) and ordinary (GPT-4o-mini) models**, our method achieves significant improvements over baselines, indicating our approach is fully applicable to those non-frontier models.
>
>
>
> > Q1: "Is there a more challenging dataset now? This is necessary to verify the effectiveness of the method in this paper."
>
> **Response:** Thank you very much for the suggestion, which is very helpful! Following your suggestion, we have added **three more challenging benchmarks:**
>
> - **AR-LSAT**, which consists of analytical reasoning questions from LSAT exams and requires multi-step combinatorial reasoning over constraints.
> - **FOLIO**, a human-annotated benchmark for natural language reasoning with paired first-order-logic (FOL) formulas and logically complex premises–hypotheses.
> - **Chinese LogiQA-V2**, an improved logical reasoning reading-comprehension dataset adapted from the Chinese Civil Service Entrance Examination, focusing on fine-grained logical reasoning in Chinese.
>
> The full results are reported in **Tables R1.1–R1.4**, where our method consistently outperforms direct LLM reasoning, symbolic baselines, and multi-agent baselines, confirming its effectiveness on harder settings.
>
>
>
> > Q2: "How is the confidence score obtained in sparse communication?"
>
> **Response:** The confidence score is generated by each LLM agent in the same response turn as its predicted answer. Concretely, every agent is prompted to output: (i) its predicted label, (ii) the reasoning trace, and (iii) a scalar confidence value in [0,1] during communication.

---

> > ### Comment · Reviewer_awiG · 2025-11-27
> > **Reply to the rebuttal**
> >
> > Thans for the author's reply. I greatly appreciate the author's detailed explanation of relpy and their efforts.
> >
> > I have two questions:
> > 1. Are the confidence scores generated by LLM itself reliable?
> > 2. Given the amount of experimentation the author has added, I worry whether these changes will be fully incorporated into the final version, meaning the current version is quite incomplete.

---

> ### Author Response · Authors · 2025-11-24
>
> **Table R1.1: Performance of different methods on Qwen2.5-7B-Instruct.**
>
> | Method                   | ProofWriter | ProntoQA  | LogiDeduct | AR-LSAT   | FOLIO     | Chinese LogiQA-V2 |
> | ------------------------ | ----------- | --------- | ---------- | --------- | --------- | ---------------- |
> | Direct Answer            | 40.17       | 55.20     | 42.67      | 24.24     | 32.35     | 62.96             |
> | CoT                      | 40.50       | 75.60     | 40.33      | 29.87     | 53.24     | 59.32             |
> | LogicLM                  | 61.63       | 73.80     | 63.00      | 14.29     | 59.61     | 25.33             |
> | SymbCoT                  | 70.17       | 80.60     | 60.00      | 32.03     | 57.39     | 65.44             |
> | CortexDebate (w/o NL-SL) | 47.50       | 78.20     | 39.67      | 19.91     | 53.92     | 62.96             |
> | CortexDebate (w/ NL-SL)  | 75.83       | 84.00     | 67.00      | 35.93     | 63.73     | 66.92             |
> | Ours (w/o sparse)        | 74.67       | 85.20     | 68.33      | 35.06     | 62.24     | 67.98             |
> | **Ours (w/ sparse)**     | **76.50**   | **86.40** | **67.67**  | **37.20** | **65.68** | **68.11**         |
> ||
>
> **Table R1.2: Performance of different methods on GPT-4o-mini.**
>
> | Method                   | ProofWriter | ProntoQA  | LogiDeduct | AR-LSAT   | FOLIO     | Chinese LogiQA-V2 |
> | ------------------------ | ----------- | --------- | ---------- | --------- | --------- | ----------------- |
> | Direct Answer            | 53.50       | 62.60     | 56.00      | 19.91     | 59.80     | 57.31             |
> | CoT                      | 43.67       | 75.00     | 70.33      | 19.04     | 61.76     | 54.30             |
> | LogicLM                  | 58.67       | 76.00     | 73.00      | 22.94     | 34.80     | 25.00             |
> | SymbCoT                  | 70.33       | 82.80     | 75.33      | 26.00     | 69.10     | 58.87             |
> | CortexDebate (w/o NL-SL) | 60.87       | 80.40     | 70.67      | 21.21     | 62.75     | 62.02             |
> | CortexDebate (w/ NL-SL)  | 75.17       | 89.00     | 82.33      | **34.20** | 75.00     | 64.03             |
> | Ours (w/o sparse)        | 74.00       | 89.40     | 82.33      | 33.81     | 74.02     | 65.91             |
> | **Ours (w/ sparse)**     | **76.33**   | **90.60** | **84.67**  | **34.20** | **76.47** | **67.29**         |
> ||
>
> **Table R1.3: Performance of different methods on GPT-4.**
>
> | Method                   | ProofWriter | ProntoQA    | LogiDeduct | AR-LSAT   | FOLIO      | Chinese LogiQA-V2 |
> | ------------------------ | ----------- | ----------- | ---------- | --------- | ---------- | ----------------- |
> | Direct Answer            | 53.50      | 75.40      | 59.00     | 32.90     | 65.20      | 62.27            |
> | CoT                      | 67.17      | 81.20      | 69.67     | 35.06     | 70.59      | 65.22            |
> | LogicLM                  | 79.17      | 93.40      | 87.00     | 40.86     | 76.96      | 25.99            |
> | SymbCoT                  | 82.33      | 96.00      | 86.33     | 42.86     | 80.39      | 70.57            |
> | CortexDebate (w/o NL-SL) | 73.50      | 89.00      | 84.33     | 34.20     | 75.00     | 68.17            |
> | CortexDebate (w/ NL-SL)  | 90.83      | 99.60      | 92.33     | 51.08     | 84.80     | 74.13            |
> | Ours (w/o sparse)        | 90.17      | 99.40      | 94.00     | 50.42     | 84.31     | 74.01            |
> | **Ours (w/ sparse)**     | **92.00**  | **100.00** | **94.33** | **53.25** | **86.27** | **74.76**        |
> ||
>
> **Table R1.4: Performance of different methods on DeepSeek-V3.**
>
> | Method                   | ProofWriter | ProntoQA   | LogiDeduct | AR-LSAT   | FOLIO     | Chinese LogiQA-V2 |
> | ------------------------ | ----------- | ---------- | ---------- | --------- | --------- | ----------------- |
> | Direct Answer            | 68.33       | 79.20      | 85.33      | 36.80     | 66.18     | 74.33             |
> | CoT                      | 71.83       | 85.00      | 83.00      | 45.45     | 76.96     | 77.97             |
> | LogicLM                  | 80.50       | 83.20      | 93.33      | 43.72     | 78.92     | 28.63             |
> | SymbCoT                  | 85.83       | 98.00      | 94.00      | 47.02     | 81.37     | 81.98             |
> | CortexDebate (w/o NL-SL) | 83.17       | 94.00      | 90.00      | 44.16     | 79.90     | 78.97             |
> | CortexDebate (w/ NL-SL)  | 93.00       | 99.80      | 99.67      | 74.03     | 88.73     | 83.04             |
> | Ours (w/o sparse)        | 92.83       | 99.80      | **100.00** | 73.62     | 89.22     | 85.68             |
> | **Ours (w/ sparse)**     | **93.33**   | **100.00** | **100.00** | **75.76** | **90.67** | **86.93**         |
> ||
> ---
>
> Again, thank you very much for your suggestions for improvements! We are eager to hear your feedback! We’d deeply appreciate it if you could let us know whether your concerns have been addressed.

---

> ### Author Response · Authors · 2025-11-27
> **(to Q1) Analytical and empirical evidence of the effectiveness of LLM-generated confidence scores in our approach**
>
> Thank you very much for engaging with our rebuttal and the great follow-up questions!
>
> > Q1. Are the confidence scores generated by the LLM itself reliable?
>
> Thanks to your question, we have included in our manuscript the following analytical and empirical results.
>
> - FYI, our design of confidence scores follows a variety of multi-agent works, where **LLMs generate an explicit confidence score that is then used for confidence-weighted voting or debate control**, e.g., ReConcile [1], ConsensAgent [2], ConfMAD [3], Self-eval [4], VerbalConf [5].
> - In addition, a prior work [6] shows that although LLM self-reported confidence is not perfectly accurate (sometimes over‑confident), it has **strong pair-rank performance —i.e., it can reliably distinguishes which of two candidate answers is more likely to be correct**. Our sparse gate uses self-reported confidence **only as a relative ranking signal, via ratios such as $C_i / C_j$**, not as an absolute probability. This design is exactly the use-case where self-reported confidence is empirically reliable.
> - Empirically, we added an ablation where we exclude the confidence term. We find that LLM-generated confidence scores provides a stably better signal for the sparse communication selection in our multi-agent approach.
>
>   | Model | Variant   | ProofWriter | FOLIO  | Chinese LogiQA-V2 |
>   |--|----------:|------------:|-------:|----:|
>   | Qwen2.5-7B-Instruct| w/o conf  | 75.33       | 64.23  | 67.92 |
>   | Qwen2.5-7B-Instruct| w/ conf   | **76.50**       | **65.68**  | **68.11** |
>   | GPT-4 | w/o conf  | 90.87       | 84.80  | 74.26|
>   | GPT-4 | w/ conf   | **92.00**       | **86.27**  | **74.76** |
>
> We have incorporate the above results to Appendix J in our revised PDF highlighted in "$\textcolor{blue}{blue}$".
>
> ***
>
> ### **What makes our work unique and important rather than LLM confidence?**
>
> We would say, our main idea is **not** extending well-known patterns (e.g., **LLM confidence;** multi-agent debate; sparse topologies like SparseMAD/CortexDebate), but **focusing on the shortcomings of current LLM logical reasoning approaches and tackling these issues**. And we have carefully rewrite a few contribution claim in "Introduction" section highlighted in "$ \textcolor{blue}{blue} $".
>
> In summary, **we try to improve both (i) symbolic translation stage and (ii) neuro-symbolic reasoning stage by noting that different symbolic languages and reasoning approaches can effectively complement each other's shortcomings.** Please kindly note that **this observation and solution is specific to LLM logical reasoning**, which is the main contribution of this paper.
>
> Specifically, current LLM reasoning methods can generally be **summarized as two stages: the translation stage and the neuro-symbolic reasoning stage:**
>
> - In the first stage of **symbolic translation**, prior work translates a natural language (NL) problem into a specific symbolic language (SL) such as LP, FOL, SAT, etc. However, each SL has its own advantages and disadvantages. Therefore, we employ multi-agent debate to **exploit multiple SL to correct the translation errors through mutual refinement, ultimately enhancing the accuracy of the translation stage.**
>   - **Prior works:** only pre-define a specific SL, and translate the NL to that SL via prompting;
>   - **Motivation and our contribution:** by noting that different SL can capture different important features of raw NL, and they can benefit each other if we translate a same NL to multiple SL. That is, we find these SL can benefit each other translation through mutual refinement.
> - In the second stage of **neuro-symbolic reasoning**, methods are typically categorized into two approaches: reasoning via SL solver and reasoning via LLM.
>   - **Prior works:** either perform neuro-symbolic reasoning via SL solver or via LLM;
>   - **Motivation and our contribution:** for **SL solver**, though they can enable rigorous reasoning, they will get wrong results (or even be unable to run the solver) when the translation process is imperfect, i.e., **strong reasoning, weak robustness**. For neuro-symbolic reasoning via **LLM**, it can tolerate for inaccurate translation, but there will be hallucinations in reasoning, i.e., **strong robustness, weak reasoning**. Thus we introduce multi-agent debate to synthesize the strengths of both approaches, achieving optimal performance in the reasoning stage.
>
> In a nutshell, we are the first to recognize these two challenges in LLM logical reasoning community and propose an extended multi-agent framework to leverage the strengths of different symbolic languages and reasoning methods, achieving better performance in both translation and reasoning stages. The core focus lies in **why we need to combine different languages for SL translation** and **why we need perform neuro-symbolic reasoning via both the SL solver and the LLM**, rather than the LLM confidence or multi-agent debate or sparse topologies itself.

---

> ### Author Response · Authors · 2025-11-27
> **(to Q2) All changes have been fully incorporated into our uploaded revised version**
>
> > Q2. Given the amount of experimentation the author has added, I worry whether these changes will be fully incorporated into the final version
>
> Thanks for your kind reminder and we understand this is an important concern. Also thanks to ICLR allowing us to upload the revised PDF - we have incorporated *all* changes, which highlighted in "$\textcolor{blue}{blue}$" for facilitating checking.
>
> FYI, we made the following changes:
>
> - In response to ```W1``` and ```Q1```, we extend our evaluation to real-world datasets (FOLIO, AR-LSAT, Chinese LogiQA-V2) in Table 2.
> - In response to ```W2```, we follow previous works to set *temperature* as 0 (so there is no randomness and error bars) in our original manuscript, and follow your suggestion to report error bars using 3 prompt paraphrases to incorporate randomness and perform significance testing in Table 4.
> - In response to ```W3```, we include smaller/ordinary base models using both Qwen2.5-7B-Instruct and GPT-4o-mini in Table 3 and Table 19 to demonstrate the effectiveness of our method under smaller models.
> - In response to ```Q2```, we analytically and empirically show the effectiveness of LLM-generated confidence scores in our approach in Appendix J.
>
> We kindly notice that you wrote ```My opinion may be negative at this stage, if the author can address my concerns, I will increase my score```. We sincerely hope the reviewer will re-evaluate our paper based on the additional experiments and explanations provided above. We look forward to your feedback if you have any further questions!
>
> Thanks for your time,
>
> Authors of #14334
>
> ***
>
> **References**
>
> [1] Chen Justin, et al. “Reconcile: Round-table conference improves reasoning via consensus among diverse llms.” ACL 2024.
>
> [2] Pitre Priya, et al. “Consensagent: Towards efficient and effective consensus in multi-agent llm interactions through sycophancy mitigation.” ACL 2025.
>
> [3] Lin Zijie, et al. "Enhancing Multi-Agent Debate System Performance via Confidence Expression." ACL 2025.
>
> [4] Ren Jie, et al. "Self-evaluation improves selective generation in large language models." NeurIPS 2023 Workshop.
>
> [5] Yang Daniel, et al. "On verbalized confidence scores for LLMs." arXiv preprint 2024.
>
> [6] Xiong Miao, et al. “Can llms express their uncertainty? an empirical evaluation of confidence elicitation in llms.” ICLR 2024.

---

> > ### Comment · Reviewer_awiG · 2025-11-28
> > **Further reply to the rebuttal**
> >
> > Thanks for the author's reply, I appreciate your efforts.
> >
> > I've decided to raise my overall assessment score to 6.

---

### Official Review · Reviewer_UAB7 · 2025-10-29

**Soundness:** 3
**Presentation:** 3
**Contribution:** 3
**Rating:** 6
**Confidence:** 4

**Summary:**

The authors introduce a multi-agentic debate system to elicit logical reasoning in LLMs, by integrating symbolic and natural language reasoning. They do this through translating natural language to symbolic language using multiple agents, with the refinement process done through agentic debate. For reasoning, the symbolic output from the translation stage and natural language are used to determine an output through majority voting. To enhance computational efficiency, the authors use a sparse communication strategy within the debate framework.

**Strengths:**

- The use of general purpose (i.e. GPT) and reasoning-enhanced (i.e. deepseek) LLMs helps the reader understand the generalizability of the proposed approach.
- The results are impressive, as they achieve 100% across several set-ups.
- The number of baselines used to compare to the proposed approach is sufficient, showcasing the true performance capability of the framework.
- The use of an adaptive sparse communication strategy to enhance computational efficiency is a very important advantage of this framework.

**Weaknesses:**

- The whole pipeline depends on the translation quality in the first stage. There is some missing validation of the symbolic translations to enhance the validity of the pipeline.
- The majority-vote process could highlight systematic biases if multiple agents share similar pretraining biases or translation tendencies.

**Questions:**

See above.

---

> ### Author Response · Authors · 2025-11-24
>
> Thank you very much for your valuable reviews and comments! We have carefully addressed each of the concerns as follows:
>
> > W1: "The whole pipeline depends on the translation quality in the first stage. There is some missing validation of the symbolic translations to enhance the validity of the pipeline."
>
> **Response:** Thank you for raising this thoughtful concern. We have now added two complementary forms of validation.
>
> - **Failing rate of SL translations.** We quantify SL translation failures in **Table R1**. For each of the three SL agents (LP/FOL/SAT) and each question, we mark the translation as correct or incorrect, and report the rate that **n** specific SL agents are all wrong (T-CER_n). Thus T-CER_1 is the average single-agent translation error rate, and **T-CER_3 is the probability that all three SL agents fail on the same example.** As shown in Table R1, the probability that all three SL translators fail simultaneously (T-CER_3) is very small.
> - **Direct validation against gold SL on FOILIO.** To further validate the correctness of our translations, we evaluate on **FOILIO**, one of the few datasets with human-annotated FOL formulas. For each example we compare our FOL translation with the gold FOL using an LLM judge to decide whether they are semantically equivalent. As shown in **Table R2**, **the translation-debate stage consistently improves FOL translation accuracy**.
>
> **Table R1. Translation Common Error Rate (T-CER\_n) for 3 SL agents (LP/FOL/SAT).**
>
> | #SL agents | GPT-4 ProofWriter | GPT-4 ProntoQA | GPT-4 LogiDeduct | DeepSeek ProofWriter | DeepSeek ProntoQA | DeepSeek LogiDeduct |
> | ---------- | ----------------- | -------------- | ---------------- | -------------------- | ----------------- | ------------------- |
> | **1**      | 0.185000          | 0.103333       | 0.445556         | 0.176111             | 0.145333          | 0.535556            |
> | **2**      | 0.025000          | 0.012000       | 0.193333         | 0.014444             | 0.028667          | 0.137778            |
> | **3**      | 0.003333          | 0.002000       | 0.040000         | 0.001667             | 0.010000          | 0.056667            |
> ||
>
>
> **Table R2. FOL translation quality on FOILIO.**
> Numbers are LLM-judged semantic correctness of the FOL translations (%).
>
> | Model       | w/o translation debate | w/ translation debate |
> | ----------- | ---------------------- | --------------------- |
> | GPT-4o-mini | 68.14                  | 75.49                 |
> | GPT-4       | 76.47                  | 84.80                 |
> | DeepSeek-V3 | 75.98                  | 88.24                 |
> ||

---

> ### Author Response · Authors · 2025-11-24
>
> > W2: "The majority-vote process could highlight systematic biases if multiple agents share similar pretraining biases or translation tendencies."
>
> **Response:** Thank you for raising this insightful concern! We address it both **theoretically** and **empirically**, and also provide a **robustness study across voting methods**.
>
> - **Theoretical Analysis of a Lower Bound on Majority Vote Accuracy Under Correlated Agent Errors**. We **add a multiclass ensemble theoretical analysis in our revised paper**. We consider a setting with $k$ classes and $m$ agents $\{h_i\}_{i=1}^m$, where each agent is better than random, i.e., $\mathbb{P}(h_i(x)=y)=p>1/k$ , and errors are uniformly distributed over the $k-1$ wrong labels. For a fixed incorrect class $c\neq y$, we define $$T\_i = \mathbf{1}[h_i(x)=y] - \mathbf{1}[h_i(x)=c]$$ and assume $\mathrm{Var}(T\_i)=\sigma^2$ and $\mathrm{Cov}(T_i,T_j)=\rho\sigma^2$ for $i\neq j$, where $\rho\in[0,1)$ is the average pairwise class-wise correlation. Let $\delta = p - \frac{1-p}{k-1}>0$. Using Chebyshev’s inequality plus a union bound over the $k-1$ wrong classes, we obtain the following lower accuracy bound for the majority-vote ensemble $H$: $$\mathbb{P}(H(x)=y)\;\ge\;1 - (k-1)\cdot\frac{\sigma^2\,[1+(m-1)\rho]}{m\,\delta^2}.$$ Two corollaries that are relevant here:
>
>   - **(i) If errors are independent $\rho=0$ , the lower bound goes to 1 as $m\to\infty$.**
>   - **(ii) If errors are positively but moderately correlated $\rho>0$, the bound converges to $1 - (k-1)\frac{\rho\sigma^2}{\delta^2}$ as $m\to\infty$** ,
>
>   showing that majority vote remains well-behaved unless agents are highly correlated. This formalizes a key intuition in our system: **since our agents come from distinct SL/NL reasoning paradigms, their error correlation is substantially below the regime that leads to the failure mode of high spurious agreement.**
>
> - **Empirical quantification of shared mistakes.** To measure correlated errors directly, we introduce the **Common Error Rate** $\mathrm{CER}\_n$. For any subset $S$ of $n$ agents, let the answer given by agent $a$ be $y_{a,q}$, the indicator of correctness of answer $y_{a,q}$ be $c_{a,q}\in\{0,1\}$, $$\mathrm{CER}\_S = \frac{1}{|Q|}\big|\{q\in Q:\ \forall a\in S,\ c\_{a,q}=0\ \text{and}\ y\_{a,q}\ \text{are identical}\}\big|$$ i.e., the fraction of questions on which all agents in $S$ pick the same wrong option, and we average over all $\binom{5}{n}$ subsets to obtain $\mathrm{CER}_n$. Across all three datasets and GPT-4 / DeepSeek-V3 **(Table R3)**:
>
>   - $\mathrm{CER}_n$ drops rapidly as $n$ increases.
>   - Our sparse gating further largely reduces these correlated mistakes.
>
>   This shows that **“many agents has similar bias” is already rare**, and our sparse gating mechanism further reduces such shared mistakes.
>
> - **Robustness to voting methods.** We added a sensitivity study on three aggregation rules **(Table R4)**: (i) Majority vote (ii) Confidence-weighted vote (iii) “LLM-as-judge” (agents debate, and an independent LLM reads all rationales and produces final prediction). The gap between all three methods is **≤ 1 percentage point**. This indicates that:
>
>   - Our improvements are not due to a particular voting method.
>   - The gains primarily come from the **SL+NL multi-agent debate and sparse communication**, while the final voter is easily changebale.

---

> ### Author Response · Authors · 2025-11-24
>
> **Table R3. Common Error Rate (CER_n) across agent subsets (lower is better).**
>
> | #Agents | Variant    | GPT-4 ProofWriter | GPT-4 ProntoQA | GPT-4 LogiDeduct | DeepSeek ProofWriter | DeepSeek ProntoQA | DeepSeek LogiDeduct |
> | ------- | ---------- | ----------------- | -------------- | ---------------- | -------------------- | ----------------- | ------------------- |
> | **1**   | w/o sparse | 0.101667          | 0.002400       | 0.070000         | 0.259333             | 0.001600          | 0.006000            |
> |         | w/ sparse  | 0.106000          | 0.001600       | 0.085463         | 0.247500             | 0.000800          | 0.004667            |
> | **2**   | w/o sparse | 0.090000          | 0.002000       | 0.055534         | 0.095500             | 0.000200          | 0.003333            |
> |         | w/ sparse  | 0.050333          | 0.000200       | 0.044636         | 0.064333             | 0.000000          | 0.000333            |
> | **3**   | w/o sparse | 0.065333          | 0.002000       | 0.042621         | 0.053667             | 0.000000          | 0.003333            |
> |         | w/ sparse  | 0.018333          | 0.000000       | 0.021249         | 0.020667             | 0.000000          | 0.000000            |
> | **4**   | w/o sparse | 0.042500          | 0.002000       | 0.040194         | 0.036667             | 0.000000          | 0.003333            |
> |         | w/ sparse  | 0.011667          | 0.000000       | 0.010142         | 0.012500             | 0.000000          | 0.000000            |
> | **5**   | w/o sparse | 0.020000          | 0.002000       | 0.038252         | 0.028333             | 0.000000          | 0.003333            |
> |         | w/ sparse  | 0.006667          | 0.000000       | 0.009681         | 0.009500             | 0.000000          | 0.000000            |
> ||
>
> **Table R4. Robustness to voting methods (Accuracy %, higher is better).**
>
> | Model           | Dataset     | Majority Vote | Conf-Weighted | LLM-as-Judge |
> | --------------- | ----------- | ------------- | ------------- | ------------ |
> | **GPT-4o-mini** | ProofWriter | 76.33%        | 75.50%        | 76.00%       |
> |                 | ProntoQA    | 89.60%        | 88.00%        | 90.40%       |
> |                 | LogiDeduct  | 82.33%        | 82.67%        | 83.33%       |
> | **GPT-4**       | ProofWriter | 92.00%        | 91.60%        | 92.00%       |
> |                 | ProntoQA    | 100.00%       | 100.00%       | 100.00%      |
> |                 | LogiDeduct  | 94.33%        | 93.67%        | 95.00%       |
> | **DeepSeek-V3** | ProofWriter | 93.33%        | 93.50%        | 93.50%       |
> |                 | ProntoQA    | 100.00%       | 100.00%       | 100.00%      |
> |                 | LogiDeduct  | 100.00%       | 99.80%        | 100.00%      |
> ||
>
>
> ------
>
> We are eager to hear your feedback! We’d deeply appreciate it if you could let us know whether your concerns have been addressed.

---

> > ### Author Response · Authors · 2025-11-27
> >
> > Dear reviewer UAB7,
> >
> > Thank you once again for your detailed review and the valuable time you have dedicated to our work! We have uploaded our revised PDF incorporating *all* mentioned changes, which highlighted in "$\textcolor{blue}{blue}$" for facilitating checking.
> >
> > FYI, we made the following changes:
> >
> > - In response to ```W1```, we **provide detailed translation-quality analyses via Translation Common Error Rate (T-CER_n) and direct validation against gold FOL formulas on FOLIO** in Section 5.5 (Tables 8–9).
> >
> > - In response to ```W2```, we **add a sensitivity analysis on aggregation rules** for majority vote in Table 18.
> >
> > - In response to ```W2```, we **provide empirical quantification of shared mistakes** in Table 21, showing that “many agents choosing the same wrong label” is rare and that our sparse gate further reduces shared spurious agreements.
> >
> > - In response to ```W2```, we **add a formal theoretical lower-bound analysis of majority-vote accuracy under heterogeneous SL/NL agents** in Section 4, with detailed proofs in Appendix E.
> >
> > As the discussion deadline approaches, we are wondering whether our responses have properly addressed your concerns? Your feedback would be extremely helpful to us. If you have further comments or questions, we hope for the opportunity to respond to them.
> >
> > Thanks for your time,
> >
> > Authors of #14334

---

> > > ### Comment · Reviewer_UAB7 · 2025-11-27
> > >
> > > Thank you for the clarifications and the additional information. I appreciate the authors’ effort in addressing the points raised in my review. After considering the clarification, my overall assessment and score remain unchanged.

---

> ### Author Response · Authors · 2025-11-28
> **Thank you for standing on the positive side of our paper acceptance!**
>
> Thank you for being positive toward our paper acceptance, and recognizing the Soundness, Presentation, and Contribution  of our paper! We really appreciate your valuable suggestions which have undoubtedly contributed to improving the quality of our paper.

---

### Official Review · Reviewer_DhnF · 2025-10-31

**Soundness:** 3
**Presentation:** 3
**Contribution:** 2
**Rating:** 4
**Confidence:** 3

**Summary:**

The paper proposes a two-stage, sparse multi-agent debate framework to improve complex logical reasoning with LLMs by explicitly combining symbolic-language (SL) and natural-language (NL) paradigms: multiple agents first translate an NL problem into diverse SL formalisms (LP, FOL, SAT) and refine these translations via debate; then SL-solver agents (Pyke/Prover9/Z3 results verbalized) and NL-reasoning agents (e.g., CoT, Plan-and-Solve) debate for several rounds before a majority vote determines the answer. To curb the high cost of multi-agent discussion, the authors introduce an adaptive sparse communication mechanism that gates interactions using a preference score mixing agent confidence and information gain (cosine dissimilarity), coupled with selective memory updates. Experiments on ProntoQA, ProofWriter (depth-5 subset), and BIG-Bench LogicalDeduction across GPT-4, Claude 3.7 Sonnet, and DeepSeek-V3 report new SOTA accuracies while reducing token usage compared to strong single- and multi-agent baselines, with ablations showing gains from both the translation-stage debate and heterogeneous SL+NL agent composition.

**Strengths:**

- Novel combination of symbolic and natural-language reasoning in a multi-agent debate.
- Sparse-communication mechanism effectively reduces token cost with minimal accuracy loss.
- Strong empirical performance and ablation studies across several reasoning benchmarks.

**Weaknesses:**

- The paper’s two main ideas extend well-known patterns (multi-agent debate; sparse topologies like SparseMAD/CortexDebate) rather than fundamentally rethinking them; the conceptual gap from past methods looks narrow.
- No theoretical framework proving that the proposed agents can perform valid NL to FOL/LP/SAT translation.
- With heterogeneous agents, simple majority voting can entrench shared biases or spurious agreement.
- Critical components (e.g., $C_i^d$, Algorithm 1) are under-defined or inconsistent.
- Lack of significance testing and limited evaluation scope on real-world reasoning tasks.

**Questions:**

Please concretely differentiate your contributions from recent multi-agent debate and sparse/topology papers (methodologically, not just empirically). Could you add an ablation that (a) removes the SL-NL cross-paradigm stage and (b) replaces your sparse gating with a strongest prior baseline, to quantify each idea’s standalone lift?

How often do translation mistakes occur, and how does the system recover when all SL agents share the same systematic error?

Please include sensitivity analyses for λ and the similarity metric, and compare majority vote against confidence-weighted or learned adjudicators.

---

> ### Author Response · Authors · 2025-11-24
>
> Thank you for your detailed review and valuable suggestions! We address your questions point-to-point in the following.
>
> > W1: "The paper’s two main ideas extend well-known patterns (multi-agent debate; sparse topologies like SparseMAD/CortexDebate) rather than fundamentally rethinking them; the conceptual gap from past methods looks narrow."
>
> **Response:** Thanks for your comments, and we understand this is an important concern. However, our main idea is not extending well-known patterns (multi-agent debate; sparse topologies like SparseMAD/CortexDebate), but **focusing on the shortcomings of current LLM logical reasoning approaches and tackling these issues**.
>
> In summary, **we try to improve both (i) symbolic translation stage and (ii) neuro-symbolic reasoning stage by noting that different symbolic languages and reasoning approaches can effectively complement each other's shortcomings.** Please kindly note that **this observation and solution is specific to LLM logical reasoning**, which is the main contribution of this paper.
>
> Specifically, current LLM reasoning methods can generally be **summarized as two stages: the translation stage and the neuro-symbolic reasoning stage:**
>
> - In the first stage of **symbolic translation**, prior work translates a natural language (NL) problem into a specific symbolic language (SL) (LP, FOL, SAT, etc.). However, each SL has its own advantages and disadvantages. Therefore, we employ multi-agent debate to **exploit multiple SL to correct the translation errors through mutual refinement, ultimately enhancing the accuracy of the translation stage.**
>   - **Prior works:** only pre-define a specific SL, and translate the NL to that SL via prompting;
>   - **Motivation and our contribution:** by noting that different SL can capture different important features of raw NL, and they can benefit each other if we translate a same NL to multiple SL. That is, we find these SL can benefit each other translation through mutual refinement.
> - In the second stage of **neuro-symbolic reasoning**, methods are typically categorized into two approaches: reasoning via SL solver and reasoning via LLM.
>   - **Prior works:** either perform neuro-symbolic reasoning via SL solver or via LLM;
>   - **Motivation and our contribution:** for **SL solver**, though they can enable rigorous reasoning, they will get wrong results (or even be unable to run the solver) when the translation process is imperfect, i.e., **strong reasoning, weak robustness**. For neuro-symbolic reasoning via **LLM**, it can tolerate for inaccurate translation, but there will be hallucinations in reasoning, i.e., **strong robustness, weak reasoning**. Thus we introduce multi-agent debate to synthesize the strengths of both approaches, achieving optimal performance in the reasoning stage.
>
> In a nutshell, we are the first to recognize these two challenges in LLM logical reasoning community and propose an extended multi-agent framework to leverage the strengths of different symbolic languages and reasoning methods, achieving better performance in both translation and reasoning stages. The core focus lies in **why we need to combine different languages for SL translation** and **why we need perform neuro-symbolic reasoning via both the SL solver and the LLM**, rather than the multi-agent debate or sparse topologies itself.
>
> > W2: "No theoretical framework proving that the proposed agents can perform valid NL to FOL/LP/SAT translation."
>
> **Response:** Thank you for the concern. We agree that no existing work—including ours—can theoretically prove that a black-box LLM can performs valid NL→FOL/LP/SAT translation. This limitation is shared by all recent neuro-symbolic systems. However, **many prior evidence shows that modern LLMs (GPT-4/Claude) can reliably generate executable symbolic forms without any fine-tuning**, as demonstrated in LogicLM [1], CLOVER [2], Aristotle [3], Adaptive Translation [4].
>
> To supplement these findings, we also report **direct NL→SL (LP/FOL/SAT) translation accuracy** of GPT-4 and DeepSeek-V3 across three datasets **(Table R1)**. These results show that:
>
> 1. **LLMs can achieve well symbolic translation accuracy**, consistent with prior work.
> 2. **Different SL languages succeed/fail on different questions**, motivating our use of multiple SL translators and NL agents in a unified debate.
>
> **Table R1. Direct NL→LP/FOL/SAT translation accuracy (%) without debate.**
>
> | Model       | SL   | ProntoQA | ProofWriter | LogicalDeduction |
> | ----------- | ---- | -------- | ----------- | ---------------- |
> | GPT-4       | LP   | 94.00%   | 78.83%      | 58.33%           |
> | GPT-4       | FOL  | 94.20%   | 62.33%      | 20.00%           |
> | GPT-4       | SAT  | 63.6%    | 65.00%      | 94.67%           |
> | DeepSeek-V3 | LP   | 94.60%   | 93.00%      | 56.33%           |
> | DeepSeek-V3 | FOL  | 99.40%   | 93.50%      | 26.00%           |
> | DeepSeek-V3 | SAT  | 82.80%   | 65.33%      | 90.67%           |
> ||

---

> ### Author Response · Authors · 2025-11-24
>
> > W3: "With heterogeneous agents, simple majority voting can entrench shared biases or spurious agreement."
>
> **Response:** Thank you for raising this insightful concern! We address it both **theoretically** and **empirically**, and also provide a **robustness study across aggregation methods**.
>
> - **Theoretical Analysis of a Lower Bound on Majority Vote Accuracy Under Correlated Agent Errors**. We **add a multiclass ensemble theoretical analysis in our revised paper**. We consider a setting with $k$ classes and $m$ agents $\{h_i\}_{i=1}^m$, where each agent is better than random, i.e., $\mathbb{P}(h_i(x)=y)=p>1/k$ , and errors are uniformly distributed over the $k-1$ wrong labels. For a fixed incorrect class $c\neq y$, we define $$T_i = \mathbf{1}[h_i(x)=y] - \mathbf{1}[h_i(x)=c]$$ and assume $\mathrm{Var}(T_i)=\sigma^2$ and $\mathrm{Cov}(T_i,T_j)=\rho\sigma^2$ for $i\neq j$, where $\rho\in[0,1)$ is the average pairwise class-wise correlation. Let $\delta = p - \frac{1-p}{k-1}>0$. Using Chebyshev’s inequality plus a union bound over the $k-1$ wrong classes, we obtain the following lower accuracy bound for the majority-vote ensemble $H$: $$\mathbb{P}(H(x)=y)\;\ge\;1 - (k-1)\cdot\frac{\sigma^2\,[1+(m-1)\rho]}{m\,\delta^2}.$$ Two corollaries that are relevant here:
>
>   - **(i) If errors are independent $\rho=0$ , the lower bound goes to 1 as $m\to\infty$.**
>   - **(ii) If errors are positively but moderately correlated $\rho>0$, the bound converges to $1 - (k-1)\frac{\rho\sigma^2}{\delta^2}$ as $m\to\infty$** ,
>
>   showing that majority vote remains well-behaved unless agents are highly correlated. This formalizes a key intuition in our system: **since our agents come from distinct SL/NL reasoning paradigms, their error correlation is substantially below the regime** **that leads to the failure mode of high spurious agreement.**
>
> - **Empirical quantification of shared mistakes.** To measure correlated errors directly, we introduce the **Common Error Rate** $\mathrm{CER}\_n$. For any subset $S$ of $n$ agents, let the answer given by agent $a$ be $y_{a,q}$, the indicator of correctness of answer $y_{a,q}$ be $c_{a,q}\in\{0,1\}$, $$\mathrm{CER}\_S = \frac{1}{|Q|}\big|\{q\in Q:\ \forall a\in S,\ c\_{a,q}=0\ \text{and}\ y\_{a,q}\ \text{are identical}\}\big|$$ i.e., the fraction of questions on which all agents in $S$ pick the same wrong option, and we average over all $\binom{5}{n}$ subsets to obtain $\mathrm{CER}_n$. Across all three datasets and GPT-4 / DeepSeek-V3 **(Table R2)**:
>
>   - $\mathrm{CER}_n$ drops rapidly as $n$ increases.
>   - Our sparse gating further largely reduces these correlated mistakes.
>
>   This shows that **“many agents choosing the same wrong label” is already rare**, and our sparse gating mechanism further reduces such shared mistakes.
>
> - **Robustness to aggregation rules.** To show that our results are not due to a specific voting rule, we added a sensitivity study on three aggregation rules **(Table R3)**: (i) Majority vote (ii) Confidence-weighted vote (iii) “LLM-as-judge” (agents debate, and an independent LLM reads all rationales and produces final prediction). Across **3 base LLMs × 3 datasets**, the gap between all three methods is **≤ 1 percentage point**. This indicates that:
>
>   - Our improvements are not due to a particular voting method.
>   - The gains primarily come from the **SL+NL multi-agent debate and sparse communication**, while the final aggregator is easily changebale.
>
>
>
> > W4: "Critical components (e.g., $C_i^{d}$, Algorithm 1) are under-defined or inconsistent."
>
> **Response:** Sorry for the unclear statement. We now clarify that $C_i^{d}$ is the self-reported confidence generated by each LLM agent in the same response turn as its predicted answer. Concretely, every agent is prompted to output: (i) its predicted label, (ii) the reasoning trace, and (iii) a scalar confidence value in [0,1]. We have also revised the paper to make the definitions explicit and eliminate potential ambiguity.

---

> ### Author Response · Authors · 2025-11-24
>
> **Table R2. Common Error Rate (CER_n) across agent subsets (lower is better).**
>
> | #Agents | Variant    | GPT-4 ProofWriter | GPT-4 ProntoQA | GPT-4 LogiDeduct | DeepSeek ProofWriter | DeepSeek ProntoQA | DeepSeek LogiDeduct |
> | ------- | ---------- | ----------------- | -------------- | ---------------- | -------------------- | ----------------- | ------------------- |
> | **1**   | w/o sparse | 0.101667          | 0.002400       | 0.070000         | 0.259333             | 0.001600          | 0.006000            |
> |         | w/ sparse  | 0.106000          | 0.001600       | 0.085463         | 0.247500             | 0.000800          | 0.004667            |
> | **2**   | w/o sparse | 0.090000          | 0.002000       | 0.055534         | 0.095500             | 0.000200          | 0.003333            |
> |         | w/ sparse  | 0.050333          | 0.000200       | 0.044636         | 0.064333             | 0.000000          | 0.000333            |
> | **3**   | w/o sparse | 0.065333          | 0.002000       | 0.042621         | 0.053667             | 0.000000          | 0.003333            |
> |         | w/ sparse  | 0.018333          | 0.000000       | 0.021249         | 0.020667             | 0.000000          | 0.000000            |
> | **4**   | w/o sparse | 0.042500          | 0.002000       | 0.040194         | 0.036667             | 0.000000          | 0.003333            |
> |         | w/ sparse  | 0.011667          | 0.000000       | 0.010142         | 0.012500             | 0.000000          | 0.000000            |
> | **5**   | w/o sparse | 0.020000          | 0.002000       | 0.038252         | 0.028333             | 0.000000          | 0.003333            |
> |         | w/ sparse  | 0.006667          | 0.000000       | 0.009681         | 0.009500             | 0.000000          | 0.000000            |
> ||
>
>
>
> **Table R3. Robustness to aggregation methods (Accuracy %, higher is better).**
>
> | Model           | Dataset     | Majority Vote | Conf-Weighted | LLM-as-Judge |
> | --------------- | ----------- | ------------- | ------------- | ------------ |
> | **GPT-4o-mini** | ProofWriter | 76.33%        | 75.50%        | 76.00%       |
> |                 | ProntoQA    | 89.60%        | 88.00%        | 90.40%       |
> |                 | LogiDeduct  | 82.33%        | 82.67%        | 83.33%       |
> | **GPT-4**       | ProofWriter | 92.00%        | 91.60%        | 92.00%       |
> |                 | ProntoQA    | 100.00%       | 100.00%       | 100.00%      |
> |                 | LogiDeduct  | 94.33%        | 93.67%        | 95.00%       |
> | **DeepSeek-V3** | ProofWriter | 93.33%        | 93.50%        | 93.50%       |
> |                 | ProntoQA    | 100.00%       | 100.00%       | 100.00%      |
> |                 | LogiDeduct  | 100.00%       | 99.80%        | 100.00%      |
> ||
>
>
> > W5: "Lack of significance testing and limited evaluation scope on real-world reasoning tasks"
>
> **Response:** Thank you for the valuable suggestion! We have added:
>
> - **Significance testing.** Our original setting used temperature = 0 for deterministic reproducibility. Now we repeated main experiments using **three semantically equivalent prompt rewrites** (via GPT-5) to introduce controlled variance. We now report **mean ± std** **(Table R4)**, with several improvements statistically significant to the most competitive baseline (marked *).
> - **Real-world reasoning datasets.** To broaden evaluation scope, we additionally include **Chinese LogiQA-V2** (a real-world Chinese logical reasoning benchmark adapted from the Chinese Civil Service Entrance Examination) and **AR-LSAT** (a real-world benchmark constructed from analytical reasoning questions in official LSAT exams, requiring multi-step reasoning over constraints to solve) and **FOLIO** (a human-annotated benchmark for natural language reasoning) in **Table R5.1-5.2**.

---

> ### Author Response · Authors · 2025-11-24
>
> **Table R4: Mean ± standard deviation over 3 prompt paraphrases. `*`  means statistical significance using pairwise t-test
> with p-value <0.05.**
>
> | Model       | Method       | ProofWriter       | ProntoQA         | LogiDeduct        | AR-LSAT           | FOLIO             | Chinese LogiQA-V2 |
> | ----------- | ------------ | ----------------- | ---------------- | ----------------- | ----------------- | ----------------- | ----------------- |
> | GPT-4       | w/o sparse   | 89.78 ± 0.35      | 99.20 ± 0.20     | 93.78 ± 0.19      | 50.29 ± 0.12      | 84.47 ± 0.28      | 74.11 ± 0.19      |
> | GPT-4       | CortexDebate | 90.78 ± 0.09      | 99.67 ± 0.31     | 92.44 ± 0.51      | 51.42 ± 0.98      | 85.13 ± 0.57      | 73.95 ± 0.22      |
> | GPT-4       | w/ sparse    | **91.83 ± 0.17*** | **99.87 ± 0.12** | **94.61 ± 0.26*** | **53.17 ± 0.14*** | **86.60 ± 0.43*** | **74.66 ± 0.14*** |
> | DeepSeek-V3 | w/o sparse   | 92.61 ± 0.38      | 99.80 ± 0.20     | 99.11 ± 0.51      | 73.31 ± 0.27      | 89.22 ± 0.49      | 85.75 ± 0.08      |
> | DeepSeek-V3 | CortexDebate | 92.83 ± 0.17      | 99.80 ± 0.20     | 99.67 ± 0.33      | 73.88 ± 0.25      | 89.18 ± 0.44      | 83.04 ± 0.27      |
> | DeepSeek-V3 | w/ sparse    | **93.50 ± 0.14*** | **99.93 ± 0.12** | **99.89 ± 0.19**  | **75.76 ± 0.44*** | **90.85 ± 0.28*** | **86.55 ± 0.48*** |
> ||
>
>
> **Table R5.1: Performance of different methods on GPT-4.**
>
> | Method                   | ProofWriter | ProntoQA    | LogiDeduct | AR-LSAT   | FOLIO      | Chinese LogiQA-V2 |
> | ------------------------ | ----------- | ----------- | ---------- | --------- | ---------- | ----------------- |
> | Direct Answer            | 53.50%      | 75.40%      | 59.00%     | 32.90     | 65.20      | 62.27%            |
> | CoT                      | 67.17%      | 81.20%      | 69.67%     | 35.06     | 70.59      | 65.22%            |
> | LogicLM                  | 79.17%      | 93.40%      | 87.00%     | 40.86     | 76.96      | 25.99%            |
> | SymbCoT                  | 82.33%      | 96.00%      | 86.33%     | 42.86     | 80.39      | 70.57%            |
> | CortexDebate (w/o NL-SL) | 73.50%      | 89.00%      | 84.33%     | 34.20     | 75.00%     | 68.17%            |
> | CortexDebate (w/ NL-SL)  | 90.83%      | 99.60%      | 92.33%     | 51.08     | 84.80%     | 74.13%            |
> | Ours (w/o sparse)        | 90.17%      | 99.40%      | 94.00%     | 50.42     | 84.31%     | 74.01%            |
> | **Ours (w/ sparse)**     | **92.00%**  | **100.00%** | **94.33%** | **53.25** | **86.27%** | **74.76%**        |
> ||
>
>
> **Table R5.2: Performance of different methods on DeepSeek-V3.**
>
> | Method                   | ProofWriter | ProntoQA   | LogiDeduct | AR-LSAT   | FOLIO     | Chinese LogiQA-V2 |
> | ------------------------ | ----------- | ---------- | ---------- | --------- | --------- | ----------------- |
> | Direct Answer            | 68.33       | 79.20      | 85.33      | 36.80     | 66.18     | 74.33             |
> | CoT                      | 71.83       | 85.00      | 83.00      | 45.45     | 76.96     | 77.97             |
> | LogicLM                  | 80.50       | 83.20      | 93.33      | 43.72     | 78.92     | 28.63             |
> | SymbCoT                  | 85.83       | 98.00      | 94.00      | 47.02     | 81.37     | 81.98             |
> | CortexDebate (w/o NL-SL) | 83.17       | 94.00      | 90.00      | 44.16     | 79.90     | 78.97             |
> | CortexDebate (w/ NL-SL)  | 93.00       | 99.80      | 99.67      | 74.03     | 88.73     | 83.04             |
> | Ours (w/o sparse)        | 92.83       | 99.80      | **100.00** | 73.62     | 89.22     | 85.68             |
> | **Ours (w/ sparse)**     | **93.33**   | **100.00** | **100.00** | **75.76** | **90.67** | **86.93**         |
> ||

---

> ### Author Response · Authors · 2025-11-24
>
> > Q1: "Please concretely differentiate your contributions from recent multi-agent debate and sparse/topology papers (methodologically, not just empirically). Could you add an ablation that (a) removes the SL-NL cross-paradigm stage and (b) replaces your sparse gating with a strongest prior baseline, to quantify each idea’s standalone lift?"
>
> **Response:**
>
> To quantify the standalone effect of each component in our method, we add the ablation variants in **Table R6**. The ablations are divided into two groups: **(A) SL–NL cross-paradigm ablations** and **(B) sparse-communication/topology ablations**.
>   - **SL–NL Cross-Paradigm Ablations.**
>     - **COT + P&S only (NL-only).** Remove all symbolic translators and solvers. Only two NL agents (Chain-of-Thought and Plan-and-Solve) debate.
>     - **LP + FOL + SAT only (SL-only).** Remove all NL agents. Keep only the three solver-based agents (LP/Pyke, FOL/Prover9, SAT/Z3)
>     - **No SL–NL interaction in reasoning debate**. Keep all 5 agents, but force SL agents to debate only with SL agents and NL only with NL (two disjoint debates).
>     - **Translation debate rounds = 0.** Disable the translation-stage debate (D_trans = 0). SL translations are generated once and used as-is by solvers.
>     - **Five-agent direct vote (no debate).** All 5 agents (SL and NL) answer once independently.
>   - **Sparse Communication / Topology Ablations.**
>     - **Pure NL 5-agent chat, fully-connected.** Remove all SL agents. Use 5 identical NL agents. Everyone reads everyone (full graph).
>     - **Pure NL 5-agent chat + SparseMAD.** Same pure NL setup. Replace communication with SparseMAD’s static neighbor graph.
>     - **Pure NL 5-agent chat + CortexDebate.** Same pure NL setup. Replace communication with CortexDebate’s trust-weighted sparse graph.
>     - **Pure NL 5-agent chat + our sparse gate.** Same pure NL setup, but use our **confidence + information-gain** gate.
>     - **Replace our sparse gate with SparseMAD (full SL+NL pipeline).** Use our full pipeline (translators, solvers, NL agents). Replace our gate with the SparseMAD topology.
>     - **Replace our sparse gate with CortexDebate (full SL+NL pipeline).** Same full pipeline. Replace our gate with CortexDebate’s trust graph.
>     - **Ours (full SL+NL pipeline + our sparse gate)**
>
> **Table R6. Ablations on (A) SL–NL cross-paradigm reasoning and (B) sparse communication strategies.**
> Accuracy (%). Columns = (GPT-4 / DeepSeek-V3) × (ProofWriter / ProntoQA / LogiDeduct)
>
> | Setting / Variant                                           | GPT4-PW   | GPT4-PQA   | GPT4-LD   | DS-PW     | DS-PQA     | DS-LD      |
> | ----------------------------------------------------------- | --------- | ---------- | --------- | --------- | ---------- | ---------- |
> | **(A) SL–NL Cross-Paradigm Ablations**                      |           |            |           |           |            |            |
> | COT + P&S only (NL-only)                                    | 79.33     | 95.60      | 84.67     | 86.17     | 96.00      | 93.00      |
> | LP + FOL + SAT only (SL-only)                               | 90.67     | 99.20      | 94.00     | 90.00     | 99.20      | 98.00      |
> | No SL–NL interaction in debate                              | 90.83     | 99.20      | 93.00     | 90.17     | 99.20      | 97.33      |
> | Translation debate rounds = 0                               | 89.17     | 99.40      | 90.00     | 92.67     | 99.60      | 97.33      |
> | 5-agent direct vote (no debate)                             | 86.50     | 97.00      | 82.67     | 90.00     | 97.60      | 91.67      |
> |                                                             |           |            |           |           |            |            |
> | **(B) Sparse Topology / Communication Ablations (5-agent)** |           |            |           |           |            |            |
> | Pure NL 5-agent chat, fully-connected                       | 73.00     | 91.40      | 84.00     | 82.83     | 93.00      | 88.33      |
> | Pure NL 5-agent chat + SparseMAD                            | 72.83     | 91.00      | 85.00     | 80.17     | 93.20      | 87.67      |
> | Pure NL 5-agent chat + CortexDebate                         | 73.50     | 89.00      | 84.33     | 83.17     | 94.00      | 90.00      |
> | Pure NL 5-agent chat + our sparse gate                      | 75.67     | 90.20      | 85.33     | 84.33     | 94.00      | 91.33      |
> |                                                             |           |            |           |           |            |            |
> | Replace our sparse gate w/ SparseMAD                        | 89.50     | 99.80      | 88.67     | 92.50     | 98.00      | 95.33      |
> | Replace our sparse gate w/ CortexDebate                     | 90.83     | 99.60      | 92.33     | 93.00     | 99.80      | 98.33      |
> | **Ours (full SL+NL, full debate, our gate)**                | **92.00** | **100.00** | **94.33** | **93.33** | **100.00** | **100.00** |
> ||

---

> ### Author Response · Authors · 2025-11-24
>
> > Q2: "How often do translation mistakes occur, and how does the system recover when all SL agents share the same systematic error?"
>
> **Response:**
>
> - **How often do translation mistakes occur?** We quantify SL translation failures in **Table R7**. For each of the three SL agents (LP/FOL/SAT) and each question, we mark the translation as correct or incorrect, and report the rate that **n** specific SL agents are all wrong (T-CER_n). Thus T-CER_1 is the average single-agent translation error rate, and **T-CER_3 is the probability that all three SL agents fail on the same example (upper-bounding the “shared systematic error” case)**.
>
> - **How does the system recover when all SL agents share the same systematic error?** The results in the **Table R7** shows that **“all SL agents sharing the same error” is relatively rare (T-CER_3)**. Even in these rare cases where all SL translations fail, the system is not forced into a bad answer:
>   - We still have two independent NL agents (CoT and Plan-and-Solve) that reason directly in natural language and do not rely on the symbolic forms.
>   - In the reasoning debate stage, **NL agents see the solver traces extracted from symbolic solvers**. When a wrong but executable symbolic translation leads to an inconsistent or implausible proof, the NL agents can explicitly challenge and overturn the SL conclusions during debate.
>
> **Table R7. Translation Common Error Rate (T-CER\_n) for 3 SL agents (LP/FOL/SAT).**
>
> | #SL agents | GPT-4 ProofWriter | GPT-4 ProntoQA | GPT-4 LogiDeduct | DeepSeek ProofWriter | DeepSeek ProntoQA | DeepSeek LogiDeduct |
> | ---------- | ----------------- | -------------- | ---------------- | -------------------- | ----------------- | ------------------- |
> | **1**      | 0.185000          | 0.103333       | 0.445556         | 0.176111             | 0.145333          | 0.535556            |
> | **2**      | 0.025000          | 0.012000       | 0.193333         | 0.014444             | 0.028667          | 0.137778            |
> | **3**      | 0.003333          | 0.002000       | 0.040000         | 0.001667             | 0.010000          | 0.056667            |
> ||
>
>
> > Q3: "Please include sensitivity analyses for λ and the similarity metric, and compare majority vote against confidence-weighted or learned adjudicators."
>
> **Response:** Thank you for the suggestions! **We added (i) a sensitivity analysis for the sparsity hyperparameter λ** **(Table R8)**, **(ii) a comparison of similarity metrics in the information-gain term** **(Table R9)**, **and (iii) different voting strategies** **(Table R3)**.
>
> - For **λ**,  accuracy is stable once λ ≥ 0.5 (variations ≈1 pp), while the token saving rate increases; so we choose λ = 1.0 as a simple acc–token trade-off.
> - For the **similarity metric**, cosine is better to ROUGE-L.
> - For **voting**, majority vote, confidence-weighted voting, and an LLM-as-judge differ by at most ≈1 pp, suggesting that our gains do not rely on a particular voting method.
>
> **Table R8. Sensitivity of accuracy and token saving rate to λ in the sparse gate.**
>  “Tok” = token saving rate.
>
> | λ    | Model    | PW Acc    | PW Tok    | PQA Acc    | PQA Tok   | LD Acc     | LD Tok    |
> | ---- | -------- | --------- | --------- | ---------- | --------- | ---------- | --------- |
> | 0    | GPT-4    | 90.17     | 8.52      | 99.20      | 11.45     | 92.67      | 5.63      |
> | 0.5  | GPT-4    | **92.17** | 13.62     | 99.60      | 18.31     | 94.00      | 10.29     |
> | 1.0  | GPT-4    | 92.00     | 17.03     | **100.00** | 22.89     | **94.33**  | 12.35     |
> | 1.5  | GPT-4    | 91.50     | 18.73     | 99.80      | 25.18     | 93.67      | 13.02     |
> | 2.0  | GPT-4    | 91.50     | **19.59** | **100.00** | **26.32** | 93.33      | **13.31** |
> |      |          |           |           |            |           |            |           |
> | 0    | DeepSeek | 92.17     | 3.88      | 99.60      | 18.01     | 98.33      | 17.29     |
> | 0.5  | DeepSeek | 93.00     | 6.02      | 99.80      | 28.82     | 98.67      | 27.66     |
> | 1.0  | DeepSeek | 93.33     | 13.15     | 100.00     | 36.02     | **100.00** | 34.57     |
> | 1.5  | DeepSeek | **93.67** | 15.15     | 99.60      | 36.53     | 99.33      | 38.44     |
> | 2.0  | DeepSeek | 93.33     | **15.34** | **100.00** | **36.87** | 99.67      | **39.31** |
> ||
>
>
> **Table R9. Sensitivity to the similarity metric.**
> Accuracies (%) for cosine similarity vs. ROUGE-L.
>
> | Metric  | Model    | PW Acc    | PQA Acc    | LD Acc     |
> | ------- | -------- | --------- | ---------- | ---------- |
> | Cosine  | GPT-4    | **92.00** | **100.00** | **94.33**  |
> | ROUGE-L | GPT-4    | 91.00     | 97.80      | **93.33**  |
> |         |          |           |            |            |
> | Cosine  | DeepSeek | **93.33** | **100.00** | **100.00** |
> | ROUGE-L | DeepSeek | 92.00     | **98.40** | 98.33      |
> ||

---

> ### Author Response · Authors · 2025-11-24
>
> Please let us know if we have properly addressed your questions and we are more than happy to discuss more!
>
> ------
>
> **References**
>
> [1] Pan, Liangming, et al. "Logic-lm: Empowering large language models with symbolic solvers for faithful logical reasoning." Findings of EMNLP 2023.
>
> [2] Ryu, Hyun, et al. "Divide and translate: Compositional first-order logic translation and verification for complex logical reasoning." ICLR 2025.
>
> [3] Xu, Jundong, et al. "Aristotle: Mastering logical reasoning with a logic-complete decompose-search-resolve framework." ACL 2025
>
> [4] Xu, Lei, et al. "Adaptive LLM-Symbolic Reasoning via Dynamic Logical Solver Composition." arXiv preprint 2025

---

> ### Author Response · Authors · 2025-11-27
>
> Dear reviewer DhnF,
>
> Thank you once again for your detailed review and the valuable time you have dedicated to our work! We have uploaded our revised PDF incorporating *all* mentioned changes, which highlighted in "$\textcolor{blue}{blue}$" for facilitating checking.
>
> FYI, we made the following changes:
>
> - In response to ```W1```, we **rewrite part of the Introduction to more clearly highlight our main idea**—combining heterogeneous SL/NL reasoning agents—rather than focusing the multi-agent debate framework itself.
>
> - In response to ```W2```, we **test LLMs for direct NL -> SL translation**, empirically showing LLMs have ability to perform valid NL to FOL/LP/SAT translation
>
> - In response to ```W3```, we **add a formal theoretical lower-bound analysis of majority-vote accuracy under heterogeneous SL/NL agents** in Section 4, with detailed proofs in Appendix E.
>
> - In response to ```W3```, we **provide empirical quantification of shared mistakes** in Table 21, showing that “many agents choosing the same wrong label” is rare and that our sparse gate further reduces shared spurious agreements.
>
> - In response to ```W3```, we also **add a sensitivity analysis on aggregation rules** (majority vote, confidence-weighted vote, LLM-as-judge) in Table 18.
>
> - In response to ```W4```, we **refine the notation and algorithmic description of the sparse communication mechanism** for better readability and implementation clarity in Section 3 and Appendix N.
>
> - In response to ```W5```, we **perform significance testing and prompt-sensitivity analysis by rewriting prompts into three paraphrases** in Table 4.
>
> - In response to ```W5```, we **add direct single-agent reasoning baselines (Direct Answer, CoT), extend evaluation to real-world datasets (FOLIO, AR-LSAT, Chinese LogiQA-V2)**, and **include smaller/ordinary base models (Qwen2.5-7B-Instruct, GPT-4o-mini)** in Tables 1–3 and 19 to demonstrate the effectiveness of our method across tasks and model scales.
>
> - In response to ```Q1```, we **add comprehensive ablation studies to quantify the standalone contribution of each component** (SL–NL cross-paradigm design and sparse communication) in Table 20.
>
> - In response to ```Q2```, we **provide detailed translation-quality analyses via Translation Common Error Rate (T-CER_n) and direct validation against gold FOL formulas on FOLIO** in Section 5.5 (Tables 8–9).
>
> - In response to ```Q3```, we **add sensitivity experiments on $\lambda$** in Table 7, **sensitivity experiments on aggregation rules** (majority vote, confidence-weighted vote, LLM-as-judge) in Table 18, and **sensitivity experiments on similarity metric** in Table 23.
>
> We sincerely hope the reviewer will re-evaluate our paper based on the additional experiments and explanations provided above. We look forward to your feedback if you have any further questions!
>
> Thanks for your time,
>
> Authors of #14334

---

### Author Response · Authors · 2025-11-27
**General responses and manuscript revision summary**

Dear reviewers and AC,

We sincerely thank all reviewers and AC for their great effort and constructive comments on our manuscript. During the authors-reviewers  period, we have been focusing on these beneficial suggestions from the reviewers and doing our best to add many experiments and revise our manuscript. We believe our current carefully revised manuscript can address all the reviewers’ concerns.

As reviewers highlighted, we believe our paper introduces a **novel combination of symbolic and natural-language reasoning** (Reviewer DhnF) , **achieving very good performance on multiple datasets** (Reviewer awiG). We also appreciate that the reviewers found the proposed **adaptive sparse communication strategy** to be **a very important advantage** that **effectively reduces token cost with minimal accuracy loss** (Reviewer UAB7, Reviewer DhnF). Reviewers also commended the **technical soundness** (Reviewer Q5K8), **strong empirical performance and ablation studies** (Reviewer DhnF) and **impressive results** (Reviewer UAB7), noting that the **experiments are very substantial** (Reviewer awiG) with a **sufficient number of baselines** (Reviewer UAB7).

Moreover, we thank the reviewers for pointing out limitations of the evaluation scope, specifically for the suggestions for verifying performance on **more challenging datasets** (Reviewer awiG, Reviewer Q5K8). We also appreciate comments for more comprehensive analysis, including **sensitivity analyses for $\lambda$** (Reviewer DhnF),  **validation of the NL-to-SL translations** (Reviewer UAB7, Reviewer DhnF), **token cost comparison and timing analysis** (Reviewer Q5K8), **confidence score acquisition** (Reviewer awiG), **more analysis and comparison for majority-vote** (Reviewer UAB7, Reviewer DhnF), **significance testing** with **variance estimates** (Reviewer DhnF, Reviewer Q5K8), and **applicability to smaller models** (Reviewer awiG). In response to these comments, we have carefully revised and enhanced our manuscript with the following important changes with the added experiments:

- [Reviewer DhnF] We **rewrite part of the Introduction to more clearly highlight our main idea**—combining heterogeneous SL/NL reasoning agents—rather than focusing the multi-agent debate framework itself.

- [Reviewer DhnF] We **add comprehensive ablation studies to quantify the standalone contribution of each component** (SL–NL cross-paradigm design and sparse communication) in Table 20.

- [Reviewer DhnF, Reviewer UAB7] We **add a formal theoretical lower-bound analysis of majority-vote accuracy under heterogeneous SL/NL agents** in Section 4, with detailed proofs in Appendix E.

- [Reviewer DhnF, Reviewer UAB7] We **provide empirical quantification of shared mistakes** in Table 21, showing that “many agents choosing the same wrong label” is rare and that our sparse gate further reduces shared spurious agreements.

- [Reviewer DhnF, Reviewer UAB7] We **add a sensitivity analysis on aggregation rules** (majority vote, confidence-weighted vote, LLM-as-judge) in Table 18.

- [Reviewer DhnF, Reviewer UAB7] We **provide detailed translation-quality analyses via Translation Common Error Rate (T-CER_n) and direct validation against gold FOL formulas on FOLIO** in Section 5.5 (Tables 8–9).

- [Reviewer awiG] We **analytically and empirically show the effectiveness of LLM-generated confidence scores in our approach** in Appendix J.

- [Reviewer DhnF, Reviewer Q5K8] We **add sensitivity experiments on $\lambda$** in Table 7.

- [Reviewer DhnF, Reviewer Q5K8] We **perform significance testing and prompt-sensitivity analysis by rewriting prompts into three paraphrases** in Table 4.

- [Reviewer DhnF, Reviewer awiG, Reviewer Q5K8] We **add direct single-agent reasoning baselines (Direct Answer, CoT), extend evaluation to real-world datasets (FOLIO, AR-LSAT, Chinese LogiQA-V2)**, and **include smaller/ordinary base models (Qwen2.5-7B-Instruct, GPT-4o-mini)** in Tables 1–3 and 19 to demonstrate the effectiveness of our method across tasks and model scales.

- [Reviewer Q5K8] We **compare token cost and accuracy between Direct Answer, CoT, and our method**, clarifying that our token-saving claim is defined relative to a fully connected multi-agent debate baseline in Tables 14–15.

- [Reviewer Q5K8] We **add timing analysis for symbolic solvers** to show that solver timeouts are rare and do not dominate overall cost in Table 10.

- [Reviewer Q5K8] We **introduce vote-entropy–based consensus analysis** to show that sparse pruning preserves (and sometimes slightly improves) consensus in multi-agent debate in Appendix F.

These updates are temporarily highlighted in "$\textcolor{blue}{blue}$" for facilitating checking.

We hope our response and revision could address all the reviewers' concerns, and are more than eager to have further discussions with the reviewers in response to these revisions.

Many thanks,

Submission14334 Authors

---

### Public Comment · ~Zhiyu_Ni1 · 2026-04-28
**can the author open-source the code?**

may i ask when will the authors open-source the code? i did not find it in the paper or github.

---

### Meta-Review · Area_Chair_ZmWF · 2026-01-04

**Summary:**

The paper presents a well-engineered framework that effectively combines symbolic and natural-language reasoning through multi-agent debate. It achieves strong empirical performance across a wide range of logical reasoning benchmarks. The adaptive sparse communication mechanism is a practical contribution that significantly reduces token and computation costs with minimal or no loss in accuracy. One reviewer raised concerns that the core ideas largely extend existing multi-agent debate and sparse communication frameworks, with limited conceptual novelty over prior work. Some reviewers questioned whether gains stem from fundamentally new insights or from combining known components with powerful LLM backends.

**Reviewer Concerns:**

The relationship to prior methods and where the gains come from are two major concerns, from reading the review comments. I believe the authors did a decent job addressing both of them.

**Reviewer Scores:**

I think the two negative reviewers are likely to increase their scores based on the rebuttal. One reviewer has a back-and-forth discussion already with the authors and indicated satisfaction at the end of the discussion.

---

### Decision · Program_Chairs · 2026-01-26

Accept (Poster)